# Conformal Policy Control

**Drew Prinster** [1][2]  **Clara Fannjiang** [1]  **Ji Won Park** [1]  **Kyunghyun Cho** [3][4]  **Anqi Liu** [2]  **Suchi Saria** [2]
**Samuel Stanton** [4]

## Abstract

An agent must try new behaviors to explore and improve. In high-stakes environments, an agent that violates safety constraints may cause harm and must be taken offline, curtailing any future interaction. Imitating old behavior is safe, but excessive conservatism discourages exploration. How much behavior change is too much? We show how to use any safe reference policy as a probabilistic regulator for any optimized but untested policy. Conformal calibration on data from the safe policy determines how aggressively the new policy can act, while provably enforcing the user's declared risk tolerance. Unlike conservative optimization methods, we do not assume the user has identified the correct model class nor tuned any hyperparameters. Unlike previous conformal methods, our theory provides finite-sample guarantees even for non-monotonic bounded loss functions. Our experiments on applications ranging from natural language question answering to biomolecular engineering show that safe exploration is not only possible from the first moment of deployment, but can also improve performance.

## 1. Introduction

Safe exploration is one of the most studied problems in AI safety (Amodei et al., 2016). The choice between probable safety and the hope of possible discovery presents a dilemma known by many names, from the explore-exploit tradeoff in reinforcement learning to the validity-power tradeoff in statistics. The optimal balance is problem-dependent, yet general solutions must be problem-independent. So, to apply to real-world problems, any general solution must en-

code this balance as a hyperparameter, which then must be tuned from data or derived from assumptions about problem structure. In consequential settings, collecting data from an unsafe behavior policy (a context-dependent strategy or distribution over actions) may be unacceptable. Deriving the balance from assumptions about the problem reintroduces a subtler form of problem-dependence, namely the practitioner's understanding of the setting itself. Yet consequential, poorly understood settings are precisely where machine learning offers the greatest potential benefit.

Consider an agent that observes contexts (i.e., states) and takes actions to maximize reward while satisfying safe operating constraints to within some tolerance. For example, a language model responding to medical questions aims to give answers that are as helpful as possible while avoiding false claims. In biomolecular engineering, a generative model proposing improvements to lead molecules aims to maximize efficacy of its designs while ensuring that they are synthesizable. In such settings, the agent has candidate policies optimized for performance but does not yet know which, if any, are safe to deploy. From past experience, the agent usually has a default safe policy it knows controls the risk, but playing it safe may be far from optimal.

What the safe policy does supply is calibration data: Reweighting each observation by a candidate-to-safe likelihood ratio gives an estimate of each candidate's risk. But which candidate should the agent deploy? The natural move is to estimate every candidate's risk and pick the most aggressively optimized one that still looks safe, but this approach introduces bias: Selection corrupts the procedure by favoring estimates that understate their true risk. Yet the change in risk the agent must control is due to the policy shift it will author. The agent does not know how the world will respond to a given action, but it knows exactly how each candidate would distribute its actions, which is the only difference between the distributions it has experienced in the past and those it would encounter in the future.[1]

We introduce conformal policy control (CPC, Figure 1), a method for safe exploration that uses the calibration data to interpolate between any safe and any optimized policy while provably respecting the risk tolerance. CPC requires

[1]Prescient Design, Genentech, U.S.A. [2]Johns Hopkins University, Baltimore, U.S.A. [3]New York University, NYC, U.S.A. [4]Formerly Prescient Design, Genentech. Correspondence to: Drew Prinster <drewprinster@gmail.com>, Samuel Stanton <sdstanton1@gmail.com>.

*Proceedings of the 43rd International Conference on Machine Learning*, Seoul, South Korea. PMLR 306, 2026. Copyright 2026 by the author(s).

---

[1]If we assume the environment is stationary.

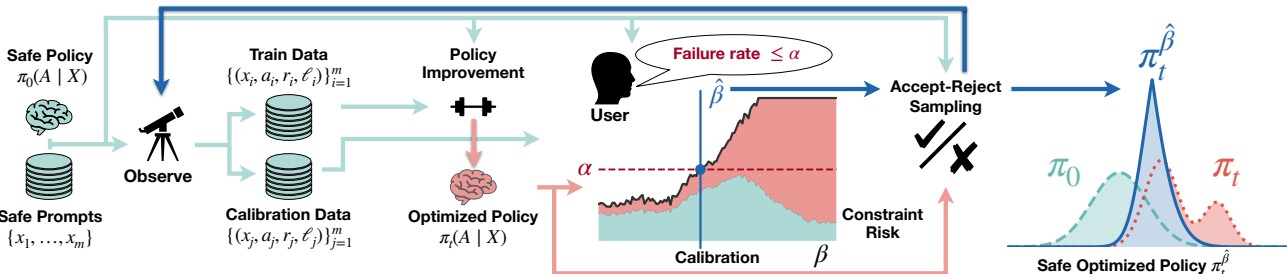

*Figure 1.* An illustration of safe exploration with conformal policy control. Starting with a safe policy $\pi_0$ and safe context prompts $\mathcal{X}_0 := \{x_1, \ldots, x_m\}$, we observe the safe policy's actions $a_i$ and corresponding reward $r_i \in \mathbb{R}$ and constraint violation $\ell_i \in \{0, 1\}$. We split the observations into train and calibration data, and optimize the safe policy to obtain $\pi_t$. We then query the user for their constraint violation risk tolerance $\alpha$, and apply conformal risk control to find a bound on the likelihood ratio $\pi_t/\pi_0$ which guarantees the calibrated risk is controlled at level $\alpha$. Finally we apply rejection sampling to probabilistically regulate the optimized policy for deployment and observation, allowing the user to flexibly trade off reward, constraint risk, and test-time compute.

no assumptions on the reward or constraint functions, no access to the optimized policy's training process, and no additional samples to tune the balance beyond what a safe policy already provides. The key idea is to invert the usual importance-weighting paradigm: Rather than estimating risk for a fixed policy, we search over candidate policies, parameterized by a likelihood-ratio threshold, for the most aggressive one such that all evaluated candidates still satisfy the risk tolerance (with a weighted conservative adjustment). Conformal calibration on the safe policy's data determines both the weights and the threshold. At deployment, the interpolated policy can be realized through rejection sampling (or other Monte Carlo methods), which allows the agent to probabilistically self-regulate to a zone of competence determined by its calibration data. Because CPC operates entirely at test time, the same safe and optimized policies can be reused under different risk tolerances, trading test-time compute for risk control without retraining.

Providing finite-sample guarantees in this setting poses several challenges for uncertainty quantification. The deployed policy depends on the calibration data, introducing subtle, data-dependent distribution shift. The relevant losses are general constraint functions, not just coverage indicators. Conformal risk control (CRC) (Angelopoulos et al., 2024) can address these challenges, but assumes a monotone relationship between the control parameter and the loss. In CPC, the loss function does not depend on the control parameter, so the monotonicity condition does not hold. We generalize CRC (gCRC) to non-monotonic bounded loss functions, and we prove that CPC maintains finite-sample guarantees when the control parameter governs the policy, not the loss.

We validate our methods on three tasks. In medical question answering, gCRC controls the false discovery rate, a non-monotonic loss that standard CRC cannot handle, achieving tighter control and better claim recall than existing methods. In constrained active learning, CPC provides finite-sample guarantees under feedback-loop covariate shifts where pre-

vious approaches offered only asymptotic control. In black-box biomolecular sequence optimization, CPC extends conformal methods to a setting where no prior guarantees existed, and we find that moderate risk control can counterintuitively improve optimization performance by reducing wasted evaluations on infeasible actions. Safe exploration is not only safe, it can also be more efficient.

## 2. Preliminaries and Background

**Notation:** We consider a decision-making agent that interacts with an environment over a sequence of rounds. At each round $t$, the agent observes a context (i.e., prompt) $X_t \in \mathcal{X}$ and selects an action $A_t \in \mathcal{A}$ according to a policy $\pi(A \mid X)$. If the policy is parameterized by an LLM, then $\mathcal{A} \subseteq \mathcal{X}$. The agent then observes (potentially noisy) rewards $R_t \in \mathbb{R}$ and losses $L_t \in \mathbb{R}$. Throughout, uppercase letters denote random variables and lowercase letters their realized values. The reward function $r(x, a)$ quantifies the degree to which a context-action pair is aligned with the agent's objective, and the loss function $\ell(x, a)$ quantifies the degree to which the context-action pair violates the agent's constraints. While a policy can be defined implicitly by solving an optimization problem with respect to estimated reward and loss surrogates (Watkins & Dayan, 1992; Jones et al., 1998), in this paper we focus on controlling explicit policies parameterized by a conditional distribution (Williams, 1992), such as an autoregressive transformer.

### 2.1. The Safe Policy Improvement Problem

The setting just introduced provides a general framework for sequential decision making under uncertainty (Langford & Zhang, 2008), capturing many problems of practical interest (Li et al., 2010). In natural language question answering (Rajpurkar et al., 2016), the context could be a user query, the action a generated response, and the reward and loss could reflect response completeness and (in)correctness, respectively. In constrained black-box optimization (Gardner

et al., 2014), the context could be the previous evaluations, the action the next solution to evaluate, and the reward and loss functions could be black-box oracle objective and constraint functions, respectively (Chen et al., 2025). For example, in protein engineering, the loss function may indicate whether a proposed sequence lies outside a feasibility set $\mathcal{F}$ of expressible proteins, i.e., $\ell(X_t, A_t) := \mathbb{1}\{A_t \notin \mathcal{F}\}$.

The agent's goal is to learn a policy, $\pi$, that maximizes expected reward while controlling expected loss:

$$\max_{\pi} \quad \mathbb{E}_{X \sim p} \, \mathbb{E}_{A \sim \pi(\cdot|X)} \left[ r(X, A) \right]$$
$$\text{subject to} \quad \mathbb{E}_{X \sim p} \, \mathbb{E}_{A \sim \pi(\cdot|X)} \left[ \ell(X, A) \right] \leq \alpha, \quad (1)$$

where $\alpha \in [0, B]$ (for some $B < \infty$) is a user-specified risk tolerance, and $p(X)$ is the marginal context distribution.[2]

In practice, the agent usually begins with a safe reference policy $\pi_0$, whether inherited from a prior deployment or trained as a density model of known safe actions, and seeks to improve upon it. In the LLM literature, policy improvement is also known as post-training and encompasses methods such as supervised fine-tuning (SFT; Wei et al., 2022), reinforcement learning with human feedback (RLHF; Christiano et al., 2017) or direct preference optimization (DPO; Rafailov et al., 2023). The quality of the improved policy depends on the post-training dataset, the fidelity of any reward or preference models, and design decisions governing how far the policy may diverge from the safe reference.

Post-training design decisions immediately prompt two questions: which policy divergence measure is best, and how much divergence is allowable to maintain an acceptable risk profile? The first question has been studied extensively (see Section 3). The second, however, is typically treated as a hyperparameter tuning problem (Bergstra et al., 2011; Feurer & Hutter, 2019), in which the user must specify a constraint on a "control parameter" (such as a divergence budget or penalty weight) and the algorithm optimizes subject to that constraint. The resulting indirection creates a gap between what the user actually wants and the control parameters that are accessible in available algorithms (Hutchins et al., 1986; Norman, 1988). The user's goal is declarative (i.e., a desired outcome of the program): control risk at some level $\alpha$. The algorithms demand imperative instructions (i.e., the steps the program should execute): bound divergence at some level, or set the penalty weight to some value. Without a principled translation, the user must learn the mapping from outcome to instruction by trial and error, spending the very samples the agent seeks to use efficiently.

As previewed in the introduction, importance weighting offers a more direct route. Given data from the safe policy, the agent can estimate the expected loss under any candidate policy by reweighting each observation by the likelihood

ratio between the candidate and safe policies, via the identity

$$\mathbb{E}_{\substack{X \sim p(\cdot) \\ A \sim \pi_t(\cdot|X)}} \left[ \ell(X, A) \right] = \mathbb{E}_{\substack{X \sim p(\cdot) \\ A \sim \pi_0(\cdot|X)}} \left[ \frac{\pi_t(A \mid X)}{\pi_0(A \mid X)} \ell(X, A) \right].$$

Bounding this ratio at some threshold $\beta$ simultaneously defines a constrained policy and bounds the importance weights, so the safe policy's data suffices to determine whether the candidate still respects the risk tolerance.[3]

## 2.2. Conformal Risk Control

What remains is a principled way to choose $\beta$ from finite safe-policy data, with a guarantee on the risk of the resulting deployed policy. Conformal risk control (CRC; Angelopoulos et al., 2024) provides this for a restricted class of problems: it sets a control parameter $\lambda$ to control the expected value of a user-specified loss $L_i(\lambda)$. Given exchangeable (e.g., IID) calibration loss functions, $L_1(\cdot), \ldots, L_n(\cdot)$, and test loss function, $L_{n+1}(\cdot)$, the key assumption is that each $L_i(\lambda)$ is monotonically nonincreasing with the control parameter, $\lambda$. For example, conformal prediction (Vovk et al., 2005) is a special case of CRC where the loss is the miscoverage indicator $L(\lambda) = \mathbb{1}\{Y \notin \hat{C}_\lambda(X)\}$ for labels $Y$ and prediction sets $\hat{C}_\lambda$, and $\lambda$ controls the size of the prediction set: increasing the prediction set size can only decrease miscoverage. CRC extends conformal prediction to cases such as setting an inclusion threshold on predicted probabilities for multilabel classification with false negative rate (FNR) control (i.e., controlling the expected proportion of true labels that are not included in the set). Like miscoverage, FNR is monotonic in $\lambda$: adding labels to the prediction set can only increase the proportion of true labels contained. By selecting $\hat{\lambda}$ as the smallest value such that the (conservatively adjusted) empirical risk falls below $\alpha$, CRC attains the finite-sample guarantee $\mathbb{E}[L_{n+1}(\hat{\lambda})] \leq \alpha$. In each case, increasing $\lambda$ can only decrease the loss. This monotonicity is what lets CRC search from aggressive to conservative and stop at the first value that satisfies the guarantee.

For policy control, however, it is not immediately clear how CRC could be applied. Many losses of practical interest admit no tunable control parameter at all. Consider again the feasibility indicator $L_t := \mathbb{1}\{A_t \notin \mathcal{F}\}$: a prediction set $\hat{C}_\lambda$ can be grown to reduce miscoverage, but the feasibility set $\mathcal{F}$ is fixed by the environment, and there is no $\lambda$ that makes a given action more or less feasible. We instead place the control parameter on the policy, governing the distribution of actions rather than the loss they incur. Even so, the resulting expected loss is not necessarily monotonic in that parameter, a theoretical challenge we address in Section 4.

---

[2]In constrained optimization notation, $r$ and $\ell$ map to $f$ and $g$.

[3]Bounding the likelihood ratio $\pi_t(a \mid x)/\pi_0(a \mid x) \leq \beta$ uniformly implies a bound on the KL divergence and other $f$-divergences between $\pi_t$ and $\pi_0$. In this sense, likelihood ratio clipping is a strict form of divergence constraint.

# 3. Abbreviated Related Work

**Conservative Model-Based Optimization:** A standard approach to safe policy improvement relies on the observation that the risk of a new policy can be bounded in terms of the risk of a reference policy plus a penalty that grows with the divergence between them (Kakade & Langford, 2002). When the reference policy is known to satisfy the safety constraint, controlling this divergence provides a mechanism for controlling risk. The idea of bounding risk through bounded divergence motivates a broad family of locality-constrained policy optimization methods (Kim et al., 2025). Prior work has proposed entropy-regularized control and KL-penalized objectives (Todorov, 2009; Fox et al., 2016), trust-region policy optimization (Schulman et al., 2015; 2017), conservative value penalties in offline reinforcement learning (Kumar et al., 2020; Trabucco et al., 2021), trust-region methods in black-box optimization (Eriksson et al., 2019), and uncertainty-penalized safe Bayesian optimization (Sui et al., 2015). Damani et al. (2023) proposed using a likelihood ratio as a safety filter for model-based optimization, but treated the ratio threshold as a hyperparameter.

From an optimization perspective, our work is most similar to Stanton et al. (2023), who proposed constraining a Bayesian optimization policy to regions where the resulting conformal prediction sets would have bounded width given a declared miscoverage tolerance $\alpha$. However, their approach was limited by three weaknesses. First, their approach was restricted to the miscoverage loss. Second, their use of an implicit policy created an awkward fixed-point problem: finding the policy that induced an action distribution that controlled the coverage. Third, they did not fully address the multi-step feedback covariate shift induced by sequential model-based optimization.[4] Indeed, Stanton et al. (2023) noted that, at the time, the existing conformal theory had not yet proved general enough guarantees for that setting.

**Conformal Methods:** Stanton et al. (2023) drew motivation from Fannjiang et al. (2022), who first formalized the phenomenon of *feedback covariate shift* to generalize conformal prediction to the setting of a single round of model-based optimization, building on techniques introduced by Tibshirani et al. (2019). Prinster et al. (2024) further extended these techniques to generalize conformal prediction for any data distribution, including showing coverage guarantees for the multi-round model-based optimization setting.

However, the techniques used throughout this line of work do not enable one to prescribe a data distribution (e.g., select a policy) that guarantees coverage. That is, these methods can provide prediction sets with coverage guarantees for actions generated by any *fixed* policy (a "descriptive" use

---

[4]A form of covariate shift in which the test distribution is dependent on previously observed data.

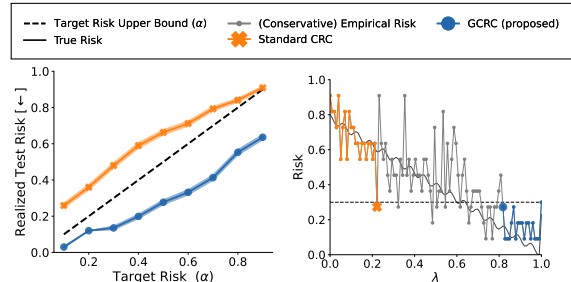

*Figure 2.* Empirical comparison of standard CRC (Angelopoulos et al., 2024) (orange) and the proposed generalized CRC (blue) on synthetic non-monotonic losses. gCRC achieves risk control across various target risk levels while CRC does not (**left**). On an example empirical risk trajectory for these experiments, CRC underestimates the risk while gCRC maintains risk control by searching from safe to aggressive hyperparameter values (**right**).

of conformal techniques), but using those prediction sets to then *select* a policy (a "prescriptive" use) introduces further, unaccounted-for dependence between the observed data and test policy that nullifies coverage guarantees.

To enable this prescriptive use case, we build upon techniques from conformal risk control (CRC) (Angelopoulos et al., 2024), a general framework for selecting a hyperparameter with finite-sample guarantees for any monotonic loss, of which conformal prediction can be cast as a special case. When the loss of interest is non-monotonic, Angelopoulos et al. (2024) also showed that deploying the procedure on a monotonic relaxation of the losses provides asymptotic guarantees of control. However, they left the problem of finite-sample guarantees for non-monotonic losses open. In the next section, we develop a variant of CRC that drops the monotonicity requirement, and instead relies on smoothness of the losses and a form of stability of the calibration procedure to control the risk on the test point. Finally, we apply techniques from Prinster et al. (2024) to prove finite-sample guarantees for our procedure in the multi-round model-based optimization setting. See Appendix A for an extended discussion of related work.

Concurrently, Angelopoulos (2026) also present CRC guarantees for non-monotonic losses, but for parameterized losses, rather than the policy-control setting we introduce here where the control parameter tunes the policy instead of the loss. Even in the comparable setting of parameterized losses, however, the contributions are distinct, with neither subsuming the other: Relative to the univariate method in Angelopoulos (2026), our method (Section 4.1) will always be at least as conservative *a priori* (e.g., as in Figure 2, right); consequently, whereas that paper's analogous guarantee relies on leave-one-out stability, our results use replace-one stability, which is generally a more relaxed assumption (Bousquet & Elisseeff, 2002; Shalev-Shwartz et al., 2010).

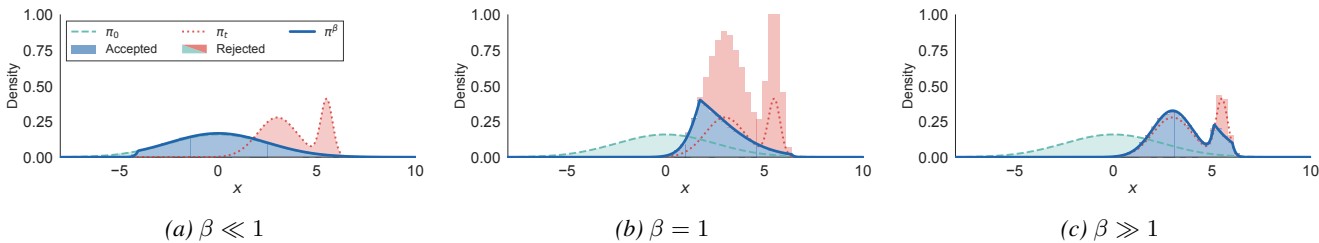

*Figure 3.* Rejection sampling for policy interpolation at three $\beta$ values. As $\beta$ increases from near-zero to large values, the constrained policy $\pi_t^{(\beta)}$ interpolates from the safe policy $\pi_0$ to the optimized policy $\pi_t$. Blue histogram bars show accepted samples matching $\pi_t^{(\beta)}$, while teal/red bars show rejected samples from the proposal distribution. (a) Small $\beta$ nearly recovers the safe policy (b) Intermediate $\beta$ shows intermediate interpolation with frequent rejections. (c) Large $\beta$ approaches the optimized policy with minimal rejection.

## 4. Methods and Theory

### 4.1. Extending Conformal Risk Control to Non-Monotonic Losses

We begin with intuition. Standard CRC's proof (Angelopoulos et al., 2024) has two key steps: First, prove that the oracle solution (using knowledge of the test point's loss), $\lambda^*$, achieves risk control; and second, show that the empirical CRC solution (with a conservative adjustment for the unknown test loss), $\hat{\lambda}$, is deterministically greater than or equal to the oracle, $\hat{\lambda} \geq \lambda^*$. Note that here, larger $\lambda$ are safer. In particular, for standard CRC, monotonicity of the loss function is then necessary and sufficient[5] for concluding that the empirical CRC algorithm's risk is controlled: $\mathbb{E}[L_{n+1}(\hat{\lambda})] \leq \mathbb{E}[L_{n+1}(\lambda^*)] \leq \alpha$.

Without monotonicity, what can be done? The crux is how to find a modified oracle solution, $\lambda_+^*$, and a modified empirical algorithm, $\hat{\lambda}_+$, such that establishing the relation $\hat{\lambda}_+ \geq \lambda_+^*$ enables analyzing the risk of the empirical algorithm *by construction*, rather than by monotonicity. Consider the family of hyperparameters defined as

$$\widehat{\lambda}_+(\langle L_{1:m}\rangle, \alpha) \qquad (2)$$
$$:= \inf\left\{\lambda_0 \in \Lambda \ : \ \forall \lambda \geq \lambda_0, \ \sum_{i=1}^{m} \frac{L_i(\lambda)}{m} \leq \alpha\right\},$$

when the set is nonempty, and otherwise define $\widehat{\lambda}_+(\langle L_{1:m}\rangle, \alpha) := \lambda_{\max}$, where $\lambda_{\max}$ is a "safe hyperparameter" assumed to exist as in Angelopoulos et al. (2024): $\lambda_{\max} := \sup \Lambda \in \Lambda$ and $L_i(\lambda_{\max}) \leq \alpha$. In words, $\widehat{\lambda}_+$ is the smallest (most aggressive) hyperparameter such that the empirical risk is controlled at level $\alpha$ *for all hyperparameters at least as large as it*; this "for all $\lambda \geq \lambda_0$" qualification is the key algorithmic modification to standard CRC. We then define the oracle solution by Eq. (2) fit on the bag (or multiset) containing both the calibration and test loss

---

[5]Sufficient given the other CRC assumptions: boundedness and right-continuity of the loss, and existence of a safe $\lambda_{\max}$.

functions, $\langle L_{1:n+1}\rangle$:

$$\lambda_+^* := \widehat{\lambda}_+(\langle L_{1:n+1}\rangle, \alpha). \qquad (3)$$

Similarly, define the empirical algorithm as Eq. (2) fit on the bag of the calibration losses, with the bound $B$ in place of $L_{n+1}$ to conservatively adjust for the unknown test loss:

$$\hat{\lambda}_+ := \widehat{\lambda}_+(\langle L_{1:n}, B\rangle, \alpha). \qquad (4)$$

Our empirical algorithm, $\hat{\lambda}_+$, reduces to CRC when the losses are monotonic, and it can be viewed as searching hyperparameter space from *a priori* safest to most aggressive, the opposite direction of standard CRC (Figure 2). Also note that our generalized CRC (gCRC) is conservatively designed to be *a priori* safer than the oracle, that is, $\hat{\lambda}_+ \geq \lambda_+^*$ almost surely. Unfortunately, this construction is not itself sufficient for $\hat{\lambda}_+$ to achieve validity at level $\alpha$ (Appendix B.3). Relating the empirical algorithm to the oracle, however, does provide finite-sample guarantees for $\hat{\lambda}_+$ under Lipschitz continuity and depending on algorithmic stability.

**Definition 4.1** ($\epsilon$-replace-one stability). We say that a function $h$ is $\epsilon$-replace-one stable if for all $i, j \in [n+1]$,

$$\mathbb{E}\left[\left|h(\langle Z_{1:n+1}\rangle \backslash\{z_i\}) - h(\langle Z_{1:n+1}\rangle \backslash\{Z_j\})\right|\right] \leq \epsilon.$$

The following result then provides finite-sample guarantees for $\hat{\lambda}_+$ even for non-monotonic losses.

**Theorem 4.2.** *Assume exchangeable $L_i(\lambda)$. Define $\lambda_{\max} := \sup \Lambda \in \Lambda$ and assume*

$$L_i(\lambda_{\max}) \leq \alpha, \qquad \sup_{\lambda} L_i(\lambda) \leq B < \infty \quad almost\ surely.$$

*If the $L_i(\lambda)$ are $K$-Lipschitz in $\lambda$ and $\hat{\lambda}_+$ is $\epsilon$-replace-one stable, then*

$$\mathbb{E}\left[L_{n+1}(\hat{\lambda}_+)\right] \leq \alpha + K\epsilon.$$

**Proof sketch:** Conditional on the bag of data, $\langle z_{1:n+1}\rangle$, we show that the oracle $\lambda_+^*$ achieves risk control, which implies

**Algorithm 1** Pseudo-code for $\beta$ calibration.

**Require:** Safe policy $\pi_0$; optimized policies $\pi_1, ..., \pi_t$; sets of prompts $X^{(0)}, ..., X^{(t)}$; calibration data $\mathcal{D}_{\text{cal}}$; proposal data $\mathcal{D}_{\text{prop}} \sim \pi_t$; target risk $\alpha$; loss upper bound $B$.

**Ensure:** Likelihood ratio bound $\hat{\beta}$

$\rho_i \leftarrow \pi_t(A_i|X^{(t)}) / \pi_0(A_i|X^{(0)})$ for $i \in \mathcal{D}_{\text{cal}} \cup \mathcal{D}_{\text{prop}}$

$G \leftarrow \text{PREPAREGRID}(\{\rho_i\})$

Initialize: $\hat{\beta} \leftarrow \beta_{\min}$

**for** $\beta \in G$ (ascending) **do**

    # Compute constrained PDF, mixture of past PDFs

    $\psi_\beta \leftarrow \text{ESTIMATEPSI}(\{\rho_i\}, \beta)$

    $\pi_t^{(\beta)}(A_i) \leftarrow \min(\pi_t(A_i), \beta \cdot \pi_0(A_i)) / \psi_\beta$ for $i \in \mathcal{D}_{\text{cal}}$

    $\pi_{0:t-1}^{\text{mix}}(A_i) \leftarrow \text{MIXPDF}(A_i, \pi_{0:t-1})$ for $i \in \mathcal{D}_{\text{cal}}$

    # Compute conformal weights, check empirical risk

    $w_i \leftarrow \pi_t^{(\beta)}(A_i)/\pi_{0:t-1}^{\text{mix}}(A_i)$ for $i \in \mathcal{D}_{\text{cal}}$

    $w_{\max} \leftarrow \text{ESTIMATEWMAX}(\{w_i\}, \pi_0, \pi_t)$

    $\tilde{w}_i \leftarrow w_i/(\sum_{j \in \mathcal{D}_{\text{cal}}} w_j + w_{\max})$ for $i \in \mathcal{D}_{\text{cal}}$

    **if** $\sum_{i \in \mathcal{D}_{\text{cal}}} \tilde{w}_i \cdot l_i + \tilde{w}_{\max} \cdot B > \alpha$ **then**

        **Return:** $\hat{\beta}$

    **end if**

    $\hat{\beta} \leftarrow \beta$

**end for**

**Return:** $+\infty$

---

risk control for all $\lambda \geq \lambda_+^*$ that are constant conditional on $\lfloor z_{1:n+1} \rfloor$. We then show $\hat{\lambda}_+ \geq \lambda_+^*$ almost surely. Lastly, using Lipschitz continuity and replace-one stability, we relate the risk of $\hat{\lambda}_+$ to that of some $\lambda \geq \lambda_+^*$ that is constant conditional on $\lfloor z_{1:n+1} \rfloor$. We defer the proof to App. B.1.

*Remark* 4.3 (Attaining $\alpha$-validity if $K$ is known). If $K$ is known, then a more conservative significance level, $\hat{\alpha} \leq \alpha$, can be used for level $\alpha$ validity, regardless of instability:

$$\mathbb{E}\left[L_{n+1}\left(\hat{\lambda}_+(\hat{\alpha})\right)\right] \leq \alpha,$$

where $\hat{\alpha}$ is defined in Appendix B.4.

*Remark* 4.4 (Monotone envelope interpretation). The $\hat{\lambda}_+$ in Eq. (4) can be viewed as the smallest hyperparameter such that the "monotonized empirical risk" is controlled at $\alpha$. The procedure is presented with this interpretation in Angelopoulos et al. (2024)'s App. A, but that paper noted that finite-sample guarantees for $\hat{\lambda}_+$ were at the time unknown.

## 4.2. Conformal Policy Control: Selecting a Risk-Controlling Policy

We first state the only critical assumptions required by CPC: access to a safe policy and subsequent optimized policies. For ease of notation we will describe CPC in an online setting where one action is taken or sample generated each

policy improvement step, but it also works similarly in a multiround batch setting.

**Assumption 1: An Initial Safe Policy** Assume access to an initial safe policy, $\pi_0$, for which the true risk is controlled at the desired level $\alpha$; that is, assume $\mathbb{E}_{X \sim p} \mathbb{E}_{A \sim \pi_0(\cdot|X)} [\ell(X, A)] \leq \alpha$ holds for $\pi_0$. Note that this assumption is weaker than it may appear: any existing policy (a previous deployment, a domain default, or even a uniform baseline) qualifies as $\pi_0$ so long as its risk is at most $\alpha$. When little is known, $\alpha$ can be set loosely and any reasonable baseline suffices; the strength of the resulting guarantee scales with how much is actually known about what is safe.

**Assumption 2: Access to Each Optimized Policy** At each time $t = 1, 2, ...$, also assume access to the optimized model, $\pi_t$, which may be trained to maximize expected reward (as in the "unconstrained" part of Eq. (1)). We do not require any assumptions on how $\pi_t$ is trained, although better optimized performance (with respect to both the constraint and the reward) may translate into better efficiency and reward of the constrained policy.

**Constraining Policies by Clipping Likelihood Ratios** We define the constrained policy $\pi_t^{(\beta)}$ by clipping the likelihood ratio $\pi_t/\pi_0$ at a bound $\beta > 0$ and renormalizing. Actions whose likelihood ratio exceeds $\beta$ are capped at $\beta \cdot \pi_0$, limiting how far the constrained policy can deviate from the safe policy. That is,

$$\pi_t^{(\beta)}(a \mid x) \propto \min\left(\pi_t(a \mid x), \beta \cdot \pi_0(a \mid x)\right). \quad (5)$$

Assume there is some small, positive $\beta_{\min} \ll 1$ such that $\beta_{\min} \leq \pi_t(a \mid x)/\pi_0(a \mid x)$ for all $x \in \mathcal{X}$, $a \in \mathcal{A}$. As $\beta \to \beta_{\min}$ the constrained policy approaches the safe one, $\pi_t^{(\beta)} \to \pi_0$, and as $\beta \to \infty$ it approaches the optimized one, $\pi_t^{(\beta)} \to \pi_t$ (Figure 3). Crucially, because $\beta$ governs the likelihood ratio and not the loss, any smoothness needed for a guarantee will be smoothness the agent can compute and verify. Theorem 4.5 makes this precise.

**Calibrating $\beta$:** We calibrate $\beta$ analogously to $\hat{\lambda}_+$ (Section 4.1), searching from what we *a priori* expect is safest to most aggressive. For $\beta$, this means searching in increasing order in a grid (or set of intervals) $G \subset (0, \infty]$, to find the largest value where the conservatively-adjusted weighted empirical risk is still controlled (Algorithm 1):

$$\hat{\beta}\left(\lfloor L_{0:t-1}, B \rfloor, (\tilde{w}_{0:t-1}^{(\cdot)}, \tilde{w}_{\max}^{(\cdot)}), \alpha\right) \quad (6)$$

$$:= \sup\left\{\beta' \in G \ : \ \forall \beta \leq \beta', \ B\tilde{w}_{\max}^{(\beta)} + \sum_{i=0}^{t-1} l_i \tilde{w}_i^{(\beta)} \leq \alpha\right\},$$

where the $\tilde{w}_i^{(\beta)}$ and $\tilde{w}_{\max}^{(\beta)}$ are (normalized) conformal importance weights. These weights correct for the distribution

shift between the policies that generated the calibration data and the candidate policy $\pi_t^{(\beta)}$: since the agent's past actions were drawn from earlier policies, each observation must be reweighted to obtain a valid risk estimate under $\pi_t^{(\beta)}$. This is the importance weighting described in Section 2, now made precise for the multi-round setting.

Formally, we define $\tilde{w}_i^{(\beta)}$ as in Prinster et al. (2024) via the likelihoods of all permutations of the calibration data $z_{0:t-1}$ and a hypothetical test sample $z_t \in \mathcal{Z} := \mathcal{X} \times \mathcal{A}$. A key subtlety is that $z_t$ is not yet known, as CPC calibration occurs prior to sampling. For any $z_t$, define the unnormalized weight

$$w_i^{(\beta)} := \sum_{\sigma:\sigma(t)=i} \pi_{0:t}^{(\beta)}(z_{\sigma(0)}, ..., z_{\sigma(t)})$$

for all $i \in \{0, ..., t\}$ and for permutations $\sigma$ of the indices. For $i \in \{0, ..., t\}$, further denote the normalized weights as

$$\tilde{w}_i^{(\beta)} := w_i^{(\beta)} / \sum_{i=0}^{t} w_i^{(\beta)}, \quad \tilde{w}_{\max}^{(\beta)} := \sup_{z_t \in \mathcal{Z}} (\tilde{w}_t^{(\beta)}),$$

where $\tilde{w}_{\max}^{(\beta)}$ is a conservative estimate of the test point weight. Hereon, we refer to normalized calibration and (conservative) test point weights, that is, assuming $\sum_{i=0}^{t-1} \tilde{w}_i^{(\beta)} + \tilde{w}_{\max}^{(\beta)} = 1$. In practice these weights are not computationally tractable, but we estimate the $\tilde{w}_i^{(\beta)}$ by approximating past policies as one mixture distribution, $\pi_{0:t-1}^{\text{mix}} \approx \pi_{0:t-1}$, which yields a fast linear runtime; additionally, we estimate $\tilde{w}_{\max}^{(\beta)}$ by an empirical maximum over samples from $\pi_0$ and $\pi_t$. If the weights are computed exactly, CPC achieves the following guarantee.

**Theorem 4.5.** *Assume that* $\mathbb{E}_{\pi_0}[L_t] \leq \alpha$, $L_i \in [0, B]$, $B < \infty$ *for all* $i$, *and* $\hat{\beta}$ *is* $\epsilon$-*replace-one stable. If* $w_i^{(\beta)}$ *is* $K$-*Lipschitz in* $\beta$ *for all* $i$, *then*

$$\mathbb{E}_{\pi_{0:t}^{(\hat{\beta})}}[L_t] \leq \alpha + BK\epsilon.$$

*If* $(l_i - \alpha) \cdot w_i^{(\beta)}$ *is* $K'$-*Lipschitz in* $\beta$, *then*

$$\mathbb{E}_{\pi_{0:t}^{(\hat{\beta})}}[L_t] \leq \alpha + K'\epsilon.$$

A key advantage of CPC over gCRC is where the Lipschitz condition falls. In Theorem 4.2, smoothness is required of $l_i(\lambda)$, the loss as a function of the control parameter, which is a property of the unknown environment. In Theorem 4.5, smoothness is required of $w_i^{(\beta)}$ or $(l_i - \alpha) \cdot w_i^{(\beta)}$ as a function of $\beta$. Since $l_i$ does not depend on $\beta$, this regularity is determined entirely by the conformal weights, which depend on the likelihood ratio between the safe and optimized policies. The assumption thus concerns the agent's own

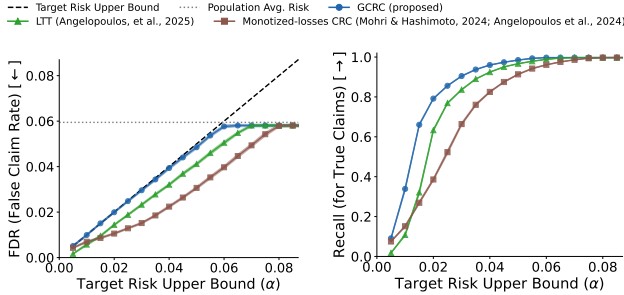

*Figure 4.* Medical QA factuality control results on MedLFQA. **Left:** Empirical FDR (fraction of retained claims that are false) vs. target risk level $\alpha$. The dashed line indicates $y = x$; valid methods should fall at or below this line. **Right:** Recall (fraction of true claims retained) vs. $\alpha$. gCRC tightly controls FDR at the target level while achieving superior recall to baselines. Results averaged over 25 random splits. Error regions are standard errors.

policies rather than the unknown constraint function, and is in principle verifiable or enforceable by the practitioner.

**Sampling from the Constrained Policy in Combinatorial Action Spaces** In large action spaces, the normalization constant of $\pi_t^{(\hat{\beta})}$ is intractable. However, accept-reject (AR) sampling (Von Neumann et al., 1963) avoids normalization entirely. When $\hat{\beta}$ is small, proposals from $\pi_0$ are efficient, with the AR envelope constant set to $\hat{\beta}$. When $\hat{\beta}$ is large, proposals from $\pi_t$ are more efficient, since $\pi_t^{(\hat{\beta})} \approx \pi_t$. See Figure 3 for an illustration and Appendix C.2 for details.

## 5. Experiments

We present three experiments here. First, we apply generalized CRC (gCRC) to control the false discovery rate in medical question answering, a non-monotonic loss. Second, we apply CPC to constrained active learning, where feedback-loop shifts break exchangeability. Third, we apply CPC to black-box sequence optimization with a language model. Additionally, Appendix G presents a fourth experimental setting that compares CPC to standard conservative optimization on a constrained Bayesian optimization task.

### 5.1. Factual Medical Question-Answering

**Setup:** We evaluate CRC methods on the MedLFQA dataset (Jeong et al., 2024), which aggregates medical question-answering benchmarks. We use the dataset of Cherian et al. (2024), which contains GPT-3.5-Turbo responses, atomic subclaims extracted via GPT-4o, and factuality annotations obtained by checking each subclaim against reference answers. Given a confidence score $p(C_{ij} \mid X_i)$ for each subclaim, we filter claims below a threshold $\tau$ and measure the false discovery rate (FDR), the fraction of retained claims that are false, as our loss. We measure recall, the fraction of total true claims retained, as our utility metric. Unlike

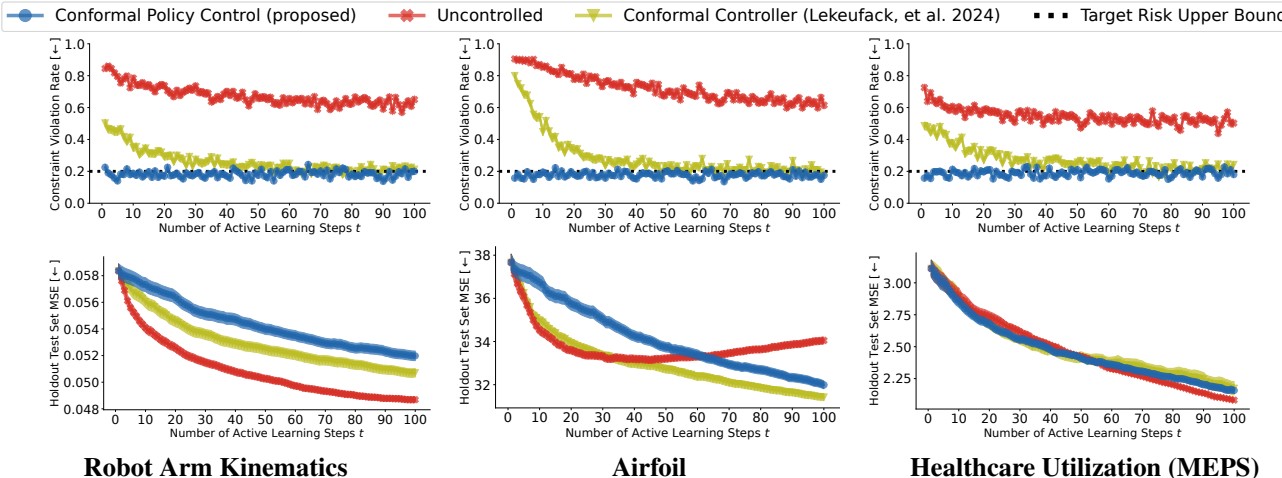

*Figure 5.* Conformal policy control in the active learning setting. We train Gaussian process regression models on tabular datasets, providing a small amount of initial data for training and selecting the remaining training data via exponential tilting toward the posterior predictive variance. We introduce a feasibility constraint based on alignment with the leading principal component of the Gram matrix to make the task more difficult. CPC controls the constraint violation risk at our desired threshold $\alpha = 0.2$, while simultaneously reducing test MSE. Surprisingly, in some cases the risk-controlled data selection policy attains lower test MSE than the uncontrolled policy.

the binary loss used by Mohri & Hashimoto (2024) and Cherian et al. (2024), FDR is non-monotonic in $\tau$, making it a natural test case for gCRC. We compare gCRC against CRC applied to monotonized losses and Learn Then Test (LTT) (Angelopoulos et al., 2025). See Appendix D for details.

**Results:** Figure 4 shows FDR and recall across varying risk levels $\alpha$. gCRC controls FDR at or below the target level across all $\alpha$ values while achieving recall at least as high, or often significantly higher, than both the monotonized-losses CRC and LTT baselines. Relative to gCRC, LTT is conservative because it uses concentration inequalities to obtain a high-probability guarantee. Monotonized-loss CRC is conservative as a consequence of Jensen's inequality: monotonizing each loss is far more conservative than monotonizing the empirical risk. These results demonstrate that gCRC provides valid risk control for non-monotonic losses where standard CRC may not be valid, and empirically with higher power than baselines.

### 5.2. Constrained Active Learning

**Setup:** We evaluate CPC in a pool-based active learning setting on three regression datasets: **Robot Arm Kinematics**, **Airfoil**, and **Healthcare Utilization (MEPS)**. Starting from a small initial training set, the learner iteratively selects records from the pool to minimize test MSE. To create a challenging constrained setting, we construct synthetic feasibility constraints based on the first principal component of the covariate matrix. Records that follow the dominant covariation pattern (high PC1 values) are likely feasible, while records that deviate from this pattern are likely infea-

sible. Initial training data is sampled from the high-PC1 region, so the learner begins in a high-feasibility region. Uncertainty-based acquisition naturally targets the opposite end of the spectrum. We fit a Gaussian process regression model and define an acquisition policy via exponential tilting toward posterior variance: $\pi(a) \propto \exp(\lambda \cdot \hat{\sigma}_a^2)$. CPC constrains this policy relative to the source distribution used for initialization. See Appendix E for details.

**Results:** In Figure 5 we present results across all three datasets. CPC successfully controls constraint violation risk at the target $\alpha$ while reducing test MSE. Surprisingly, in some cases the risk-controlled policy attains *lower* test MSE than the unconstrained policy, suggesting that avoiding infeasible regions improves sample efficiency.

### 5.3. Constrained Black-Box Optimization

**Setup:** Now consider a more difficult setting, black-box sequence optimization. Here, our goal is to identify a sequence of tokens $a \in \mathcal{X}$ to maximally improve the objective value of a current solution $x \in \mathcal{X}$. We evaluate on Ehrlich functions (Chen et al., 2025), synthetic test functions designed to capture the geometric structure of biomolecular sequence optimization problems. We use **Ehr(32, 32)-4-4**: sequences of length 32 over a vocabulary of size 32, with 4 motifs of length 4. We create an initial data package by running a genetic algorithm from a single feasible solution. We train $\pi_{\theta_0}$ by applying SFT to a Pythia 14M-parameter decoder-only transformer (Biderman et al., 2023) on improving pairs from the GA history. In later rounds, we sample from the current policy, collect oracle feedback, and fine-tune via DPO. CPC constrains each improved policy

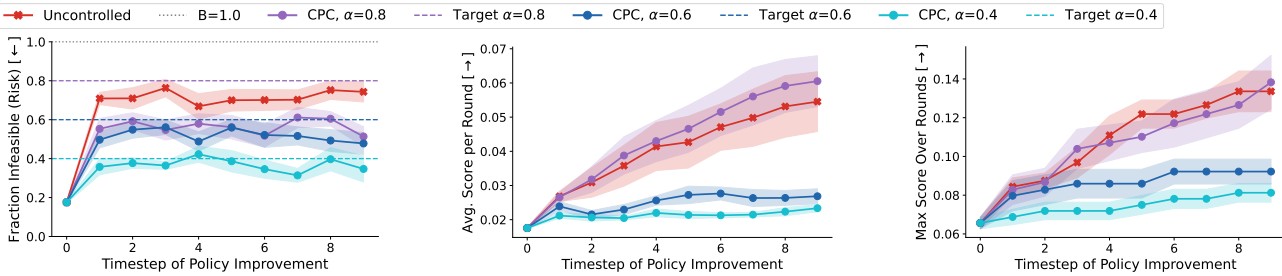

*Figure 6.* We apply CPC to constrained black-box sequence optimization scored with Ehrlich functions. **Left:** average infeasibility rate per round. **Center:** average objective value of sequences sampled from the policy. **Right:** best objective value found so far. CPC allows direct control over constraint violation risk via $\alpha$. Without CPC, the infeasibility risk quickly rises to nearly 80% of policy samples. Moderate risk control ($\alpha > 0.6$) stabilizes optimization and improves overall performance by reducing wasted samples on infeasible actions.

$\pi_{\theta_t}$ before sampling. See Appendix F for details.

**Results:** In Figure 6 we show that CPC effectively controls the risk of generating infeasible sequences while still improving the objective value over time. Again we find that moderate risk control ($\alpha > 0.6$) can produce controlled policies that slightly *outperform* their uncontrolled counterparts, likely by wasting fewer evaluations on infeasible actions.

## 6. Discussion

We set out to let an agent choose how far to deviate from a safe policy without the act of choosing biasing the estimate that justified the choice. The resolution exploits the fact that the only shift between the safe policy's data and any candidate's deployment is the agent's own: it does not know how the world will respond, but it knows exactly how each candidate would distribute its actions. Conformal policy control parameterizes that choice as a likelihood-ratio threshold and calibrates it directly on the safe policy's data, so that selection is the calibrated act rather than a step that follows it. Because the threshold bounds how far the candidate can deviate from the safe policy, the resulting guarantee applies to the chosen policy itself, not merely each candidate in isolation.

More broadly, CPC instantiates a prescriptive use of conformal methods: rather than characterizing uncertainty under a fixed distribution, it searches over distributions to find one that satisfies a finite-sample guarantee. This inversion applies wherever one selects a data-generating process subject to a risk constraint, from experimental design, to active data collection, to generative model deployment. We hope that the ideas in this paper encourage further development of conformal methods as tools for decision-making, beyond their traditional role in uncertainty quantification.

The guarantees we provide are marginal: risk is controlled on average over context and calibration data draws. This is the natural guarantee for deployment decisions, where one asks whether a policy is safe to deploy across a popula-

tion. It is less informative for individual decisions, where one asks whether a particular action is safe for a particular context. Stronger conditional guarantees are possible but require additional assumptions or more conservative bounds (Thomas et al., 2019). Our guarantees also depend on the stability of the context distribution. If the context distribution begins to shift then recalibration may be necessary, or some risk control gap may need to be tolerated (Barber et al., 2023). Monitoring methods based on conformal martingales offer a principled approach to detecting such shifts (Prinster et al., 2025). We have also assumed the policy likelihoods are computable in closed form. If the policy likelihoods are intractable, then density ratio estimation (Sugiyama et al., 2012) or neural ratio estimation (Cranmer et al., 2020) techniques may be applicable.

A recurring theme in this work is that assumptions should reflect what the practitioner knows rather than what is theoretically convenient. CPC requires a safe policy, but any baseline whose risk the practitioner is willing to accept will do. It requires smoothness, but of the agent's own policy likelihood ratios, not of the unknown constraint loss function. It requires a risk tolerance, but as a direct declaration rather than an indirect penalty weight. Safety and exploration, in this view, need not run counter to each other. In the right balance they correct each other's deficiencies, avoiding complacency on the one hand and dissipation on the other. The balance must be set from what is actually known, not what is hoped. It can be enough, it turns out, to know what works, and to know what we want to do next.

## Code Availability

Code to reproduce the results is available at https://github.com/samuelstanton/conformal-policy-control.

## Impact Statement

The dominant paradigm for deploying machine learning systems might be characterized as "train, deploy, and pray". Collect data, train a model, tune hyperparameters until empirical performance seems acceptable, deploy, and hope that test-time behavior remains within tolerable bounds. When failures occur, the response is typically reactive, patching the system after harm has been observed. This paradigm persists not because practitioners prefer it, but because principled alternatives have been lacking.

This work contributes to a shift from safety-through-patching toward safety-by-design. Rather than discovering failure modes empirically and addressing them post-hoc, conformal policy control allows practitioners to specify acceptable risk levels before deployment and obtain guarantees that these levels will be respected. The hyperparameter tuning phase becomes about improving performance within safety constraints, rather than searching for a configuration that might satisfy unspecified feasibility requirements.

If such methods become widespread, the implications for where and how machine learning is deployed could be substantial. High-stakes domains that have resisted ML adoption due to liability concerns or regulatory barriers, such as clinical decision support, autonomous systems, and financial services, may become more accessible when developers can provide formal guarantees rather than empirical assurances. Such a change in framing would align ML deployment with the safety certification practices already standard in aviation, pharmaceuticals, and other regulated industries.

## Acknowledgments

D.P., A.L., and S. Saria were partially funded by the Gordon and Betty Moore Foundation grant #12128. The authors would like to thank Hannah Lawrence and Neha Verma for helpful feedback.

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

# Supplementary Material

## Table of Contents

## A. Extended Related Work

### A.1. Conservative Model-Based Optimization

The tension between optimization performance and reliability has deep roots in decision theory, from early work on optimal decision-making under uncertainty (Raiffa & Schlaifer, 1961) to stochastic methods for optimization under noise (Kushner, 1964). The *optimizer's curse* (Smith & Winkler, 2006) captures the key challenge: when decisions are selected by maximizing an estimated value function, the selected action is systematically biased toward outcomes whose value is overestimated. Crucially, this bias persists even when the value estimates are unbiased on average and arises from the act of selection itself. Among many uncertain alternatives, those with the largest positive estimation errors are most likely to be chosen. In the context of sequential decision making and policy optimization, this effect manifests when an optimized policy is trained to maximize expected reward under limited or noisy feedback, leading it to concentrate probability mass on actions whose apparent advantage is driven by estimation error rather than true performance. As a result, policies that appear optimal in expectation may perform poorly when deployed, particularly when they extrapolate beyond regions where reliable evidence is available. In sequential settings, this effect compounds. Aggressive optimization shifts the data distribution away from regions where the estimators are reliable, further degrading the estimates and creating more error for the optimizer to exploit.

Various communities have aimed at guiding models toward high-performing solutions while explicitly constraining distribution shift away from trusted data or policies. In stochastic control and reinforcement learning, entropy-regularized control and KL-penalized objectives constrain optimized policies to stay close to a reference distribution (Todorov, 2009; Fox et al., 2016). Related formulations arise in path integral control and KL-regularized path measures (Kappen, 2005; Theodorou et al., 2010). Trust-region policy optimization methods can be viewed as a practical instantiation that explicitly bound policy updates with local constraints (Schulman et al., 2015; 2017). In offline and batch reinforcement learning, conservative objectives penalize actions whose value estimates rely on out-of-distribution extrapolation (Kumar et al., 2020; Trabucco et al., 2021). More recently, the control-theoretic perspective has been extended to generative model fine-tuning. Fine-tuning diffusion and flow models to maximize the expected reward while retaining the pretrained model information via KL regularization has been framed as an entropy-regularized optimal control problem (Uehara et al., 2024; Domingo-Enrich et al., 2025), and generalized to other utilities and divergence measures (De Santi et al., 2025).

Finally, in black-box optimization, trust-region Bayesian optimization restricts search to local regions where surrogate models are reliable (Eriksson et al., 2019), while safe Bayesian optimization enforces hard safety constraints to prevent evaluation in unsafe regions (Sui et al., 2015; Berkenkamp et al., 2016; Kirschner et al., 2019; Berkenkamp et al., 2023). All of these methods operationalize safety by bounding how far the optimized distribution may deviate from a known baseline.

In practice, however, the strength of this constraint is governed by a scalar hyperparameter that has no intrinsic semantic interpretation. Whether expressed as a KL penalty weight, a likelihood ratio threshold, or a trust-region radius, this parameter controls the degree of aggressiveness in optimization but does not directly correspond to any declarative risk criterion, such as a target failure rate. As a result, practical deployment requires solving a secondary *hyperparameter selection problem*: one must select many candidate values of the constraint parameter and empirically evaluate their downstream risk in order to

*Table 1.* Summary comparison of selected related work on conformal prediction for decision making under agent-induced shifts.

| Method Group References | 1. Agent-Induced Data Shift | 2. Action Space | 3. Type of Risk Control | | |
| --- | --- | --- | --- | --- | --- |
| | | | Prediction Set Miscoverage $\mathbb{1}\{Y_t \notin \hat{C}_\alpha(X_t)\}$ | $\lambda$-Parameterized Losses $L_t(A_t(\lambda))$ | Unparameterizable Losses ($\beta$-Param. Policy) $L_t(A_t),\ A_t \sim p_t(\beta)$ |
| **Conformal Prediction** Vovk et al. (2005) Vovk & Bendtsen (2018) | ✗ | Small/ Finite | ✓ | ✗ | ✗ |
| **Weighted Conformal Pred.** Tibshirani et al. (2019) Fannjiang et al. (2022) Stanton et al. (2023) Nair & Janson (2023) Prinster et al. (2024) | Single-Round ⎯⎯⎯ **Multi-Round** | Small/ Finite ⎯⎯⎯ Contin. ⎯⎯⎯ Small/ Finite | ✓ | ✗ | ✗ |
| **Conformal Risk Control** Angelopoulos et al. (2024) Lekeufack et al. (2024) | Single-Round Eventually safe | Small/ Finite | ✓ | ✓ | ✗ |
| **Conformal Policy Control** (Proposed) | **Multi-Round** | Large/ $\infty$ | ✓ | ✓ | ✓ |

identify a satisfactory operating point. Critically, this calibration must be performed *on-policy*, since the risk of interest depends on the distribution induced by each specific choice of the hyperparameter. This requirement is costly and introduces further feedback-loop dependencies (Bergstra et al., 2011; Feurer & Hutter, 2019).

Yet the distribution shift these methods seek to control is self-inflicted. The agent chooses its own policy, and therefore knows the likelihood ratio between any candidate policy and the safe baseline in closed form. The distribution shift is not an external unknown but a consequence of the agent's own design. This means the hyperparameter selection problem can be solved with off-policy data from the safe policy alone, provided the calibration is done carefully. Conformal techniques provide exactly this kind of careful calibration.

### A.2. Conformal Methods

**Conformal prediction under distribution shift in high-dimensional spaces**

Conformal prediction, originally developed for exchangeable or i.i.d. data by (Vovk et al., 2005), has been extended to a variety of distribution shift settings. Closest to the current paper are *weighted* conformal methods that *proactively adapt* to distribution shift by re-weighting calibration data using knowledge of the change, though another line of work develops conformal-inspired methods for *retroactively reacting* to adversarial shifts over time (e.g., Gibbs & Candes (2021); Zaffran et al. (2022); Bastani et al. (2022); Angelopoulos et al. (2023)). Among weighted conformal methods, those most relevant to the present work are methods tailored for covariate shift (Tibshirani et al., 2019; Prinster et al., 2022; Yang et al., 2024; Jonkers et al., 2024), and especially its extensions in which the training and test data are dependent, such as one-step (Fannjiang et al., 2022; Prinster et al., 2023) and multi-step (Nair & Janson, 2023; Prinster et al., 2024) feedback covariate shift, as is the case in the policy optimization setting. Also relevant are the fixed-weight methods of Barber et al. (2023), which bound the coverage violation due to misspecified data-independent weights, and the unifying weighted conformal framework of Barber & Tibshirani (2025).

When deviating from the assumption of exchangeable calibration and test data, any analysis of proactive conformal prediction methods invokes quantities characterizing the relationship between the calibration and test distributions, such as their density ratios or divergences. Therefore, despite the advances in theory, deployment of conformal prediction methods in practice in distribution shift settings has been hamstrung by the practical difficulty of evaluating such quantities, particularly in high-dimensional spaces. Consider the combinatorially large action spaces involved in high-impact applications such as biological sequence design. Unless modeling the distribution using factorizations amenable to tractable computation (e.g., autoregressive generative models), it is generally computationally intractable to evaluate the density of distributions over sequence space, as exact computation of normalizing constants is infeasible. As a workaround, some approaches have resorted to density ratio estimation (Stanton et al., 2023; Laghuvarapu et al., 2023; Fannjiang & Park, 2025; Correia & Louizos, 2025), a task that is also challenging and ill-posed in high dimensions (Rhodes et al., 2020), and which nullifies

validity guarantees due to estimation error. In the present work, we circumvent the difficulty of evaluating density ratios between the calibration and test policies by making their density ratio itself the tunable control parameter, as follows. We first assume existence of an initial safe policy, $\pi_0$, that satisfies the safety constraint, as well as naively optimized policies at each time, $\pi_t$. At each time, we define a family of "constrained policies" that interpolates between $\pi_0$ and $\pi_t$ by clipping their density ratio, $\pi_t(\cdot)/\pi_0(\cdot)$, at an upper bound, $\beta_t$ (Eq. (5)). By making this density ratio upper bound, $\beta_t$, the tunable control parameter, we have done away with the need to estimate the density ratio altogether.

**Conformal methods for decision-making in the offline setting**

Conformal methods, by which we mean conformal prediction techniques and their generalizations to settings beyond constructing instance-wise prediction sets with coverage guarantees, have also been developed for various decision-making settings. Vovk & Bendtsen (2018) laid the groundwork for conformal decision-making, demonstrating how conformal prediction (specifically conformal predictive distributions (Vovk et al., 2017)) can be used to make decisions that satisfy a notion of efficiency. They noted, however, that they had "limited [themselves] to analyzing a given batch of data, without attempting active learning, and this limitation has made it possible to develop a fairly systematic theory."

Indeed, following Vovk & Bendtsen (2018), subsequent work on conformal methods for decision making has largely also assumed an offline (i.e., non-active) setting, where training and calibration are entirely independent of test-time decisions: that is, where pre-deployment training and calibration cannot influence distributions at test time, and vice versa (without breaking statistical guarantees). For instance, see Hullman et al. (2025) for a recent overview. Approaches in this offline setting have included conformal prediction sets for individual treatment effect estimates (Lei & Candès, 2021), off-policy prediction (Taufiq et al., 2022; Zhang et al., 2023), extending conformal prediction sets to handle notions of risk beyond miscoverage (Bates et al., 2021; Angelopoulos et al., 2024), risk assessment (Prinster et al., 2022; Singh et al., 2023; Zhou et al., 2025), developing connections with Bayesian decision theory (Hoff, 2023; Snell & Griffiths, 2025), and designing conformal sets for (or the filtering of) outputs of fixed/pre-trained generative models (Quach et al., 2024; Teneggi et al., 2023; Mohri & Hashimoto, 2024; Cherian et al., 2024; Overman et al., 2024; Jiang et al., 2025). Another line of work has developed conformal-inspired optimization procedures to simulate decision making during training (Colombo & Vovk, 2020; Stutz et al., 2022; Cortes-Gomez et al., 2025; Yeh et al., 2024; 2025), but notably, these methods all require a further split of the data beyond that of split conformal (Papadopoulos et al., 2002), thus effectively side-stepping the "optimizer's curse" at the cost of reduced data efficiency.[6]

While Kiyani et al. (2025) theoretically demonstrated that prediction sets are particularly suitable for risk-averse decision-making and several works have empirically demonstrated the utility of prediction sets for human decision makers (Straitouri et al., 2023; Cresswell et al., 2024; Zhang et al., 2024a), there nonetheless remains a gap between methodological progress and practical impact of conformal methods (Hullman et al., 2025). Arguably, this persistent gap is due to the fact that predictions (and consequently, prediction sets) are almost never the end goal in practice. Instead, the end goal in practice is often to control how prediction error affects the specific downstream decisions, or the policy, of interest.

**Conformal methods for decision-making in the active setting**

In the active setting, where an agent's actions may influence the observed data distribution, there are two main threads of literature that parallel corresponding methods for conformal prediction under distribution shift. The first thread uses ideas from weighted conformal to proactively adapt prediction sets to agent-induced shifts: Fannjiang et al. (2022) extended the weighted conformal methods of Tibshirani et al. (2019) to a one-step instance of "feedback covariate shift," where the input distribution of a single test batch can depend on the training data. Inspired by this work, Stanton et al. (2023) incorporated weighted conformal methods into Bayesian optimization, aiming to close the decision-making loop by allowing conformal calibration and optimization to bidirectionally influence each other; however, they noted gaps in the theory that prevented the desired guarantees from being realized. Prinster et al. (2023) extended Fannjiang et al. (2022)'s methods to cross-validation-style conformal calibration; Nair & Janson (2023) introduced an extension to multiple rounds of dependent covariate shifts; and Prinster et al. (2024) extended weighted conformal methods to any data distribution, with multiround agent-induced covariate shifts as a main focus. However, these prior works differ from the present paper in several ways: they all operate *descriptively* to adapt prediction sets for a fixed policy, whereas our methods function *prescriptively*, using conformal calibration to *find* a risk-controlling policy; in combinatorial action spaces, they are bottlenecked by challenges of

---

[6]That is, these "conformal training" works typically require at least three disjoint splits of the data to maintain statistical validity: a "proper" training set for learning, a validation set used for hyperparameter tuning (or simulating conformal calibration), and a separate fresh calibration set at test time for inference. In contrast, the present paper's conformal policy control methods avoid such an additional data split by carefully reweighting the calibration data to adjust for data-dependent policy selection.

density-ratio estimation and normalization; and they focus on (mis)coverage rather than more general notions of risk.

The second line of conformal methods in the active setting uses ideas from online adversarial conformal (e.g., Gibbs & Candes (2021)) to retroactively learn decision rules over time: Feldman et al. (2023) extended the online algorithm of Gibbs & Candes (2021) to more general loss functions (similarly as Bates et al. (2021); Angelopoulos et al. (2024) did in the offline setting), and Lekeufack et al. (2024) extended those ideas to a conformal decision theory framework, where the online learning update directly steers an agent's decisions at the next timestep. Zhang et al. (2024b) use ideas from online adversarial conformal for Bayesian optimization and Zhao et al. (2024) for imitation learning. These works achieve bounds on the long-term average risk as the number of time steps grows indefinitely. In robotics, conformal prediction has also been used to construct predictive sets of trajectories, which are used to make planning or control procedures safer (Lindemann et al., 2023; Luo et al., 2024; Sun et al., 2023; Wang & Ning, 2025; Lindemann et al., 2024).

**Conformal selection and high-probability risk control**

Along a different vein, motivated by candidate selection problems that arise in drug discovery, conformal selection methods (Jin & Candès, 2022; 2025; Bai & Jin, 2024; Gui et al., 2025) merge ideas from conformal prediction and multiple testing to select designs from a pool of individual candidates, such that the false discovery rate, or expected proportion of selected designs whose label does not surpass a desired threshold, is controlled. While these methods assume access to a pool of exchangeable (e.g., i.i.d.) candidates, in some applications—for example, biological sequence design—this is not a natural setting. Instead, it is more relevant to reason about generating candidates from some distribution, which brings us to our problem of how to select a generative model that controls risk as desired. Fannjiang & Park (2025) tackle this problem as a multiple testing problem to arrive at high-probability guarantees of risk control. Such high-probability risk control guarantees are akin in spirit to prior works on risk controlling prediction sets (Bates et al., 2021) and the Learn-then-Test framework (Angelopoulos et al., 2025), which provide high-probability guarantees for monotonic and non-monotonic losses respectively. In contrast, we build on techniques from conformal risk control (Angelopoulos et al., 2024) to achieve guarantees in expectation, which typically provides tighter risk control. Relative to conformal risk control, we also introduce and focus on the policy control setting, wherein the control parameter directly controls the agent's policy, to fully connect the decision-making loop, circumvent challenges of density-ratio estimation, and address the optimizer's curse head-on.

### A.3. Seldonian Algorithms

Seldonian algorithms (Thomas et al., 2019) provide a framework for constructing machine learning systems that satisfy user-specified behavioral constraints with high probability. The approach builds on earlier work on high-confidence policy improvement (Thomas et al., 2015), which introduced the use of concentration inequalities to obtain probabilistic guarantees on policy performance. Rather than requiring users to tune abstract regularization weights or penalty coefficients, Seldonian algorithms allow direct specification of constraints such as "the probability of causing harm should be at most 5%." The framework uses concentration inequalities to derive high-probability bounds on constraint satisfaction, returning a solution only when sufficient evidence exists that the constraints will be met.

The Seldonian framework has been extended in several directions. Metevier et al. (2019) adapted the approach to contextual bandits with fairness constraints, demonstrating how user-specified fairness definitions can be enforced with high-probability guarantees. Chandak et al. (2020) addressed non-stationary MDPs by incorporating time-series forecasting to anticipate distribution shift. Giguere et al. (2022) extended the framework to handle demographic shift between training and deployment, providing guarantees that remain valid when the population composition changes.

Both Seldonian algorithms and conformal policy control share the goal of translating user-specified risk tolerances directly into algorithmic constraints, eliminating the need for trial-and-error hyperparameter tuning. However, the two frameworks differ in the nature of their guarantees and the distinction between *certification* and *regulation*. Seldonian algorithms provide *conditional, high-probability* bounds: with probability at least $1 - \delta$, the deployed policy satisfies the specified constraints. In contrast, conformal methods provide *marginal, expected value* guarantees: on average, the expected loss under the deployed policy is bounded by the user-specified level $\alpha$. Neither guarantee type strictly dominates the other. High-probability bounds are more conservative but offer protection against worst-case outcomes. Expected value bounds are tighter on average but permit occasional constraint violations. The greatest difference between our work and Seldonian algorithms is our method can be used to safely deploy (control the expected risk of) a model that would not be safe otherwise through the rejection sampling regulation procedure. By contrast, a Seldonian algorithm can certify that the solution it returns satisfies its constraints with high probability, however it cannot regulate an existing solution trained by some third-party algorithm.

## B. Proofs

### B.1. Proof generalized conformal risk control with exchangeable data.

**Notation:**

- **Data (sequences of random variables and observed values):** For all $i \in \mathbb{N}$, denote the random variable for the $i$-th data sample as $Z_i := (X_i, Y_i) \in \mathcal{X} \times \mathcal{Y} = \mathcal{Z}$, and denote observed value of that random variable in lower case as $z_i := (x_i, y_i) \in \mathcal{X} \times \mathcal{Y} = \mathcal{Z}$. Similarly, denote a sequence of random variables as $Z_{1:m} := (Z_1, ..., Z_m)$ and a sequence of observed data values as $z_{1:m} := (z_1, ..., z_m)$ for any $m \in \mathbb{N}$. To concisely denote indices, let $[m] := \{1, ..., m\}$.

- **Hyperparameter space:** Let $\Lambda \subseteq \mathbb{R}$ be the space of hyperparameters. So, $\lambda \in \Lambda$ denotes a particular hyperparameter in this space.

- **Loss functions:** For some real-valued function $g : \mathcal{Z} \times \Lambda \to \mathbb{R}$, and for all $\lambda \in \Lambda$ and all $i \in \mathbb{N}$, let $L_i(\lambda) := g(Z_i, \lambda)$ denote the loss function determined by a random variable $Z_i$ and parameter $\lambda \in \Lambda$, and let $\ell_i(\lambda) := g(z_i, \lambda)$ denote the loss function determined by the observed value $z_i$ and parameter $\lambda \in \Lambda$. To emphasize when no particular $\lambda$ has yet been given as an argument, we will use $L_i(\cdot) := g(Z_i, \cdot)$ to denote the loss function with randomness over $Z_i$ not yet determined, and we will use $\ell_i(\cdot) := g(z_i, \cdot)$ to denote the loss function for observed value $z_i$.

- **Bags:** For any sequence $v_{1:m} := (v_1, ..., v_m)$, let $\wr v_{1:m} \wr$ denote the *bag* (or multiset) containing the values of that sequence $v_{1:m}$ (but importantly, $\wr v_{1:m} \wr$ does *not* contain the information about the order of $v_{1:m}$). For example, $\wr (1, 7, 42, 7) \wr$ conveys that there is one 1, two 7s, and one 42 in the bag, but it does *not* convey the order that these appeared in the sequence $(1, 7, 42, 7)$. In particular, with this notation, for the random variables $Z_{1:m}$, let $\wr Z_{1:m} \wr$ denote the bag containing those random variables; for observed values $z_{1:m}$, let $\wr z_{1:m} \wr$ denote the bag containing those observed values; and thus $\wr Z_{1:m} \wr = \wr z_{1:m} \wr$ denotes the event of observing $\wr z_{1:m} \wr$, in other words, that each $Z_i$ takes a different value in $\wr z_{1:m} \wr$, but it is not yet known *which* value (for all $i \in [m]$).

- **Distributions:** Let $F_{Z_{1:m}} := F_{Z_1, ..., Z_m}$ denote the joint distribution function for $Z_1, ..., Z_m$, and let $F_{\wr Z_{1:m} \wr} := F_{\wr Z_1, ..., Z_m \wr}$ denote the (pushforward) distribution induced by the mapping $(z_1, ..., z_m) \to \wr z_{1:m} \wr$.

**Theorem B.1** (Restatement of Theorem 4.2). *Assume exchangeable $L_i(\lambda)$. Define $\lambda_{\max} := \sup \Lambda \in \Lambda$ and assume*

$$L_i(\lambda_{\max}) \leq \alpha, \qquad \sup_{\lambda} L_i(\lambda) \leq B < \infty \quad \text{almost surely.}$$

*If the $L_i(\lambda)$ are $K$-Lipschitz in $\lambda$ and $\hat{\lambda}_+$ is $\epsilon$-replace-one stable, then*

$$\mathbb{E}\big[L_{n+1}(\hat{\lambda}_+)\big] \leq \alpha + K\epsilon.$$

*Proof.*

**Step 0: Law of total expectation over draw of bag.**

Expanding $\mathbb{E}[L_{n+1}(\hat{\lambda}_+)]$ via the law of total expectation over the event $\wr Z_{1:n+1} \wr = \wr z_{1:n+1} \wr$, we have

$$\mathbb{E}\big[L_{n+1}(\hat{\lambda}_+)\big] = \mathbb{E}_{F_{\wr Z_{1:n+1} \wr}} \Big[\mathbb{E}\big[L_{n+1}(\hat{\lambda}_+) \mid \wr Z_{1:n+1} \wr = \wr z_{1:n+1} \wr\big]\Big]. \tag{7}$$

We can thus see that, to prove the desired result, it is sufficient to upper bound $\mathbb{E}\big[L_{n+1}(\hat{\lambda}_+) \mid \wr Z_{1:n+1} \wr = \wr z_{1:n+1} \wr\big]$ almost surely given any draw of the bag, $\wr Z_{1:n+1} \wr \sim F_{\wr Z_{1:n+1} \wr}$.

**Step 1: Deterministic inequalities on bag-conditional risks.**

Define the following family of thresholds:

$$\hat{\lambda}_+(\wr L_{1:m} \wr, \gamma) = \inf \Big\{\lambda_0 \in \Lambda \quad : \quad \forall \quad \text{fixed } \lambda \geq \lambda_0, \quad \sum_{i=1}^{m} \frac{l_i(\lambda)}{m} \leq \gamma\Big\}.$$

When the set is empty, define $\hat{\lambda}_+(\wr L_{1:m} \wr, \gamma) = \lambda_{\max}$.

By the right-continuity of $L_i(\cdot)$ (for all $i \in \mathbb{N}$), the sum $\sum_{i=1}^{m} \frac{L_m(\lambda)}{m}$ is right-continuous for any $Z_{1:m}$, which gives us the following deterministic inequality for any $\gamma$:

$$\sum_{i=1}^{m} \frac{L_i(\lambda)}{m} \leq \gamma \qquad \forall \qquad \text{fixed } \lambda \geq \hat{\lambda}_+(\wr L_{1:m} \wr; \gamma). \tag{8}$$

**Step 2: Showing an oracle procedure deterministically controls the bag-conditional risk.**

Using the family of thresholds we defined above and applying it to the bag of loss functions for both the calibration set and the test point, define the oracle threshold as

$$\lambda_+^* := \hat{\lambda}_+(\wr L_{1:n+1} \wr, \alpha). \tag{9}$$

Now, consider the quantity $\mathbb{E}\big[L_{n+1}(\lambda_+^*) \mid \wr Z_{1:n+1} \wr = \wr z_{1:n+1} \wr\big]$, which we can view as the bag-conditional oracle risk. That is, we can interpret this quantity as the expected value that the random variable $L_{n+1}(\lambda_+^*)$ will take on, given the bag of data $\wr z_{1:n+1} \wr$. Note that conditioning on the bag of data, $\wr z_{1:n+1} \wr$, implies the event of observing the bag of loss functions, $\wr l_{1:n+1} \wr$,[7] which thereby determines the value of $\lambda_+^*$, since $\lambda_+^*$ is a symmetric function of $\wr z_{1:n+1} \wr$.

This fact, that $\lambda_+^*$ is a symmetric function of (and thus determined by) the bag of data, implies that $L_{n+1}(\lambda_+^*)$ will take a value in $\wr l_1(\lambda_+^*), ..., l_{n+1}(\lambda_+^*) \wr$ and the specific value $L_{n+1}(\lambda_+^*)$ will take on is determined by the draw of $Z_{n+1}$ from $\wr z_{1:n+1} \wr$; that is, given $\wr z_{1:n+1} \wr$, we have the equivalence $L_{n+1}(\lambda_+^*) = l_i(\lambda_+^*) \iff Z_{n+1} = z_i$ due to $g(\cdot, \lambda_+^*)$ inducing a bijection from $\wr z_1, ..., z_{n+1} \wr$ to the bag of loss values $\wr l_1(\lambda_+^*), ..., l_{n+1}(\lambda_+^*) \wr$.

First using this observation and the definition of conditional expectation, and secondly using the definition of conditional probability, we have

$$\mathbb{E}\big[L_{n+1}(\lambda_+^*) \mid \wr Z_{1:n+1} \wr = \wr z_{1:n+1} \wr\big] = \sum_{i=1}^{n+1} l_i(\lambda_+^*) \cdot \mathbb{P}\big(Z_{n+1} = z_i \mid \wr Z_{1:n+1} \wr = \wr z_{1:n+1} \wr\big)$$

$$= \sum_{i=1}^{n+1} l_i(\lambda_+^*) \cdot \frac{\mathbb{P}\big(Z_{n+1} = z_i, \wr Z_{1:n+1} \wr = \wr z_{1:n+1} \wr\big)}{\mathbb{P}\big(\wr Z_{1:n+1} \wr = \wr z_{1:n+1} \wr\big)}.$$

Applying the law of total probability over permutations ($\sigma : [n+1] \to [n+1]$) to the probabilities in the numerator and denominator of the RHS, and then simplifying using exchangeability, we have

$$\mathbb{E}\big[L_{n+1}(\lambda_+^*) \mid \wr Z_{1:n+1} \wr = \wr z_{1:n+1} \wr\big] = \sum_{i=1}^{n+1} l_i(\lambda_+^*) \cdot \frac{\sum_{\sigma : \sigma(n+1)=i} f(z_{\sigma(1)}, ..., z_{\sigma(n+1)})}{\sum_\sigma f(z_{\sigma(1)}, ..., z_{\sigma(n+1)})}$$

$$= \sum_{i=1}^{n+1} l_i(\lambda_+^*) \cdot \frac{\sum_{\sigma : \sigma(n+1)=i} f(z_1, ..., z_{n+1})}{\sum_\sigma f(z_1, ..., z_{n+1})}$$

$$= \sum_{i=1}^{n+1} l_i(\lambda_+^*) \cdot \frac{n!}{(n+1)!}$$

$$= \sum_{i=1}^{n+1} \frac{l_i(\lambda_+^*)}{n+1}$$

---

[7]That is, implies the event $\wr L_1(\cdot), ..., L_{n+1}(\cdot) \wr = \wr l_1(\cdot), ..., l_{n+1}(\cdot) \wr$, because the loss functions are defined as $l_i(\cdot) := g(z_i, \cdot)$ for all $i \in \mathbb{N}$.

Now, using Eq. (8),

$$\sum_{i=1}^{n+1} \frac{l_i(\lambda)}{n+1} \leq \alpha \qquad \forall \qquad \text{fixed } \lambda \geq \lambda_+^*, \tag{10}$$

where here by "fixed" we mean that $\lambda$ is constant conditional on the bag of data, $\lfloor Z_{1:n+1} \rfloor = \lfloor z_{1:n+1} \rfloor$.

### Step 3: Relating the empirically-selected hyperparameter to the oracle parameter

In Step 2, we have shown that an oracle procedure deterministically controls the risk, conditional on the event $\lfloor Z_{1:n+1} \rfloor = \lfloor z_{1:n+1} \rfloor$. However, this oracle procedure cannot be implemented in practice due to requiring knowledge of the test point loss, $L_{n+1}$; that is, recall we defined $\lambda_+^* := \hat{\lambda}_+(\lfloor L_{1:n+1} \rfloor, \alpha)$. In this last step of the proof we would now like to prove a deterministic relationship between the oracle parameter, $\lambda_+^*$, and the empirically-selected parameter, $\hat{\lambda}_+ := \hat{\lambda}_+(\lfloor L_{1:n}, B \rfloor, \alpha)$, to conclude valid risk control for the empirical method; that is, we would like to deterministically show $\lambda_+^* \leq \hat{\lambda}_+$.

To begin, recall that in Step 2, conditioning on the event $\lfloor Z_{1:n+1} \rfloor = \lfloor z_{1:n+1} \rfloor$ implied that $\hat{\lambda}_+(\lfloor L_{1:n+1} \rfloor, \alpha) = \hat{\lambda}_+(\lfloor l_{1:n+1} \rfloor, \alpha)$. For the empirical parameter, however, conditioning on $\lfloor Z_{1:n+1} \rfloor = \lfloor z_{1:n+1} \rfloor$ does *not* imply what may seem analogous: that is, it could be that $\hat{\lambda}_+(\lfloor L_{1:n}, B \rfloor, \alpha) \neq \hat{\lambda}_+(\lfloor l_{1:n}, B \rfloor, \alpha)$. This is because the event $\lfloor Z_{1:n+1} \rfloor = \lfloor z_{1:n+1} \rfloor$ only allows us to conclude that the $n$ random variables in $\lfloor L_{1:n} \rfloor$ take $n$ distinct values from the bag of $n+1$ observed losses, $\lfloor l_{1:n+1} \rfloor$, but we do not yet know *which* observed losses. That is, what we *can* conclude is, given $\lfloor Z_{1:n+1} \rfloor = \lfloor z_{1:n+1} \rfloor$, that

$$\lfloor Z_{1:n+1} \rfloor = \lfloor z_{1:n+1} \rfloor \implies \lfloor L_{1:n} \rfloor \in \left\{ \lfloor l_{1:n+1\setminus i} \rfloor \right\}_{i \in [n+1]},$$

where $\lfloor l_{1:n+1\setminus i} \rfloor := \lfloor l_{1:n+1} \rfloor \setminus \{l_i\}$. Accordingly, conditioning on the event $\lfloor Z_{1:n+1} \rfloor = \lfloor z_{1:n+1} \rfloor$ implies the following equivalence:

$$\lambda_+^* \leq \hat{\lambda}_+(\lfloor L_{1:n}, B \rfloor, \alpha) \text{ almost surely } \iff \lambda_+^* \leq \hat{\lambda}_+(\lfloor l_{1:n+1\setminus i}, B \rfloor, \alpha) \quad \forall \quad i \in [n+1]. \tag{11}$$

So, we will focus on proving the latter.

Recall that we have assumed that $l$ is bounded above by $B$ (i.e., $\sup_\lambda l_i(\lambda) \leq B < \infty$), for any $\lambda \in \Lambda$, deterministically we have

$$\sum_{j\in[n+1]} \frac{l_j(\lambda)}{n+1} \quad \leq \quad \frac{B}{n+1} + \sum_{j\in[n+1]\setminus i} \frac{l_j(\lambda)}{n+1} \qquad \forall \qquad i \in [n+1], \tag{12}$$

where the right-hand side of the above inequalities can be interpreted as a conservative risk obtained by replacing $l_i(\lambda)$ with the bound $B$.

Thus, for any $\lambda$, we can use (12) to conclude

$$\frac{B}{n+1} + \sum_{j\in[n+1]\setminus i} \frac{l_j(\lambda)}{n+1} \leq \alpha \implies \sum_{j\in[n+1]} \frac{l_j(\lambda)}{n+1} \leq \alpha \qquad \forall \qquad i \in [n+1]. \tag{13}$$

So, by (13) and the definition of $\hat{\lambda}_+(\lfloor l_{1:n+1\setminus i}, B \rfloor)$, it must hold that

$$\lambda_+^* \leq \hat{\lambda}_+(\lfloor l_{1:n+1\setminus i}, B \rfloor) \quad \forall \quad i \in [n+1]. \tag{14}$$

### Step 4: Analyzing the bag-conditional risk of the empirical procedure via Lipschitz continuity

Now we turn to plugging the empirical algorithm's hyperparameter into the bag-conditional risk, and then analyzing this risk using Lipschitz continuity and a type of stability called "replace-one" stability. This analysis is necessary because $\hat{\lambda}_+ = \widehat{\lambda}_+(\{\!\{Z_{1:n}, B\}\!\}, \alpha)$ is not constant conditional on $\{\!\{Z_{1:n+1}\}\!\} = \{\!\{z_{1:n+1}\}\!\}$, so Eq. (10) would not direclty apply.

Begin by expanding this quantity by LOTE over the event $Z_{n+1} = z_i$:

$$\mathbb{E}\Big[L_{n+1}(\hat{\lambda}_+) \mid \{\!\{Z_{1:n+1}\}\!\} = \{\!\{z_{1:n+1}\}\!\}\Big]$$
$$= \mathbb{E}\Big[\mathbb{E}\Big[L_{n+1}(\hat{\lambda}_+) \mid \{\!\{Z_{1:n+1}\}\!\} = \{\!\{z_{1:n+1}\}\!\}, Z_{n+1} = z_i\Big] \mid \{\!\{Z_{1:n+1}\}\!\} = \{\!\{z_{1:n+1}\}\!\}\Big]$$
$$= \sum_{i=1}^{n+1} \mathbb{E}\Big[L_{n+1}(\hat{\lambda}_+) \mid \{\!\{Z_{1:n+1}\}\!\} = \{\!\{z_{1:n+1}\}\!\}, Z_{n+1} = z_i\Big] \cdot \mathbb{P}\big(Z_{n+1} = z_i \mid \{\!\{Z_{1:n+1}\}\!\} = \{\!\{z_{1:n+1}\}\!\}\big). \quad (15)$$

Now, note that we can write the inner expectation as

$$\mathbb{E}\Big[L_{n+1}(\hat{\lambda}_+) \mid \{\!\{Z_{1:n+1}\}\!\} = \{\!\{z_{1:n+1}\}\!\}, Z_{n+1} = z_i\Big] = \mathbb{E}\Big[L_{n+1}(\hat{\lambda}_+) \mid \{\!\{Z_{1:n}\}\!\} = \{\!\{z_{1:n+1\backslash i}\}\!\}, Z_{n+1} = z_i\Big]$$
$$= \mathbb{E}\Big[L_{n+1}(\hat{\lambda}_+) \mid \{\!\{L_{1:n}\}\!\} = \{\!\{l_{1:n+1\backslash i}\}\!\}, L_{n+1} = l_i\Big].$$

And recalling that we defined $\hat{\lambda}_+ := \widehat{\lambda}_+(\{\!\{L_{1:n}, B\}\!\}, \alpha)$, we have

$$\mathbb{E}\Big[L_{n+1}(\hat{\lambda}_+) \mid \{\!\{Z_{1:n+1}\}\!\} = \{\!\{z_{1:n+1}\}\!\}, Z_{n+1} = z_i\Big]$$
$$= \mathbb{E}\Big[L_{n+1}\big(\widehat{\lambda}_+(\{\!\{L_{1:n}, B\}\!\}, \alpha)\big) \mid \{\!\{L_{1:n}\}\!\} = \{\!\{l_{1:n+1\backslash i}\}\!\}, L_{n+1} = l_i\Big]$$
$$= \mathbb{E}\Big[l_i\big(\widehat{\lambda}_+(\{\!\{l_{1:n+1\backslash i}, B\}\!\}, \alpha)\big) \mid \{\!\{L_{1:n}\}\!\} = \{\!\{l_{1:n+1\backslash i}\}\!\}, L_{n+1} = l_i\Big]$$
$$= l_i\big(\widehat{\lambda}_+(\{\!\{l_{1:n+1\backslash i}, B\}\!\}, \alpha)\big).$$

So, plugging this into Eq. (15), we have

$$\mathbb{E}\Big[L_{n+1}(\hat{\lambda}_+) \mid \{\!\{Z_{1:n+1}\}\!\} = \{\!\{z_{1:n+1}\}\!\}\Big]$$
$$= \sum_{i=1}^{n+1} l_i\big(\widehat{\lambda}_+(\{\!\{l_{1:n+1\backslash i}, B\}\!\}, \alpha)\big) \cdot \mathbb{P}\big(Z_{n+1} = z_i \mid \{\!\{Z_{1:n+1}\}\!\} = \{\!\{z_{1:n+1}\}\!\}\big)$$
$$= \sum_{i=1}^{n+1} l_i\big(\widehat{\lambda}_+(\{\!\{l_{1:n+1\backslash i}, B\}\!\}, \alpha)\big) \cdot \frac{\mathbb{P}\big(Z_{n+1} = z_i, \{\!\{Z_{1:n+1}\}\!\} = \{\!\{z_{1:n+1}\}\!\}\big)}{\mathbb{P}\big(\{\!\{Z_{1:n+1}\}\!\} = \{\!\{z_{1:n+1}\}\!\}\big)}$$

Applying the law of total probability over permutations ($\sigma : [n+1] \to [n+1]$) to the probabilities in the numerator and denominator of the RHS, and then simplifying using exchangeability, we have

$$\mathbb{E}\big[L_{n+1}(\hat{\lambda}_+) \mid \{\!\{Z_{1:n+1}\}\!\} = \{\!\{z_{1:n+1}\}\!\}\big] = \sum_{i=1}^{n+1} l_i\big(\widehat{\lambda}_+(\{\!\{l_{1:n+1\backslash i}, B\}\!\}, \alpha)\big) \cdot \frac{\sum_{\sigma:\sigma(n+1)=i} f(z_{\sigma(1)}, \ldots, z_{\sigma(n+1)})}{\sum_{\sigma} f(z_{\sigma(1)}, \ldots, z_{\sigma(n+1)})}$$
$$= \sum_{i=1}^{n+1} l_i\big(\widehat{\lambda}_+(\{\!\{l_{1:n+1\backslash i}, B\}\!\}, \alpha)\big) \cdot \frac{\sum_{\sigma:\sigma(n+1)=i} f(z_1, \ldots, z_{n+1})}{\sum_{\sigma} f(z_1, \ldots, z_{n+1})}$$
$$= \sum_{i=1}^{n+1} l_i\big(\widehat{\lambda}_+(\{\!\{l_{1:n+1\backslash i}, B\}\!\}, \alpha)\big) \cdot \frac{n!}{(n+1)!}$$
$$= \frac{1}{n+1} \sum_{i=1}^{n+1} l_i\big(\widehat{\lambda}_+(\{\!\{l_{1:n+1\backslash i}, B\}\!\}, \alpha)\big). \quad (16)$$

Importantly, note that in Eq. (16), which is the true bag-conditional risk of the empirical algorithm, that the value of $\hat{\lambda}_+ \big( \{ l_{1:n+1\setminus i}, B \}, \alpha \big)$ differs for each loss function, $l_i$, in the summation, which is why we cannot apply Eq. (10) to bound it directly. Instead, we relate Eq. (16) and Eq. (10) using Lipschitz continuity and the replace-one (in)stability of the algorithm.

For all $i \in [n]$, denote the magnitude of perturbation to the hyperparameter $\hat{\lambda}_+(\{l_{1:n}, B\}, \alpha)$ caused by replacing the loss function $l_i$ with $l_{n+1}$ as

$$\epsilon_i = \left| \hat{\lambda}_+(\{l_{1:n}, B\}, \alpha) - \hat{\lambda}_+(\{l_{1:n+1\setminus i}, B\}, \alpha) \right|.$$

Then, by $K$-Lipschitz continuity, we know that for all $i \in [n]$, that

$$\left| l_i\big(\hat{\lambda}_+(\{l_{1:n}, B\}, \alpha)\big) - l_i\big(\hat{\lambda}_+(\{l_{1:n+1\setminus i}, B\}, \alpha)\big) \right| \le K\epsilon_i.$$

Then, note that

$$\sum_{i=1}^{n+1} \frac{l_i\big(\hat{\lambda}_+(\{l_{1:n+1\setminus i}, B\}, \alpha)\big)}{n+1} \le \sum_{i=1}^{n+1} \frac{l_i\big(\hat{\lambda}_+(\{l_{1:n}, B\}, \alpha)\big) + K\epsilon_i}{n+1} = \sum_{i=1}^{n+1} \frac{l_i\big(\hat{\lambda}_+(\{l_{1:n}, B\}, \alpha)\big)}{n+1} + \frac{K}{n+1}\sum_{i=1}^{n+1} \epsilon_i$$

By Eq. (14), we know that $\hat{\lambda}_+(\{l_{1:n}, B\}, \alpha) \ge \lambda_+^*$, and because this $\hat{\lambda}_+(\{l_{1:n}, B\}, \alpha)$ does not depend on $i$, we can apply Eq. (10) to bound it above by $\alpha$, which gives

$$\sum_{i=1}^{n+1} \frac{l_i\big(\hat{\lambda}_+(\{l_{1:n+1\setminus i}, B\}, \alpha)\big)}{n+1} \le \alpha + \frac{K}{n+1}\sum_{i=1}^{n+1} \epsilon_i$$

$$\iff \mathbb{E}\big[L_{n+1}(\hat{\lambda}_+) \mid \{Z_{1:n+1}\} = \{z_{1:n+1}\}\big] \le \alpha + \frac{K}{n+1}\sum_{i=1}^{n+1} \epsilon_i.$$

Then, marginalizing over the event and applying $\epsilon$-replace-one stability, we have

$$\mathbb{E}\big[L_{n+1}(\hat{\lambda}_+)\big] \le \alpha + K\epsilon.$$

Note that the above analysis also applies for any alternative $\hat{\lambda}'_+$ that is a deterministic function of $\{l_{1:n}\}$ and where almost surely $\hat{\lambda}'_+ \ge \lambda_+^*$, and assuming, $\hat{\lambda}'_+$ is $\epsilon$-replace-one stable.

$\square$

## B.2. Proof for Conformal Policy Control Guarantee

**Notation:**

- **Data (sequences of random variables and observed values):** For all $i \in \mathbb{N}$, denote the random variable for the $i$-th data sample as $Z_i \in \mathcal{Z}$, and denote observed value of that random variable in lower case as $z_i \in \mathcal{Z}$. For example, we might have $\mathcal{Z} = \mathcal{X} \times \mathcal{A} \times \mathbb{R} \times \mathbb{R}$ if $Z = (X, A, R, L)$ is a vector of context, action, reward, and loss. Similarly, denote a sequence of random variables as $Z_{1:m} := (Z_1, ..., Z_m)$ and a sequence of observed data values as $z_{1:m} := (z_1, ..., z_m)$ for any $m \in \mathbb{N}$. To concisely denote indices, let $[m] := \{1, ..., m\}$.

- **Hyperparameters:** Let $\mathbb{R}$ be the space of hyperparameters, and let $\beta \in \mathbb{R}$ denote a particular hyperparameter.

- **Loss functions:** For some real-valued function $g : \mathcal{Z} \times \beta \to \mathbb{R}$, and for all $\beta \in \mathbb{R}$ and all $i \in \mathbb{N}$, let $L_i(\beta) := g(Z_i, \beta)$ denote the loss function determined by a random variable $Z_i$ and parameter $\beta \in \mathbb{R}$, and let $\ell_i := g(z_i, \beta)$ denote the loss function determined by the observed value $z_i$ and parameter $\beta \in \mathbb{R}$. Note that allowing the loss functions to be a function of $\beta$ does not preclude the setting where the losses are independent of $\beta$, that is where $L_i(\beta) = L_i$ for all $\beta \in \mathbb{R}$ (i.e., where changing $\beta$ does not change the loss). Such a setting motivates allowing $\beta$ to parameterize the policies.

- **Bags:** For any sequence $v_{1:m} := (v_1, ..., v_m)$, let $\lfloor v_{1:m} \rceil$ denote the *bag* (or multiset) containing the values of that sequence $v_{1:m}$ (but importantly, $\lfloor v_{1:m} \rceil$ does *not* contain the information about the order of $v_{1:m}$). For example, $\lfloor (1, 7, 42, 7) \rceil$ conveys that there is one 1, two 7s, and one 42 in the bag, but it does *not* convey the order that these appeared in the sequence $(1, 7, 42, 7)$. In particular, with this notation, for the random variables $Z_{1:m}$, let $\lfloor Z_{1:m} \rceil$ denote the bag containing those random variables; for observed values $z_{1:m}$, let $\lfloor z_{1:m} \rceil$ denote the bag containing those observed values; and thus $\lfloor Z_{1:m} \rceil = \lfloor z_{1:m} \rceil$ denotes the event of observing $\lfloor z_{1:m} \rceil$, in other words, that each $Z_i$ takes a different value in $\lfloor z_{1:m} \rceil$, but it is not yet known *which* value (for all $i \in [m]$).

- **Policies (Distributions):** Let $\pi_{1:m} := \pi_{Z_1, ..., Z_m}$ denote the policy or joint density function[8] for $Z_1, ..., Z_m$, and let $\pi_{\lfloor Z_{1:m} \rceil} := \pi_{\lfloor Z_1, ..., Z_m \rceil}$ denote the (pushforward) distribution induced by the mapping $(z_0, ..., z_m) \to \lfloor z_{1:m} \rceil$.

**Theorem B.2** (Restatement of Theorem 4.5)**.** *Assume that* $\mathbb{E}_{\pi_0}[L_t] \leq \alpha$, $L_i \in [0, B]$, $B < \infty$ *for all* $i$, *and* $\hat{\beta}$ *is* $\epsilon$-*replace-one stable. If* $w_i^{(\beta)}$ *is* $K$-*Lipschitz in* $\beta$ *for all* $i$, *then*

$$\mathbb{E}_{\pi_{0:t}^{(\hat{\beta})}}[L_t] \leq \alpha + BK\epsilon.$$

*If* $(l_i - \alpha) \cdot w_i^{(\beta)}$ *is* $K'$-*Lipschitz in* $\beta$, *then*

$$\mathbb{E}_{\pi_{0:t}^{(\hat{\beta})}}[L_t] \leq \alpha + K'\epsilon.$$

*Proof.*

For ease of notation, we conduct the proof in an online setting where only one action is taken (or sample is generated) at each policy improvement step. This will allow the indices $t = 0, 1, ...$ to denote both the timestep of policy improvement and the timestep of the action/sample, but the result will hold generally for a batch setting too (with the guarantee still holding marginally over the whole procedure).

We proceed by induction. For $t = 0$, we assumed access to a safe initial policy, $\pi_0$, satisfying our risk control criterion. That is, we assume

$$\mathbb{E}_{Z_0 \sim \pi_0}[L_0] \leq \alpha.$$

Now, for the induction hypothesis assume that risk control holds at time $t - 1$, that is, assume

$$\mathbb{E}_{Z_{0:t-1} \sim \pi_{0:t-1}^{(\hat{\beta}_1:\hat{\beta}_{t-1})}}[L_{t-1}] \leq \alpha.$$

We then want to show that risk control also holds at time $t$:

$$\mathbb{E}_{Z_{0:t} \sim \pi_{0:t}^{(\hat{\beta}_1:\hat{\beta}_t)}}[L_t] \leq \alpha,$$

where note that $\pi_{0:t}^{(\hat{\beta}_t)}(z_{0:t}) = \pi_t^{(\hat{\beta}_t)}(z_t \mid z_{0:t-1})\pi_{t-1}^{(\hat{\beta}_1:\hat{\beta}_{t-1})}(z_{0:t-1})$. Note that at time $t$, we will only consider tuning $\hat{\beta}_t$, and we treat $\hat{\beta}_1, ..., \hat{\beta}_{t-1}$ as previously tuned. Accordingly, to lighten notation, we hereon denote the expectation we wish to bound as $\mathbb{E}_{\pi_{0:t}^{(\hat{\beta})}}[L_t] := \mathbb{E}_{Z_{0:t} \sim \pi_{0:t}^{(\hat{\beta}_1:\hat{\beta}_t)}}[L_t]$, where note that we also let $\hat{\beta} := \hat{\beta}_t$.

For any control parameter $\beta$, any policies $\pi_{0:t}^{(\beta)}$, and any bag of data $\lfloor z_{0:t} \rceil$, following Tibshirani et al. (2019); Prinster et al. (2024) define the oracle normalized conformal weights as

$$\tilde{w}_i^{(\beta)} := \tilde{w}_i\big(\pi_{0:t}^{(\beta)}, \lfloor z_{0:t} \rceil\big) = \mathbb{P}_{\pi_{0:t}^{(\beta)}}(Z_t = z_i \mid \lfloor z_{0:t} \rceil) = \frac{\sum_{\sigma: \sigma(t)=i} \pi_{0:t}^{(\beta)}(z_{\sigma(0)}, ..., z_{\sigma(t)})}{\sum_\sigma \pi_{0:t}^{(\beta^*)}(z_{\sigma(0)}, ..., z_{\sigma(t)})} \tag{17}$$

for all $i \in \{0, ..., t\}$, as well as the unnormalized weights,

$$w_i^{(\beta)} := w_i\big(\pi_{0:t}^{(\beta)}, \lfloor z_{0:t} \rceil\big) = \sum_{\sigma: \sigma(t)=i} \pi_{0:t}^{(\beta)}(z_{\sigma(0)}, ..., z_{\sigma(t)}), \tag{18}$$

---

[8]In a slight abuse of notation for the policy control setting, we do not distinguish between CDFs and density functions here.

so note that $\tilde{w}_i^{(\beta)} = \frac{w_i^{(\beta)}}{\sum_{i=0}^{t} w_i^{(\beta)}}$, and define the vector $\tilde{w}_{0:t}^{(\beta)} := (\tilde{w}_0^{(\beta)}, ..., \tilde{w}_t^{(\beta)})$.

Also define the conservative (normalized) weight for an unknown test point as

$$\tilde{w}_{\max}^{(\beta)} := \sup_{z_t \in \mathcal{Z}} (\tilde{w}_t^{(\beta)}).$$

Assume there is some small, positive $\beta_{\min} \ll 1$ such that $\beta_{\min} \leq \pi_t(x)/\pi_0(x)$ for all $x \in \mathcal{X}$. This is the safe hyperparameter value, for which $\pi_t^{(\beta_{\min})} = \pi_0$.

**Step 0: Law of total expectation over draw of bag.**

Expanding $\mathbb{E}_{\pi_{0:t}^{(\hat{\beta})}}[L_t]$ via the law of total expectation over the event $\{Z_{0:t}\} = \{z_{0:t}\}$, we have

$$\mathbb{E}_{\pi_{0:t}^{(\hat{\beta})}}\left[L_t\right] = \mathbb{E}\left[\mathbb{E}_{\pi_{0:t}^{(\hat{\beta})}}\left[L_t \mid \{Z_{0:t}\} = \{z_{0:t}\}\right]\right]. \tag{19}$$

We can thus see that, to prove the desired result, it is sufficient to upper bound $\mathbb{E}_{\pi_{0:t}^{(\hat{\beta})}}\left[L_t \mid \{Z_{0:t}\} = \{z_{0:t}\}\right]$ almost surely given any draw of the bag, $\{Z_{0:t}\}$.

**Step 1: Deterministic inequalities on bag-conditional risks.**

Define the following family of hyperparameters:

$$\hat{\beta}(\{l_{1:m}\}, \tilde{w}_{1:m}^{(\cdot)}, \gamma) = \sup\left\{\beta' \in \beta \quad : \quad \forall \quad \beta \leq \beta', \quad \sum_{i=1}^{m} l_i \cdot \tilde{w}_i^{(\beta)} \leq \gamma\right\}.$$

When the set is empty, define $\hat{\beta}(\{l_{1:m}\}, \tilde{w}_{1:m}^{(\cdot)}, \gamma) = \beta_{\min}$.

By the continuity of $\tilde{w}_i^{(\cdot)}$ (and $L_i(\cdot)$) for all $i \in \mathbb{N}$, the sum $\sum_{i=1}^{m} l_i \cdot \tilde{w}_i^{(\beta)}$ is right-continuous for any $Z_{1:m}$, which gives us the following deterministic inequality for any $\gamma$:

$$\sum_{i=1}^{m} l_i \cdot \tilde{w}_i^{(\beta)} \leq \gamma \quad \forall \quad \beta \leq \hat{\beta}(\{L_{1:m}\}, \tilde{w}_{1:m}^{(\cdot)}, \gamma). \tag{20}$$

**Step 2: Showing an oracle procedure deterministically controls the bag-conditional risk.**

Using the family of thresholds we defined above and applying it to the bag of loss functions for both the calibration set and the test point, define the oracle threshold as

$$\beta^* := \hat{\beta}(\{L_{0:t}\}, \tilde{w}_{0:t}^{(\cdot)}, \alpha). \tag{21}$$

Now, consider the quantity $\mathbb{E}_{\pi_{0:t}^{(\beta^*)}}\left[L_t \mid \{Z_{0:t}\} = \{z_{0:t}\}\right]$, which we can view as the bag-conditional oracle risk. That is, we can interpret this quantity as the expected value, with respect to the policy $\pi_{0:t}^{(\beta^*)}$, that the random variable $L_t$ will take on, given the bag of data $\{z_{0:t}\}$. Note that conditioning on the bag of data, $\{z_{0:t}\}$, implies the event of observing the bag of loss functions, $\{l_{0:t}\}$,[9] which thereby determines the value of $\beta^*$, since $\beta^*$ is a function of $\{z_{0:t}\}$ (as well as the policy, $\pi_{0:t}^{(\cdot)}$, and the constraint function $g$).

---

[9]That is, implies the event $\{L_1, ..., L_t\} = \{l_1, ..., l_t\}$, because the loss functions are defined as $l_i := g(z_i)$ for all $i \in \mathbb{N}$.

Using the definition of conditional expectation, and secondly using the definition of conditional probability, we have

$$
\mathbb{E}_{\pi_{0:t}^{(\beta^*)}}\big[L_t \mid \{\!\{Z_{0:t}\}\!\} = \{\!\{z_{0:t}\}\!\}\big] = \sum_{i=0}^{t} l_i \cdot \mathbb{P}\big(Z_t = z_i \mid \{\!\{Z_{0:t}\}\!\} = \{\!\{z_{0:t}\}\!\}\big)
$$

$$
= \sum_{i=0}^{t} l_i \cdot \frac{\mathbb{P}\big(Z_t = z_i,\ \{\!\{Z_{0:t}\}\!\} = \{\!\{z_{0:t}\}\!\}\big)}{\mathbb{P}\big(\{\!\{Z_{0:t}\}\!\} = \{\!\{z_{0:t}\}\!\}\big)}.
$$

Applying the law of total probability over permutations ($\sigma : \{0, ..., t\} \rightarrow \{0, ..., t\}$) to the probabilities in the numerator and denominator of the RHS, we have

$$
\mathbb{E}_{\pi_{0:t}^{(\beta^*)}}\big[L_t \mid \{\!\{Z_{0:t}\}\!\} = \{\!\{z_{0:t}\}\!\}\big] = \sum_{i=0}^{t} l_i \cdot \frac{\sum_{\sigma : \sigma(t) = i} \pi_{0:t}^{(\beta^*)}(z_{\sigma(0)}, ..., z_{\sigma(t)})}{\sum_{\sigma} \pi_{0:t}^{(\beta^*)}(z_{\sigma(0)}, ..., z_{\sigma(t)})}
$$

$$
= \sum_{i=0}^{t} l_i \cdot \tilde{w}_i^{(\beta^*)}.
$$

Now, using Eq. (20),

$$
\sum_{i=0}^{t} l_i \cdot \tilde{w}_i^{(\beta)} \le \alpha \qquad \forall \qquad \text{fixed } \beta \le \beta^*, \tag{22}
$$

where here by "fixed" we mean that $\beta$ is constant conditional on the bag of data, $\{\!\{Z_{0:t}\}\!\} = \{\!\{z_{0:t}\}\!\}$.

### Step 3: Relating the empirically-selected hyperparameter to the oracle parameter

In Step 2, we have shown that an oracle procedure deterministically controls the risk, conditional on the event $\{\!\{Z_{0:t}\}\!\} = \{\!\{z_{0:t}\}\!\}$. However, this oracle procedure cannot be implemented in practice due to requiring knowledge of the test point loss, $L_t$, as well as the weight assigned to the test point (as a function of $\beta$), $\tilde{w}_t$; that is, recall we defined $\beta^* := \hat{\beta}(\{\!\{L_{0:t}\}\!\}, \tilde{w}_{0:t}^{(\cdot)}, \alpha)$. In this next step of the proof we would now like to prove a deterministic relationship between the oracle parameter, $\beta^*$, and the empirically-selected parameter, $\hat{\beta} := \hat{\beta}(\{\!\{L_{1:t-1}, B\}\!\}, \tilde{w}_{0:t-1}^{(\cdot)}, \tilde{w}_{\max}^{(\cdot)}, \alpha)$, to conclude valid risk control for the empirical method; that is, we would like to deterministically show $\beta^* \ge \hat{\beta}$.

To begin, recall that in Step 2, conditioning on the event $\{\!\{Z_{0:t}\}\!\} = \{\!\{z_{0:t}\}\!\}$ implied that $\hat{\beta}(\{\!\{L_{0:t}\}\!\}, \tilde{w}_{0:t}^{(\cdot)}, \alpha) = \hat{\beta}(\{\!\{l_{0:t}\}\!\}, \tilde{w}_{0:t}^{(\cdot)}, \alpha)$. For the empirical parameter, however, conditioning on $\{\!\{Z_{0:t}\}\!\} = \{\!\{z_{0:t}\}\!\}$ does *not* imply what may seem analogous: that is, it could be that $\hat{\beta}(\{\!\{L_{0:t-1}, B\}\!\}, \tilde{w}_{0:t-1}^{(\cdot)}, \tilde{w}_{\max}^{(\cdot)}, \alpha) \ne \hat{\beta}(\{\!\{l_{0:t-1}, B\}\!\}, \tilde{w}_{0:t-1}^{(\cdot)}, \tilde{w}_{\max}^{(\cdot)}, \alpha)$. This is because the event $\{\!\{Z_{0:t}\}\!\} = \{\!\{z_{0:t}\}\!\}$ only allows us to conclude that the $t - 1$ random variables in $\{\!\{L_{1:t-1}\}\!\}$ take $t - 1$ distinct values from the bag of $t$ observed losses, $\{\!\{l_{0:t}\}\!\}$, but we do not yet know *which* observed losses. That is, what we *can* conclude is, given $\{\!\{Z_{0:t}\}\!\} = \{\!\{z_{0:t}\}\!\}$, that

$$
\{\!\{Z_{0:t}\}\!\} = \{\!\{z_{0:t}\}\!\} \implies \{\!\{L_{1:t-1}\}\!\} \in \left\{ \{\!\{l_{0:t\setminus i}\}\!\} \right\}_{i \in \{0, ..., t\}},
$$

where $\{\!\{l_{0:t\setminus i}\}\!\} := \{\!\{l_{0:t}\}\!\} \setminus \{l_i\}$. Accordingly, conditioning on the event $\{\!\{Z_{0:t}\}\!\} = \{\!\{z_{0:t}\}\!\}$ implies the following equivalence:

$$
\beta^* \ge \hat{\beta}(\{\!\{L_{0:t-1}, B\}\!\}, \tilde{w}_{0:t-1}^{(\cdot)}, \tilde{w}_{\max}^{(\cdot)}, \alpha) \text{ almost surely} \iff \beta^* \ge \hat{\beta}(\{\!\{l_{0:t\setminus i}, B\}\!\}, \tilde{w}_{0:t-1}^{(\cdot)}, \tilde{w}_{\max}^{(\cdot)}, \alpha) \quad \forall \quad i \in \{0, ..., t\}. \tag{23}
$$

So, we will focus on proving the latter.

Recall that we have assumed that $l$ is bounded above by $B$, for any $\beta \in \mathbb{R}$, and by the definition of $\tilde{w}_{\max}^{(\cdot)}$, deterministically we have

$$
\sum_{j \in \{0, ..., t\}} l_j \cdot \tilde{w}_j^{(\beta)} \quad \le \quad B \cdot \tilde{w}_{\max}^{(\beta)} + \sum_{j \in \{0, ..., t\} \setminus i} l_j \cdot \tilde{w}_j^{(\beta)} \quad \forall \quad i \in \{0, ..., t\}, \tag{24}
$$

where the right-hand side of the above inequalities can be interpreted as a conservative risk obtained by replacing $l_i$ with the bound $B$.

Thus, for any $\beta$, we can use (24) to conclude

$$B \cdot \tilde{w}_{\max}^{(\beta)} + \sum_{j \in \{0,...,t\} \setminus i} l_j \cdot \tilde{w}_i^{(\beta)} \leq \alpha \implies \sum_{j \in \{0,...,t\}} l_j \cdot \tilde{w}_j^{(\beta)} \leq \alpha \quad \forall \quad i \in \{0,...,t\}. \tag{25}$$

So, by Eq. (25) and the definition of $\hat{\beta}(\wr l_{0:t \setminus i}, B \wr)$, it must hold that

$$\beta^* \geq \hat{\beta}(\wr l_{0:t \setminus i}, B \wr, \tilde{w}_{0:t-1}^{(\cdot)}, \tilde{w}_{\max}^{(\cdot)}, \alpha) \quad \forall \quad i \in \{0,...,t\}. \tag{26}$$

**Step 4: Analyzing the bag-conditional risk of the empirical procedure via Lipschitz continuity**

Now we turn to plugging the empirical algorithm's hyperparameter into the bag-conditional risk, and then analyzing this risk using Lipschitz continuity and a type of stability called "replace-one" stability. This analysis is necessary because $\hat{\beta} = \hat{\beta}(\wr L_{0:t-1}, B \wr, \tilde{w}_{0:t-1}^{(\cdot)}, \tilde{w}_{\max}^{(\cdot)}, \alpha)$ is not constant conditional on $\wr Z_{0:t} \wr = \wr z_{0:t} \wr$, so Eq. (22) would not directly apply.

Begin by expanding the quantity $\mathbb{E}_{\pi_{0:t}^{(\hat{\beta})}}\left[L_t \mid \wr Z_{0:t} \wr = \wr z_{0:t} \wr\right]$ by LOTE over the event $Z_t = z_i$:

$$\mathbb{E}_{\pi_{0:t}^{(\hat{\beta})}}\left[L_t \mid \wr Z_{0:t} \wr = \wr z_{0:t} \wr\right]$$

$$= \mathbb{E}_{\pi_{0:t}^{(\hat{\beta})}}\left[\mathbb{E}_{\pi_{0:t}(\hat{\beta}(\wr L_{0:t-1} \wr))}\left[L_t \mid \wr Z_{0:t} \wr = \wr z_{0:t} \wr, Z_t = z_i\right] \mid \wr Z_{0:t} \wr = \wr z_{0:t} \wr\right]$$

$$= \sum_{i=0}^{t} \mathbb{E}_{\pi_{0:t}^{(\hat{\beta})}}\left[L_t \mid \wr Z_{0:t} \wr = \wr z_{0:t} \wr, Z_t = z_i\right] \cdot \mathbb{P}_{\pi_{0:t}^{(\hat{\beta})}}\left(Z_t = z_i \mid \wr Z_{0:t} \wr = \wr z_{0:t} \wr\right)$$

$$= \sum_{i=0}^{t} l_i \cdot \mathbb{P}_{\pi_{0:t}^{(\hat{\beta})}}\left(Z_t = z_i \mid \wr Z_{0:t} \wr = \wr z_{0:t} \wr\right). \tag{27}$$

Writing out the conditional probability in Eq. (27) using the definition of conditional probability, and then (in both the numerator and denominator) the law of total probability and chain rule,

$$\mathbb{P}_{\pi_{0:t}^{(\hat{\beta})}}\left(Z_t = z_i \mid \wr Z_{0:t} \wr = \wr z_{0:t} \wr\right)$$

$$= \frac{\sum_{\sigma:\sigma(t)=i} \pi_{0:t}^{(\hat{\beta})}(z_{\sigma(0)}, ..., z_{\sigma(t)})}{\sum_{\sigma} \pi_{0:t}^{(\hat{\beta})}(z_{\sigma(0)}, ..., z_{\sigma(t)})}$$

$$= \frac{\sum_{\sigma:\sigma(t)=i} \pi_{t|0:t-1}^{(\hat{\beta})}(z_{\sigma(t)} \mid z_{\sigma(0)}, ..., z_{\sigma(t-1)}) \cdot \pi_{t-1|0:t-2}(z_{\sigma(t-1)} \mid z_{\sigma(0)}, ..., z_{\sigma(t-2)}) \cdots \pi_0(z_{\sigma(0)})}{\sum_{\sigma} \pi_{t|0:t-1}^{(\hat{\beta})}(z_{\sigma(t)} \mid z_{\sigma(0)}, ..., z_{\sigma(t-1)}) \cdot \pi_{t-1|0:t-2}(z_{\sigma(t-1)} \mid z_{\sigma(0)}, ..., z_{\sigma(t-2)}) \cdots \pi_0(z_{\sigma(0)})},$$

and by expanding the summation in the denominator, this is equivalent to

$$\mathbb{P}_{\pi_{0:t}^{(\hat{\beta})}}\left(Z_t = z_i \mid \wr Z_{0:t} \wr = \wr z_{0:t} \wr\right)$$

$$= \frac{\sum_{\sigma:\sigma(t)=i} \pi_{t|0:t-1}^{(\hat{\beta})}(z_{\sigma(t)} \mid z_{\sigma(0)}, ..., z_{\sigma(t-1)}) \cdot \pi_{t-1|0:t-2}(z_{\sigma(t-1)} \mid z_{\sigma(0)}, ..., z_{\sigma(t-2)}) \cdots \pi_0(z_{\sigma(0)})}{\sum_i \sum_{\sigma:\sigma(t)=i} \pi_{t|0:t-1}^{(\hat{\beta})}(z_{\sigma(t)} \mid z_{\sigma(0)}, ..., z_{\sigma(t-1)}) \cdot \pi_{t-1|0:t-2}(z_{\sigma(t-1)} \mid z_{\sigma(0)}, ..., z_{\sigma(t-2)}) \cdots \pi_0(z_{\sigma(0)})}.$$

In this expanded form, examine the factor $\pi_{t|0:t-1}^{(\hat{\beta})}(z_{\sigma(t)} \mid z_{\sigma(0)}, ..., z_{\sigma(t-1)})$, and recalling that $\hat{\beta} := \hat{\beta}(\wr L_{0:t-1}, B \wr, \tilde{w}_{0:t-1}^{(\cdot)}, \tilde{w}_{\max}^{(\cdot)}, \alpha)$, note that the conditioning on $z_{\sigma(0)}, ..., z_{\sigma(t-1)}$ inside the summation $\sum_{\sigma:\sigma(t)=i}$, where

the permutations are $\sigma : \sigma(t) = i$, implies $Z_t = z_{\sigma(t)} = z_i \implies \wr L_{0:t-1} \wr = \wr l_{0:t \setminus i} \wr$ (that is, implies a specific realization of $\wr L_{0:t-1} \wr$ among $t$ possible bags of observed losses). Accordingly, let us denote $\hat{\beta}^{\setminus i} := \hat{\beta}(\wr l_{0:t \setminus i}, B \wr, \tilde{w}^{(\cdot)}_{0:t-1}, \tilde{w}^{(\cdot)}_{\max}, \alpha)$. Then, note that the numerator and the analogous inner summation in the denominator are the unnormalized weights in Eq. (18), that is, we can concisely write those summations as $\sum_{\sigma : \sigma(t) = i}(\cdots) = w_i^{(\beta^{\setminus i})}$. That is, we have

$$\mathbb{P}_{\pi^{(\hat{\beta})}_{0:t}}\big( Z_t = z_i \mid \wr Z_{0:t} \wr = \wr z_{0:t}\, \wr \big) = \frac{w_i^{(\beta^{\setminus i})}}{\sum_j w_j^{(\beta^{\setminus j})}}. \tag{28}$$

Plugging this into Eq. (27), we have

$$\mathbb{E}_{\pi^{(\hat{\beta})}_{0:t}}\Big[ L_t \mid \wr Z_{0:t} \wr = \wr z_{0:t}\, \wr \Big] = \sum_{i=0}^{t} l_i \cdot \frac{w_i^{(\beta^{\setminus i})}}{\sum_j w_j^{(\beta^{\setminus j})}} = \frac{1}{\sum_j w_j^{(\beta^{\setminus j})}} \sum_{i=0}^{t} l_i \cdot w_i^{(\beta^{\setminus i})} \tag{29}$$

Importantly, note that in Eq. (29), which can be viewed as the true bag-conditional risk of the empirical algorithm, that the unnormalized weight in the numerator, $w_i^{(\beta^{\setminus i})}$, differs for each loss function, $l_i$, in the summation, and moreover the same is true in the denominator's summation, which is why we cannot apply Eq. (22) to bound it directly.

Instead, we now move toward relating Eq. (29) and Eq. (22) using Lipschitz continuity and the replace-one (in)stability of the algorithm. However, although in the analogous step assuming parameterized losses (Section B.1) we were able to directly apply Lipschitz continuity at this stage, here we must be more careful. That is, although in Section B.1 we could conservatively assume that each replace-one perturbation $\epsilon_i$ increased its corresponding loss by at most $K\epsilon_i$, a subtle difference here is that if we artificially increased the unnormalized weight of certain (e.g., smaller-loss) indices by some amount, this may *decrease* the overall bag-conditional weighted risk after re-normalizing the probability weights, and thus not result in a valid upper bound. In other words, conservative estimates of losses monotonically result in conservative estimates of the risk (in Section B.1), but here, conservative estimates of (unnormalized) weights introduce an additional source of non-monotonicity that may not result in a conservative estimate of the weighted (bag-conditional) risk.

For all $i \in [n]$, denote the magnitude of perturbation to the hyperparameter $\hat{\beta}\big( \wr l_{0:t \setminus i}, B \wr, \tilde{w}^{(\cdot)}_{0:t-1}, \tilde{w}^{(\cdot)}_{\max}, \alpha \big)$ caused by replacing the loss function $l_i$ with $l_t$ as

$$\epsilon_i = \Big| \hat{\beta}\big( \wr l_{0:t-1}, B \wr, \tilde{w}^{(\cdot)}_{0:t-1}, \tilde{w}^{(\cdot)}_{\max}, \alpha \big) - \hat{\beta}\big( \wr l_{0:t \setminus i}, B \wr, \tilde{w}^{(\cdot)}_{0:t-1}, \tilde{w}^{(\cdot)}_{\max}, \alpha \big) \Big|$$
$$= \Big| \hat{\beta}^{\setminus t} - \hat{\beta}^{\setminus i} \Big|.$$

Making the substitution $w_i^{(\hat{\beta}^{\setminus i})} = w_i^{(\hat{\beta}^{\setminus i})} + (w_i^{(\hat{\beta}^{\setminus t})} - w_i^{(\hat{\beta}^{\setminus t})})$ in Eq. (29) and then rearranging terms, we have

$$\mathbb{E}_{\pi^{(\hat{\beta})}_{0:t}}\Big[ L_t \mid \wr Z_{0:t} \wr = \wr z_{0:t}\, \wr \Big] = \frac{1}{\sum_j w_j^{(\hat{\beta}^{\setminus j})}} \sum_{i=0}^{t} l_i \cdot w_i^{(\hat{\beta}^{\setminus t})} + (l_i \cdot w_i^{(\hat{\beta}^{\setminus i})} - l_i \cdot w_i^{(\hat{\beta}^{\setminus t})})$$
$$= \frac{1}{\sum_j w_j^{(\hat{\beta}^{\setminus j})}} \sum_{i=0}^{t} l_i \cdot w_i^{(\hat{\beta}^{\setminus t})} + \frac{1}{\sum_j w_j^{(\hat{\beta}^{\setminus j})}} \sum_{i=0}^{t} (l_i \cdot w_i^{(\hat{\beta}^{\setminus i})} - l_i \cdot w_i^{(\hat{\beta}^{\setminus t})}).$$

Now multiplying the first term by $1 = \frac{\sum_j w_j^{(\hat{\beta}^{\setminus t})}}{\sum_j w_j^{(\hat{\beta}^{\setminus t})}}$,

$$\mathbb{E}_{\pi^{(\hat{\beta})}_{0:t}}\Big[ L_t \mid \wr Z_{0:t} \wr = \wr z_{0:t}\, \wr \Big] = \frac{\sum_j w_j^{(\hat{\beta}^{\setminus t})}}{\sum_j w_j^{(\hat{\beta}^{\setminus t})}} \frac{1}{\sum_j w_j^{(\hat{\beta}^{\setminus j})}} \sum_{i=0}^{t} l_i \cdot w_i^{(\hat{\beta}^{\setminus t})} + \frac{1}{\sum_j w_j^{(\hat{\beta}^{\setminus j})}} \sum_{i=0}^{t} (l_i \cdot w_i^{(\hat{\beta}^{\setminus i})} - l_i \cdot w_i^{(\hat{\beta}^{\setminus t})})$$
$$= \frac{\sum_j w_j^{(\hat{\beta}^{\setminus t})}}{\sum_j w_j^{(\hat{\beta}^{\setminus j})}} \sum_{i=0}^{t} \frac{l_i \cdot w_i^{(\hat{\beta}^{\setminus t})}}{\sum_j w_j^{(\hat{\beta}^{\setminus t})}} + \frac{1}{\sum_j w_j^{(\hat{\beta}^{\setminus j})}} \sum_{i=0}^{t} (l_i \cdot w_i^{(\hat{\beta}^{\setminus i})} - l_i \cdot w_i^{(\hat{\beta}^{\setminus t})}),$$

Then by Eq. (22), we know that $\sum_{i=0}^{t} \frac{l_i \cdot w_i^{(\hat{\beta}\backslash t)}}{\sum_j w_j^{(\hat{\beta}\backslash t)}} \le \alpha$ (note that $\hat{\beta}\backslash t$ is fixed with respect to the summations in the term), so

$$\mathbb{E}_{\pi_{0:t}^{(\hat{\beta})}}\left[L_t \mid \lfloor Z_{0:t} \rfloor = \lfloor z_{0:t} \rfloor\right] \le \frac{\sum_j w_j^{(\hat{\beta}\backslash t)}}{\sum_j w_j^{(\hat{\beta}\backslash j)}} \cdot \alpha + \frac{1}{\sum_j w_j^{(\hat{\beta}\backslash j)}} \sum_{i=0}^{t} (l_i \cdot w_i^{(\hat{\beta}\backslash i)} - l_i \cdot w_i^{(\hat{\beta}\backslash t)}). \tag{30}$$

Plugging in $w_j^{(\hat{\beta}\backslash t)} = w_j^{(\hat{\beta}\backslash t)} + (w_j^{(\hat{\beta}\backslash j)} - w_j^{(\hat{\beta}\backslash j)})$ into the numerator of the first term of the RHS of (30) and rearranging terms,

$$\frac{\sum_j w_j^{(\hat{\beta}\backslash t)}}{\sum_j w_j^{(\hat{\beta}\backslash j)}} \cdot \alpha + \frac{1}{\sum_j w_j^{(\hat{\beta}\backslash j)}} \sum_{i=0}^{t} (l_i \cdot w_i^{(\hat{\beta}\backslash i)} - l_i \cdot w_i^{(\hat{\beta}\backslash t)})$$

$$= \frac{\sum_j (w_j^{(\hat{\beta}\backslash t)} + w_j^{(\hat{\beta}\backslash j)} - w_j^{(\hat{\beta}\backslash j)})}{\sum_j w_j^{(\hat{\beta}\backslash j)}} \cdot \alpha + \frac{1}{\sum_j w_j^{(\hat{\beta}\backslash j)}} \sum_{i=0}^{t} (l_i \cdot w_i^{(\hat{\beta}\backslash i)} - l_i \cdot w_i^{(\hat{\beta}\backslash t)})$$

$$= \alpha + \frac{\sum_j (w_j^{(\hat{\beta}\backslash t)} - w_j^{(\hat{\beta}\backslash j)}) \cdot \alpha}{\sum_j w_j^{(\hat{\beta}\backslash j)}} + \frac{\sum_{i=0}^{t} (l_i \cdot w_i^{(\hat{\beta}\backslash i)} - l_i \cdot w_i^{(\hat{\beta}\backslash t)})}{\sum_j w_j^{(\hat{\beta}\backslash j)}}$$

$$= \alpha + \frac{\sum_{i=0}^{t} \alpha \cdot w_i^{(\hat{\beta}\backslash t)} - \alpha \cdot w_i^{(\hat{\beta}\backslash i)} + l_i \cdot w_i^{(\hat{\beta}\backslash i)} - l_i \cdot w_i^{(\hat{\beta}\backslash t)}}{\sum_j w_j^{(\hat{\beta}\backslash j)}}$$

$$= \alpha + \frac{\sum_{i=0}^{t} l_i \cdot (w_i^{(\hat{\beta}\backslash i)} - w_i^{(\hat{\beta}\backslash t)}) - \alpha \cdot (w_i^{(\hat{\beta}\backslash i)} - w_i^{(\hat{\beta}\backslash t)})}{\sum_j w_j^{(\hat{\beta}\backslash j)}}$$

$$= \alpha + \frac{\sum_{i=0}^{t} (l_i - \alpha) \cdot (w_i^{(\hat{\beta}\backslash i)} - w_i^{(\hat{\beta}\backslash t)})}{\sum_j w_j^{(\hat{\beta}\backslash j)}}. \tag{31}$$

Note that $(l_i - \alpha) \le B$, so we obtain the inequality

$$\frac{\sum_j w_j^{(\hat{\beta}\backslash t)}}{\sum_j w_j^{(\hat{\beta}\backslash j)}} \cdot \alpha + \frac{1}{\sum_j w_j^{(\hat{\beta}\backslash j)}} \sum_{i=0}^{t} (l_i \cdot w_i^{(\hat{\beta}\backslash i)} - l_i \cdot w_i^{(\hat{\beta}\backslash t)}) \quad \le \quad \alpha + B \frac{\sum_{i=0}^{t} (w_i^{(\hat{\beta}\backslash i)} - w_i^{(\hat{\beta}\backslash t)})}{\sum_j w_j^{(\hat{\beta}\backslash j)}}. \tag{32}$$

Assuming that $w_i^{(\beta)}$ is $K$-Lipschitz in $\beta$, we can bound the RHS of Eq. (32) as

$$\alpha + B \frac{\sum_{i=0}^{t} (w_i^{(\hat{\beta}\backslash i)} - w_i^{(\hat{\beta}\backslash t)})}{\sum_j w_j^{(\hat{\beta}\backslash j)}} \quad \le \quad \alpha + B \frac{\sum_{i=0}^{t} K \epsilon_i}{\sum_j w_j^{(\hat{\beta}\backslash j)}}. \tag{33}$$

Using transitive property across Inequality (30), Equation (31), Inequality (32), and Inequality (33), we have

$$\mathbb{E}_{\pi_{0:t}^{(\hat{\beta})}}\left[L_t \mid \lfloor Z_{0:t} \rfloor = \lfloor z_{0:t} \rfloor\right] \le \alpha + BK \frac{\sum_{i=0}^{t} \epsilon_i}{\sum_j w_j^{(\hat{\beta}\backslash j)}},$$

and marginalizing over the event $\lfloor Z_{0:t} \rfloor = \lfloor z_{0:t} \rfloor$ gives

$$\mathbb{E}_{\pi_{0:t}^{(\hat{\beta})}}\left[L_t\right] \leq \alpha + BK\epsilon. \tag{34}$$

Alternatively, if from Eq. (31) we instead assumed that $(l_i - \alpha) \cdot w_i^{(\beta)}$ is $K$ Lipschitz in $\beta$, then from Eq. (31) we obtain the inequality

$$\alpha + \frac{\sum_{i=0}^t (l_i - \alpha)(w_i^{(\hat{\beta}\setminus i)} - w_i^{(\hat{\beta}\setminus t)})}{\sum_j w_j^{(\hat{\beta}\setminus j)}} \quad \leq \quad \alpha + \frac{\sum_{i=0}^t K\epsilon_i}{\sum_j w_j^{(\hat{\beta}\setminus j)}}, \tag{35}$$

and using transitive property across Inequality (30), Equation (31), and Inequality (35), we have

$$\mathbb{E}_{\pi_{0:t}^{(\hat{\beta})}}\left[L_t \mid \langle Z_{0:t}\rangle = \langle z_{0:t}\rangle\right] \leq \alpha + K\frac{\sum_{i=0}^t \epsilon_i}{\sum_j w_j^{(\hat{\beta}\setminus j)}},$$

and marginalizing over the event $\langle Z_{0:t}\rangle = \langle z_{0:t}\rangle$ gives

$$\mathbb{E}_{\pi_{0:t}^{(\hat{\beta})}}\left[L_t\right] \leq \alpha + K\epsilon. \tag{36}$$

$\square$

### B.3. Counterexample to gCRC risk control on arbitrary bounded losses

**Proposition B.3.** *The following assumptions are not sufficient for $\hat{\lambda}_+$ as defined in Eq. (4) to achieve the risk control guarantee at level $\alpha$:*

- *Right-continuity of $L_i(\lambda)$ in $\lambda$,*

- *Existence of a safe hyperparameter: For $\lambda_{\max} := \sup \Lambda \in \Lambda$, that $L_i(\lambda_{\max}) \leq \alpha$,*

- *Boundedness of the loss functions: $\sup_\lambda L_i(\lambda) \leq B < \infty$ almost surely.*

*Proof.* This counterexample assumes exchangeable data. Let $\Lambda = \{\lambda_1, \lambda_2, \lambda_3, \lambda_{\max}\}$ where $\lambda_1 < \lambda_2 < \lambda_3 < \lambda_{\max}$. Fix $\alpha \in (0, 1]$. Consider the following $n + 1 = 3$ loss functions, where $B = \frac{3}{2}\alpha$:

$$l_1(\lambda) = \begin{cases} B, & \lambda = \lambda_1 \\ B/2, & \lambda = \lambda_2 \\ 0, & \lambda = \lambda_3 \\ 0, & \lambda = \lambda_{\max} \end{cases}, \qquad l_2(\lambda) = \begin{cases} B/2, & \lambda = \lambda_1 \\ B, & \lambda = \lambda_2 \\ 0, & \lambda = \lambda_3 \\ 0, & \lambda = \lambda_{\max} \end{cases}, \qquad l_3(\lambda) = \begin{cases} B/2, & \lambda = \lambda_1 \\ 0, & \lambda = \lambda_2 \\ B, & \lambda = \lambda_3 \\ 0, & \lambda = \lambda_{\max} \end{cases}. \tag{37}$$

Note that $\max_{\lambda \in \Lambda} l_i(\lambda) = B$ and $l_i(\lambda_{\max}) = 0 < \alpha$ for all $i \in [3]$, satisfying our assumptions.

Consider the bag-conditional empirical risk,

$$\mathbb{E}[L_3(\hat{\lambda}) \mid \langle L_{1:3}\rangle = \langle l_{1:3}\rangle] = \mathbb{E}\left[\ \mathbb{E}[L_3(\hat{\lambda}) \mid \langle L_{1:3}\rangle = \langle l_{1:3}\rangle, L_3 = l_i]\ \middle|\ \langle L_{1:3}\rangle = \langle l_{1:3}\rangle\right], \tag{38}$$

where the inner expectation on the righthand side conditions on the value of the bag of loss functions, $\langle L_{1:3}\rangle = \langle l_{1:3}\rangle$, and additionally on the event that the test loss takes the value of the $i$-th loss function in the bag, $L_3 = l_i$. Note that the

test loss at the empirical threshold, $L_3(\hat{\lambda})$, is a deterministic function of the values of $\wr L_{1:3} \wr$ and $L_3$. Therefore, this inner expectation is simply

$$\mathbb{E}\left[L_3(\hat{\lambda}) \mid \wr L_{1:3} \wr = \wr l_{1:3} \wr, L_3 = l_i\right] = l_i(\hat{\lambda}(\wr l_{1:3 \setminus i}, B \wr, \alpha)). \tag{39}$$

Substituting this expression for the inner expectation in Eq. (38), we can write the bag-conditional empirical risk as

$$\mathbb{E}[L_3(\hat{\lambda}) \mid \wr L_{1:3} \wr = \wr l_{1:3} \wr] = \mathbb{E}[\, l_i(\hat{\lambda}(\wr l_{1:3 \setminus i}, B \wr, \alpha)) \mid \wr L_{1:3} \wr = \wr l_{1:3} \wr] = \frac{1}{3} \sum_{i=1}^{3} l_i(\hat{\lambda}(\wr l_{1:3 \setminus i}, B \wr, \alpha)). \tag{40}$$

We now calculate this quantity for the bag of loss functions given in Eqs. (37). Each case or "world" below computes one summand in the mean in Eq. (40). To simplify forthcoming calculations, note that

$$\hat{\lambda}(\wr l_{1:3 \setminus i}, B \wr, \alpha) = \min\{\lambda' \in \Lambda : \forall \lambda \geq \lambda', \sum_{j \neq i} l_j(\lambda) \leq 3\alpha - B = 2B - B = B\}.$$

- **Case 1:** $L_3 = l_1$. We have

$$\hat{\lambda}(\wr l_2, l_3, B \wr, \alpha) = \min\{\lambda' \in \Lambda : \forall \lambda \geq \lambda', l_2(\lambda) + l_3(\lambda) \leq B\} \tag{41}$$
$$= \lambda_1, \tag{42}$$

so $L_3(\hat{\lambda}) = l_1(\hat{\lambda}(\wr l_2, l_3, B \wr, \alpha)) = l_1(\lambda_1) = B$.

- **Case 2:** $L_3 = l_2$. We have

$$\hat{\lambda}(\wr l_1, l_3, B \wr, \alpha) = \min\{\lambda' \in \Lambda : \forall \lambda \geq \lambda', l_1(\lambda) + l_3(\lambda) \leq B\} \tag{43}$$
$$= \lambda_2, \tag{44}$$

so $L_3(\hat{\lambda}) = l_2(\hat{\lambda}(\wr l_1, l_3, B \wr, \alpha)) = l_2(\lambda_2) = B$.

- **Case 3:** $L_3 = l_3$. We have

$$\hat{\lambda}(\wr l_1, l_2, B \wr, \alpha) = \min\{\lambda' \in \Lambda : \forall \lambda \geq \lambda', l_1(\lambda) + l_2(\lambda) \leq B\} \tag{45}$$
$$= \lambda_3, \tag{46}$$

so $L_3(\hat{\lambda}) = l_3(\hat{\lambda}(\wr l_1, l_2, B \wr, \alpha)) = l_3(\lambda_3) = B$.

Plugging these values into Eq. (40) gives

$$\mathbb{E}[L_3(\hat{\lambda}) \mid \wr L_{1:3} \wr = \wr l_{1:3} \wr] = \frac{1}{3} \sum_{i=1}^{3} l_i(\hat{\lambda}(\wr l_{1:3 \setminus i}, B \wr, \alpha))$$
$$= \frac{1}{3}(B + B + B)$$
$$= B = \frac{3}{2}\alpha > \alpha.$$

$\square$

## B.4. Conservative adjustment to gCRC for level $\alpha$ validity if $K$ is known

Assume the gCRC setting of Theorem 4.2 in Section 4.1 (although an analogous result would also hold for conformal policy control). In practice, the deviation of the empirical gCRC algorithm due to replacing one sample can be bounded above by

$$\hat{\epsilon}_i := \max_{b \in \{0, B\}} \left| \hat{\lambda}_+(\wr l_{1:n}, B \wr, \alpha) - \hat{\lambda}_+(\wr l_{1:n \setminus i}, b, B \wr, \alpha) \right|,$$

and if the Lipschitz constant for the loss functions is known, a conservative significance level can be obtained as

$$\hat{\alpha} := \sup\left(\left\{\alpha' \le \alpha \quad : \quad \alpha' + \frac{K}{n+1}\sum_{i=1}^{n+1}\hat{\epsilon}_i(\alpha') \le \alpha\right\}\right).$$

Then, the procedure run at $\hat{\alpha}$ attains level $\alpha$ validity.

**Proposition B.4.** *In the gCRC setting of Theorem 4.2,*

$$\mathbb{E}\left[L_{n+1}\left(\hat{\lambda}_+(\hat{\alpha})\right)\right] \le \alpha.$$

# C. Detailed Methodology

This appendix provides implementation details for Conformal Policy Control. Section C.1 describes the calibration procedure for selecting the likelihood ratio bound $\beta$ via weighted conformal prediction, including grid construction, normalization constant estimation, and conformal weighting. Section C.2 describes sampling algorithms for drawing actions from the constrained policy, including rejection sampling and Independence Metropolis-Hastings.

## C.1. Conformal Calibration of $\beta$

This section details the procedure for calibrating the likelihood ratio bound $\beta$ via weighted conformal risk control. We first define the likelihood ratio and describe how to construct a grid of candidate values (Section C.1.1). We then show how to estimate the normalization constant $\psi(\beta)$ for each candidate via importance-weighted Monte Carlo integration (Section C.1.2). Finally, we describe the conformal weighting scheme that accounts for distribution shift between calibration and deployment, and the criterion for selecting the largest $\beta$ that controls risk at level $\alpha$ (Section C.1.3).

*Remark* C.1. Throughout this section we write $\pi_0$ for the reference policy used in the likelihood ratio, which generalizes $\pi_0$ from the main text: $\pi_0$ may be the initial safe policy $\pi_0$ or the previous constrained policy $\pi_{t-1}^{(\beta_{t-1})}$. We also assume a fixed context distribution $p(X)$ shared by the safe and optimized policies. Under this assumption, the context marginal cancels in the likelihood ratio $\pi_t(A_i \mid X_i)/\pi_0(A_i \mid X_i)$, and it suffices to work with conditional densities evaluated at the same context $X_i$.

### C.1.1. LIKELIHOOD RATIOS AND GRID PREPARATION

For each action $x$, define the likelihood ratio

$$\mathrm{LR}(x) := \frac{\pi_t(x)}{\pi_0(x)}, \tag{47}$$

where $\pi_t$ is the current unconstrained policy and $\pi_0$ is either the initial policy $\pi_0$ or the previous constrained policy $\pi_{t-1}^{(\beta_{t-1})}$.

To calibrate $\beta$, we require two sets of samples:

- **Calibration data** $\mathcal{D}_{\mathrm{cal}}$: samples from previous rounds with feasibility labels, used to compute conformal weights.

- **Proposal data** $\mathcal{D}_{\mathrm{prop}}$: fresh samples from $\pi_t$, used to estimate the normalization constant $\psi(\beta)$.

We construct a grid $G$ of candidate $\beta$ values from the observed likelihood ratios across both datasets:

$$G = \text{PREPAREGRID}\left(\{\mathrm{LR}(x) : x \in \mathcal{D}_{\mathrm{cal}} \cup \mathcal{D}_{\mathrm{prop}}\}\right). \tag{48}$$

The grid is sorted in increasing order, and we search from the most conservative (smallest $\beta$) to the most permissive, stopping at the first $\beta$ where the weighted infeasibility rate exceeds $\alpha$. See Figure 7 for an illustration.

### C.1.2. ESTIMATING THE NORMALIZATION CONSTANT $\psi(\beta)$

Recall that the constrained policy is defined as

$$\pi^{(\beta)}(x) = \frac{\min\{\pi_t(x), \beta \cdot \pi_0(x)\}}{\psi(\beta)}, \tag{49}$$

where $\psi(\beta) = \int \min\{\pi_t(x), \beta \cdot \pi_0(x)\}\, dx$ is the normalization constant. We estimate $\psi(\beta)$ via importance-weighted Monte Carlo integration using the proposal data $\mathcal{D}_{\mathrm{prop}}$.

When sampling from the optimistic policy $\pi_t$:

$$\psi(\beta) = \mathbb{E}_{\pi_t}\left[\frac{\min\{\pi_t(x), \beta \cdot \pi_0(x)\}}{\pi_t(x)}\right] = \mathbb{E}_{\pi_t}\left[\min\left\{\frac{\beta}{\mathrm{LR}(x)}, 1\right\}\right] \tag{50}$$

$$\approx \frac{1}{|\mathcal{D}_{\mathrm{prop}}|} \sum_{x \in \mathcal{D}_{\mathrm{prop}}} \min\left\{\frac{\beta}{\mathrm{LR}(x)}, 1\right\}. \tag{51}$$

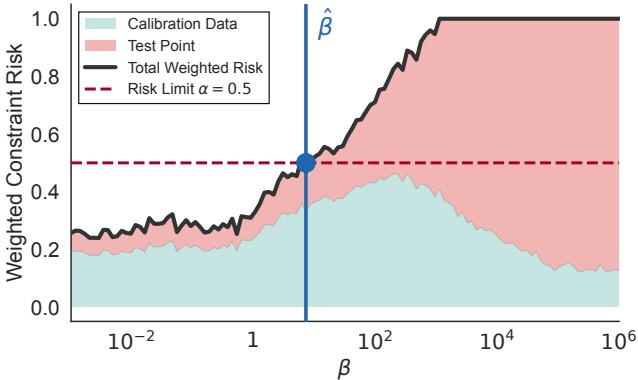

*Figure 7.* An illustration of fitting $\hat{\beta}$ to calibration data drawn from $\pi_0$. We draw a test point from the optimized policy and conservatively assume the loss attains the upper bound $B$. As $\beta$ increases, the contribution of the test point to the weighted average constraint violation increases. We search $\beta$ in increasing order, recomputing the conformal weights for each setting of $\beta$ and halting at the first point $\hat{\beta}$ where the average constraint violation exceeds $\alpha$.

When sampling from the safe policy $\pi_0$:

$$\psi(\beta) = \mathbb{E}_{\pi_0}\left[\frac{\min\{\pi_t(x), \beta \cdot \pi_0(x)\}}{\pi_0(x)}\right] = \mathbb{E}_{\pi_0}\left[\min\{\mathrm{LR}(x), \beta\}\right] \tag{52}$$

$$\approx \frac{1}{|\mathcal{D}_{\mathrm{prop}}|} \sum_{x \in \mathcal{D}_{\mathrm{prop}}} \min\{\mathrm{LR}(x), \beta\}. \tag{53}$$

In practice, all computations are performed in log space for numerical stability. Writing $r_i = \log \mathrm{LR}(x_i)$ for each sample, the log normalization constant is

$$\log \psi(\beta) = \mathrm{logsumexp}\left(\{\min(\log \beta - r_i, 0)\}_{i=1}^n\right) - \log n \tag{54}$$

for the optimistic proposal, and similarly with $\min(r_i, \log \beta)$ for the safe proposal.

### C.1.3. CONFORMAL WEIGHTING

The calibration data $\mathcal{D}_{\mathrm{cal}}$ is drawn from a mixture of policies across previous rounds. To apply conformal prediction under this distribution shift, we weight each calibration point by the likelihood ratio between the candidate constrained policy and the calibration mixture:

$$\tilde{w}_i(\beta) = \frac{\pi^{(\beta)}(x_i)}{\pi_{\mathrm{mix}}(x_i)}, \quad x_i \in \mathcal{D}_{\mathrm{cal}}, \tag{55}$$

where $\pi_{\mathrm{mix}}$ is the mixture distribution over previous constrained policies, weighted by the number of calibration samples from each round.

For the test point, we use a conservative upper bound by taking the maximum weight over the proposal samples:

$$\tilde{w}_{\mathrm{test}}(\beta) = \max_{x \in \mathcal{D}_{\mathrm{prop}}} \frac{\pi^{(\beta)}(x)}{\pi_{\mathrm{mix}}(x)}. \tag{56}$$

We normalize all weights to sum to one:

$$w_i(\beta) = \frac{\tilde{w}_i(\beta)}{\sum_j \tilde{w}_j(\beta) + \tilde{w}_{\mathrm{test}}(\beta)}, \quad w_{\mathrm{test}}(\beta) = \frac{\tilde{w}_{\mathrm{test}}(\beta)}{\sum_j \tilde{w}_j(\beta) + \tilde{w}_{\mathrm{test}}(\beta)}. \tag{57}$$

Let $B$ denote the upper bound on the loss $\ell(x)$. The weighted risk estimate is

$$\hat{R}(\beta) = \sum_{i \in \mathcal{D}_{\mathrm{cal}}} w_i(\beta)\,\ell(x_i) + w_{\mathrm{test}}(\beta) \cdot B, \tag{58}$$

where the test point is conservatively assigned the worst-case loss $B$. We select $\hat{\beta}$ as the largest value in the grid $G$ such that $\hat{R}(\beta) \leq \alpha$.

## C.2. Sampling from the Constrained/Risk-Controlling Policy

In this section, we describe three approaches to sample from $\pi^{(\beta)}$: rejection sampling with either $\pi_0$ or $\pi_t$ as the proposal (Section C.2.1), rejection sampling with a mixture proposal (Section C.2.2), and Independence Metropolis-Hastings (Section C.2.3).

### C.2.1. REJECTION SAMPLING WITH SIMPLE PROPOSALS

We first review the classic accept-reject procedure. Let $\tilde{p}(x)$ be an unnormalized target density and $q(x)$ a normalized proposal such that

$$\frac{\tilde{p}(x)}{q(x)} \leq M \quad \forall x,$$

for some envelope constant $M < \infty$.[10] To sample from $p(x) = \tilde{p}(x)/\int \tilde{p}(x)dx$, we proceed as follows: draw $X \sim q$, draw $U \sim \mathrm{Unif}(0,1)$, and accept $X$ if $U \leq \frac{\tilde{p}(X)}{Mq(X)}$. The optimal (smallest) envelope constant is

$$M^\star = \sup_x \frac{\tilde{p}(x)}{q(x)}.$$

The marginal acceptance probability is

$$
\begin{aligned}
\mathbb{P}\left[U \leq \frac{\tilde{p}(X)}{Mq(X)}\right] &= \mathbb{E}\left[\mathbb{1}\left\{U \leq \frac{\tilde{p}(X)}{Mq(X)}\right\}\right] \\
&= \mathbb{E}\left[\mathbb{E}\left[\mathbb{1}\left\{U \leq \frac{\tilde{p}(X)}{Mq(X)}\right\}\Big|X\right]\right] \\
&= \mathbb{E}\left[\mathbb{P}\left[U \leq \frac{\tilde{p}(X)}{Mq(X)}\Big|X\right]\right] \\
&= \mathbb{E}_{X\sim q}\left[\frac{\tilde{p}(X)}{Mq(X)}\right] \\
&= \int_{x:q(x)>0} q(x)\frac{\tilde{p}(x)}{Mq(x)}dx \\
&= \frac{1}{M}\int_{x:q(x)>0} \tilde{p}(x)dx
\end{aligned}
\tag{59}
$$

so tighter $M$ leads to higher acceptance rates.

**Safe policy proposal.** Take $\tilde{p}(x) = \tilde{\pi}^{(\beta)}(x) = \min\{\pi_t(x), \beta\pi_{\mathrm{safe}}(x)\}$ and $q(x) = \pi_{\mathrm{safe}}(x)$. Then for all $x$,

$$\frac{\tilde{p}(x)}{q(x)} = \min\left\{\frac{\pi_t(x)}{\pi_{\mathrm{safe}}(x)}, \beta\right\} \leq \beta,$$

so we choose $M = \beta$. The pointwise acceptance probability then is

$$\frac{\tilde{p}(x)}{\beta\pi_{\mathrm{safe}}(x)} = \min\left\{\frac{\pi_t(x)}{\beta\pi_{\mathrm{safe}}(x)}, 1\right\},\tag{60}$$

or $\frac{\pi_t(x)}{\beta\pi_{\mathrm{safe}}(x)}$ in the unclipped region and 1 in the clipped region.

---

[10]Alternatively, we can view $Mq(x)$ as the unnormalized envelope proposal with normalizing constant $M$.

**Optimized policy proposal.** If we instead take $q(x) = \pi_t(x)$, then for all $x$,

$$\frac{\tilde{p}(x)}{q(x)} = \frac{\min\{\pi_t(x), \beta\pi_{\text{safe}}(x)\}}{\pi_t(x)} = \min\left\{1, \frac{\beta\pi_{\text{safe}}(x)}{\pi_t(x)}\right\} \le 1,$$

so we may take $M = 1$. The pointwise acceptance probability is

$$\frac{\tilde{p}(x)}{\pi_t(x)} = \min\left\{1, \frac{\beta\pi_{\text{safe}}(x)}{\pi_t(x)}\right\}, \tag{61}$$

which is 1 in the unclipped region and $\frac{\beta\pi_{\text{safe}}(x)}{\pi_t(x)}$ in the clipped region.

Intuitively, the relative efficiency of the two proposals depends on the constraint parameter $\beta$ controlling how much the optimized policy is truncated toward the safe policy. When $\beta$ is small, the constrained target resembles a rescaled version of $\pi_{\text{safe}}$ over most of its support. Proposing from $\pi_{\text{safe}}$ tends to be efficient in this regime, because most points satisfy $\pi_t(x)/\pi_{\text{safe}}(x) \ll \beta$ and the acceptance probability in Equation 60, while small, varies smoothly over $x$. In contrast, proposing from $\pi_t$ can be inefficient with small $\beta$, since $\pi_t$ may place substantial mass in regions that are heavily downweighted by the constraint, leading to frequent rejections.

When $\beta$ is large, the constrained target resembles $\pi_t$. As long as $\pi_t(x)/\pi_{\text{safe}}(x) \lesssim \beta$ for most of the support of $\pi_t$, then using $\pi_t$ as the proposal tends to be more efficient in this regime, as the acceptance probability in Equation 61 is frequently close to 1. If, however, $\pi_t$ concentrates mass in regions where $\pi_t(x)/\pi_{\text{safe}}(x)$ is extremely large, this proposal can still perform poorly.

At intermediate values of $\beta$, neither proposal dominates. The constrained target may allocate comparable mass from $\pi_{\text{safe}}$ and $\pi_t$. Both proposals can suffer from mismatches with different parts of the target, motivating the use of mixture proposals that adaptively trade off between $\pi_{\text{safe}}$ and $\pi_t$.

### C.2.2. REJECTION SAMPLING WITH MIXTURE PROPOSAL

Let the proposal be the mixture

$$q_w(x) = w\pi_{\text{safe}}(x) + (1 - w)\pi_t(x), \quad w \in [0, 1].$$

The tight envelope constant is

$$M^\star(w) = \sup_x \frac{\tilde{p}(x)}{q_w(x)} = \sup_x \frac{\min\{\pi_t(x), \beta\pi_{\text{safe}}(x)\}}{w\pi_{\text{safe}}(x) + (1 - w)\pi_t(x)}.$$

Given any $M \ge M^\star(w)$, the pointwise acceptance probability is

$$\alpha_w(x) := \frac{\tilde{p}(x)}{Mq_w(x)} = \frac{\min\{\pi_t(x), \beta\pi_{\text{safe}}(x)\}}{M\left(w\pi_{\text{safe}}(x) + (1 - w)\pi_t(x)\right)}. \tag{62}$$

Choosing smaller $M$ (i.e., closer to $M^\star$) yields higher acceptance rates, as derived in Equation 59. In practice, we may choose $M$ by estimating $M^\star(w)$ from a set of points $\mathcal{S}$ covering the support of $q_w$, where $\mathcal{S}$ is obtained by drawing samples from $q_w$ or some appropriate fraction of samples from each component to cover the tails. Taking the empirical maximum

$$\max_{x \in \mathcal{S}} \frac{\tilde{p}(x)}{q_w(x)}$$

then yields an estimate of $M^\star(w)$.

We now describe a few simple methods for choosing $w$.

**Joint grid search with $M$.** We can iterate over a grid of $w$ and choose $w, M$ jointly. For each candidate value of $w$, estimate $M^\star(w)$ using the procedure described above and choose $w$ that minimizes this estimate. Note that the support points $\mathcal{S}$ used to estimate $M^\star(\cdot)$ can be reused for different values of $w$.

**Estimated overlap coefficients.** The *overlap coefficient* between two density functions $p_1, p_2$ is defined as

$$\text{OVL}(p_1, p_2) := \int \min\{p_1(x), p_2(x)\} \; dx = \int \min\left\{\frac{p_1(x)}{p_2(x)}, 1\right\} p_2(x) \; dx \quad \in [0, 1].$$

It is related to the total variation (TV) distance as $\text{OVL}(p_1, p_2) = 1 - \text{TV}(p_1, p_2)$.

Letting $p_1 = p = \pi^{(\beta)}$, our normalized target, we can estimate $\text{OVL}(\pi^{(\beta)}, \pi_{\text{safe}})$ using samples $x_1, \ldots, x_n$ drawn from $\pi_{\text{safe}}$:

$$\widehat{\text{OVL}}(\pi^{(\beta)}, \pi_{\text{safe}}) = \frac{1}{n} \sum_{i=1}^{n} \min\left\{\frac{\pi^{(\beta)}(x_i)}{\pi_{\text{safe}}(x_i)}, 1\right\} = \frac{1}{n} \sum_{i=1}^{n} \min\left\{\frac{\pi_t(x_i)}{\psi(\beta)\pi_{\text{safe}}(x_i)}, \frac{\beta}{\psi(\beta)}, 1\right\}, \tag{63}$$

and similarly $\widehat{\text{OVL}}(\pi^{(\beta)}, \pi_t)$ using samples drawn from $\pi_t$. The normalizing constant $\psi(\beta)$ is estimated with importance-weighted MC integration as described in Section C.1.2. A simple heuristic is then to set

$$w = \frac{\widehat{\text{OVL}}(\pi^{(\beta)}, \pi_{\text{safe}})}{\widehat{\text{OVL}}(\pi^{(\beta)}, \pi_{\text{safe}}) + \widehat{\text{OVL}}(\pi^{(\beta)}, \pi_t)}. \tag{64}$$

This chooses the mixture weights proportional to each component's estimated overlap with the target, biasing the proposal toward whichever component more closely matches the constrained policy, while retaining support from both.

**Adaptive update.** Rather than fixing $w$, we can adapt it online in the direction of a desirable sampling diagnostic. For example, we might define an update based on the observed acceptance rates of proposals drawn from each component, so that $w$ shifts toward the component that yields higher acceptance.

### C.2.3. INDEPENDENCE METROPOLIS-HASTINGS SAMPLING

While rejection sampling yields exact samples from the target distribution, it requires finding a finite global bound $M$ satisfying $\tilde{p}(x)/q(x) \leq M$ for all $x$. In many settings, especially in high dimensions, this bound is too loose to be practical or may not exist at all. The Metropolis-Hastings algorithm (Metropolis et al., 1953; Hastings, 1970) relaxes this requirement. In particular, here we discuss a simple variant called the independence Metropolis Hastings (IMH) algorithm, where the proposal does not depend on the current state (Tierney, 1994).

Given an unnormalized target $\tilde{p}(x)$ and a normalized proposal $q(x)$, we initialize the Markov chain at $x_0$ and the IMH algorithm evolves it as follows. From the current state $x_t$, draw a candidate state $x' \sim q$, compute the acceptance probability

$$a(x', x_t) = \min\left\{1, \frac{r(x')}{r(x_t)}\right\}, \quad r(x) := \frac{\tilde{p}(x)}{q(x)},$$

draw $U \sim \text{Unif}(0, 1)$, and accept $x'$ (i.e., set $x_{t+1} = x'$) if $U \leq a(x', x_t)$ otherwise copy the old state $x_{t+1} = x_t$. It can be shown that the invariant distribution of the resulting Markov chain $x_0, x_1, x_2 \ldots$ is the target $p(x)$. Intuitively, IMH compares each proposal only to the current state, whereas rejection sampling compares each proposal to a global worst case.

# D. Medical QA Implementation Details and Additional Results

This appendix provides implementation details for our medical question-answering experiments, connecting the general conformal risk control framework to the specific problem of controlling LLM factuality. We describe the problem setup and notation (Section D.1), prior approaches to conformal factuality (Section D.2), loss functions for factuality control including the non-monotonic FDR loss (Section D.3), experimental setup (Section D.4), and hyperparameters (Section D.5).

## D.1. Problem Setup and Notation

Following Mohri & Hashimoto (2024) and Cherian et al. (2024), we formalize the natural language factuality setting as follows. Each data sample $Z_i$ is a prompt-response-claim-annotation tuple:

$$Z_i := (X_i, X_i', \boldsymbol{C}_i, \boldsymbol{Y}_i), \tag{65}$$

where $X_i$ is the prompt (question), $X_i'$ is the full LLM response, $\boldsymbol{C}_i = (C_{i1}, \ldots, C_{ik_i})$ is the vector of atomic subclaims extracted from $X_i'$, and $\boldsymbol{Y}_i = (Y_{i1}, \ldots, Y_{ik_i})$ contains binary annotations indicating whether each subclaim is factually correct ($Y_{ij} = 1$) or incorrect ($Y_{ij} = 0$).

For each subclaim $C_{ij}$, let $p(C_{ij} \mid X_i) \in [0, 1]$ denote a prompt-conditional confidence score estimating the likelihood that the subclaim is factual. Higher scores indicate greater confidence in factuality.

## D.2. Prior Approaches to Conformal Factuality

**Mohri & Hashimoto (2024).** Mohri & Hashimoto (2024) achieve conformal factuality guarantees by filtering subclaims based on a threshold $\tau$ determined by split conformal prediction. A subclaim $C_{ij}$ is included in the filtered response if $p(C_{ij} \mid X_i) \geq \tau$. The corresponding non-conformity score for each response is the smallest threshold such that all included subclaims are factual:

$$s(Z_i) = \inf\left\{\tau : Y_{ij} = 1 \text{ for all } j \text{ with } p(C_{ij} \mid X_i) \geq \tau\right\}. \tag{66}$$

The threshold $\hat{\tau}$ is then selected as the $(1 - \alpha)$-quantile of the calibration scores:

$$\hat{\tau} = \inf\left\{\tau : \frac{1}{n}\sum_{i=1}^{n} \mathbb{1}\{s(Z_i) \leq \tau\} \geq \frac{\lceil (n+1)(1-\alpha) \rceil}{n}\right\}. \tag{67}$$

**Cherian et al. (2024).** Cherian et al. (2024) extend this approach using conditional calibration and multicalibration to provide guarantees that hold across different subgroups of prompts, addressing the limitation that marginal guarantees may not hold uniformly across topics or question types.

## D.3. Loss Functions for Factuality Control

**Binary indicator loss (prior work).** The loss function implicitly used by Mohri & Hashimoto (2024) and Cherian et al. (2024) is the binary indicator for whether any included subclaim is incorrect:

$$L_i^{\text{binary}}(\tau) = \mathbb{1}\left\{\exists j : p(C_{ij} \mid X_i) \geq \tau \text{ and } Y_{ij} = 0\right\}. \tag{68}$$

Controlling this loss at level $\alpha$ guarantees $\mathbb{P}(\text{any included claim is false}) \leq \alpha$. This loss is monotonically non-increasing in $\tau$ by construction: as the threshold increases, fewer claims are included, so the probability of including a false claim can only decrease.

**False discovery rate loss (this work).** We instead use a more granular loss: the false discovery rate (FDR), *i.e.*, the fraction of included subclaims that are false:

$$L_i(\tau) = \underbrace{\frac{\sum_{j=1}^{k_i}(1 - Y_{ij}) \cdot \mathbb{1}\{p(C_{ij} \mid X_i) \geq \tau\}}{\sum_{j=1}^{k_i} \mathbb{1}\{p(C_{ij} \mid X_i) \geq \tau\}}}_{\text{FDR}_i(\tau) = 1 - \text{Precision}_i(\tau)}, \tag{69}$$

with $L_i(\tau) = 0$ when no claims are included. This is precisely $1 - \text{Precision}_i(\tau)$, where precision is the fraction of included claims that are true. Controlling this loss at level $\alpha$ ensures:

$$\mathbb{E}[\text{FDR}_i(\tau)] \leq \alpha \quad \Longleftrightarrow \quad \mathbb{E}[\text{Precision}_i(\tau)] \geq 1 - \alpha. \tag{70}$$

**Why FDR control requires generalized CRC.** Unlike the binary loss, the FDR loss in (69) is not monotonic in $\tau$. As $\tau$ increases, both the numerator (false claims included) and denominator (total claims included) decrease, but not necessarily at the same rate. This non-monotonicity means standard CRC (Angelopoulos et al., 2024) does not apply directly, motivating our generalized CRC approach.

For methods requiring monotonicity, one can use the monotonized loss:

$$\tilde{L}_i(\tau) = \sup_{\tau' \geq \tau} L_i(\tau'), \tag{71}$$

which upper-bounds the FDR and is monotonically non-increasing by construction. However, this monotonization can be overly conservative.

**Reward metric: true claim recall.** While we control the FDR (our loss), we measure the recall of true claims as our reward metric:

$$\text{Recall}_i(\tau) = \frac{\sum_{j:Y_{ij}=1} \mathbb{1}\{p(C_{ij} \mid X_i) \geq \tau\}}{\sum_{j=1}^{k_i} Y_{ij}}, \tag{72}$$

*i.e.*, the fraction of true subclaims that are retained after filtering. For a given FDR target $\alpha$, methods that achieve higher recall preserve more useful information while satisfying the same safety constraint.

### D.4. Experimental Setup

Our experimental setup closely follows Cherian et al. (2024), with the key modification of controlling the FDR loss (69) and measuring recall (72) as a reward metric.

#### D.4.1. DATASET

We use the **MedLFQA** (Medical Long-Form Question Answering) dataset from Jeong et al. (2024), which aggregates four established medical QA benchmarks:

- **HealthSearchQA** ($n = 3047$): Consumer health questions

- **K-QA** ($n = 1077$): Knowledge-intensive medical questions, with a subset having gold-standard physician responses (K-QA Golden, $n = 201$) and the remainder having LLM-generated responses (K-QA Silver)

- **LiveQA** ($n = 100$): Real consumer health questions from the National Library of Medicine

- **MedicationQA** ($n = 627$): Questions about medications and drug interactions

Each prompt is accompanied by either an LLM or human-generated reference response, which serves as ground truth for evaluating factual accuracy.

#### D.4.2. DATA GENERATION PIPELINE

We use the annotated dataset from Cherian et al. (2024), which was constructed through a three-stage pipeline:

1. **Response generation**: GPT-3.5-Turbo was queried with each medical question to obtain a candidate response $X_i'$.

2. **Subclaim extraction**: GPT-4o parsed each response into atomic subclaims $C_i = (C_{i1}, \ldots, C_{ik_i})$.

3. **Factuality annotation**: GPT-3.5-Turbo evaluated whether each subclaim is substantiated by the reference response, yielding binary annotations $Y_{ij} \in \{0, 1\}$.

For subclaim scoring, we use **self-evaluation scores** $p(C_{ij} \mid X_i)$, where the LLM is prompted to assess its own confidence in each claim.

D.4.3. METHODS COMPARED

We compare three conformal risk control methods:

- **gCRC (proposed)**: Generalized conformal risk control as described in Section 4, which searches thresholds from safest to most aggressive and stops when the augmented empirical risk exceeds $\alpha$.

- **Monotonized-losses CRC** (Angelopoulos et al., 2024; Mohri & Hashimoto, 2024): Applies standard CRC to the monotonized loss $\tilde{L}_i(\tau) = \sup_{\tau' \geq \tau} L_i(\tau')$, which upper-bounds the FDR.

- **Learn Then Test (LTT)** (Angelopoulos et al., 2025): A multiple testing approach using Hoeffding-Bentkus $p$-values to identify valid thresholds with family-wise error rate control. Unlike CRC methods, LTT does not add a conservative test-point loss term.

D.4.4. EXPERIMENTAL PROTOCOL

- **Calibration/test split**: 70%/30% random split of the dataset

- **Risk levels**: $\alpha \in \{0.005, 0.01, 0.015, \ldots, 0.10\}$ (20 values)

- **Threshold grid**: 200 candidate thresholds sampled from the empirical distribution of subclaim scores

- **Random seeds**: Results averaged over 10 independent train/test splits

- **Score jittering**: Uniform noise ($\sim 10^{-8}$) added to break ties

## D.5. Hyperparameters

Table 2 summarizes the experimental configuration.

*Table 2.* Hyperparameters for medical QA experiments.

| Parameter | Symbol | Value |
|---|---|---|
| Calibration fraction | – | 0.7 |
| Number of trials | – | 10 |
| Threshold grid size | $|\Lambda|$ | 200 |
| Risk levels tested | $\alpha$ | $\{0.005, 0.01, \ldots, 0.10\}$ |
| Score jitter magnitude | – | $10^{-8}$ |
| Response LLM | – | GPT-3.5-Turbo |
| Subclaim extraction LLM | – | GPT-4o |
| Annotation LLM | – | GPT-3.5-Turbo |
| Subclaim scoring method | – | Self-evaluation |

## D.6. Additional Medical QA Results

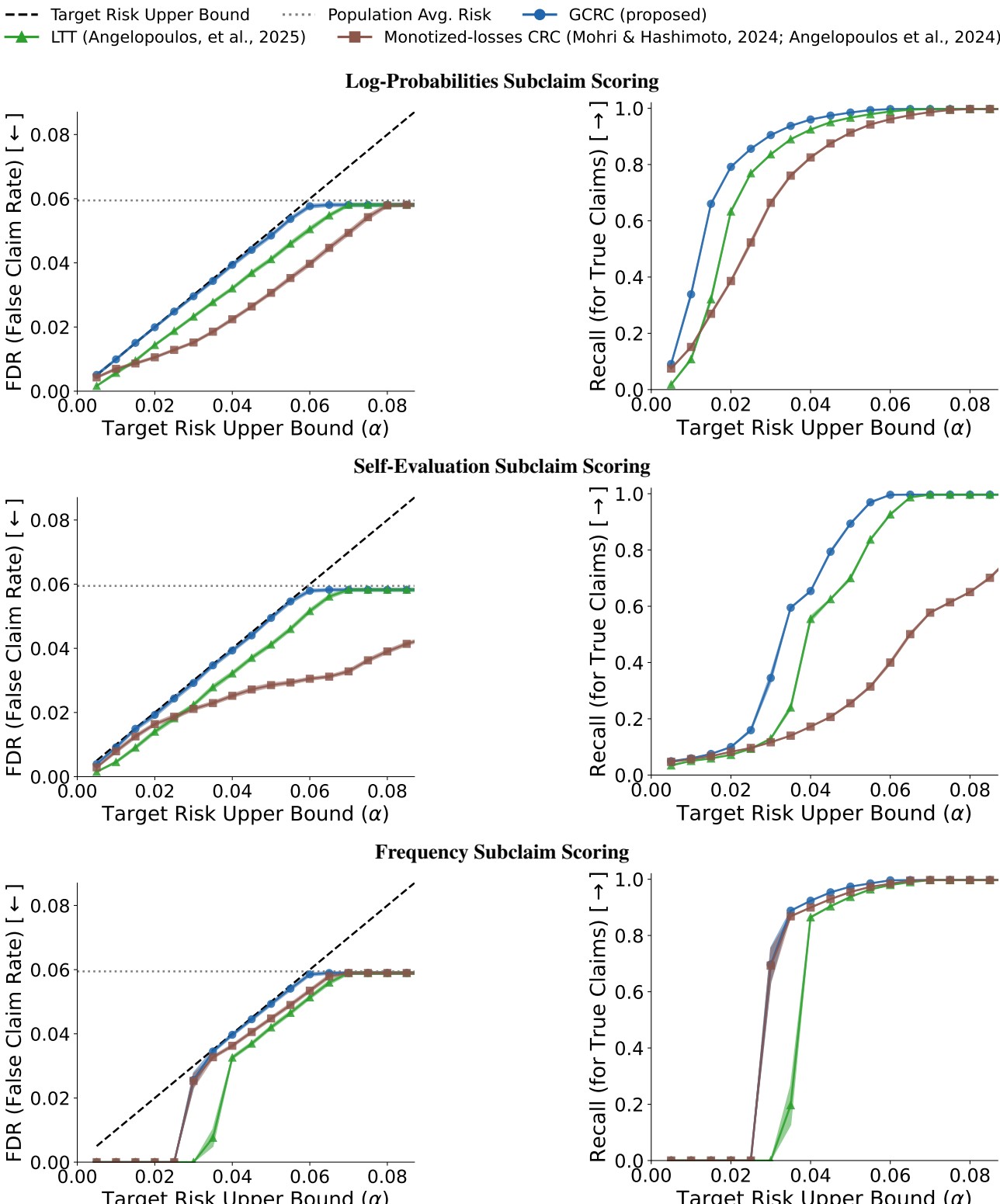

*Figure 8.* Medical QA factuality control results on MedLFQA on three subclaim scoring methods: log-probabilities **(top row)**, self-evaluation **(middle row)**, and frequency **(bottom row)**. **Left column:** Empirical FDR (fraction of retained claims that are false) vs. target risk level $\alpha$. The black dashed line indicates $y = x$; valid methods should fall at or below this line. **Right column:** Recall (fraction of true claims retained) vs. $\alpha$. gCRC tightly controls FDR at the target level while achieving superior recall to baselines. Results averaged over 25 random splits. Error regions are standard errors.

# E. Constrained Tabular Active Learning Implementation Details and Additional Results

This appendix provides implementation details for the constrained active learning experiments in Section 5.2. We describe the datasets used for evaluation (Section E.1), the construction of synthetic feasibility constraints based on principal component analysis (Section E.2), the active learning setup including surrogate model and acquisition policy (Section E.3), and hyperparameter settings (Section E.4).

## E.1. Datasets

We evaluate on three regression datasets:

- **Robot Arm Kinematics** (University of Toronto, 1996) (8192 samples, 8 features): Predict end-effector distance from joint angles of an 8-link all-revolute robot arm. We use the kin8nm variant from the DELVE repository, a highly non-linear regression task with moderate noise.

- **Airfoil Self-Noise** (Brooks et al., 1989) (1503 samples, 5 features): Predict sound pressure level from aerodynamic and geometric properties of NACA 0012 airfoil blade sections tested in an anechoic wind tunnel. We apply log transforms to frequency and suction-side displacement thickness following standard practice.

- **Medical Expenditure Panel Survey (Medical Utilization or MEPS)** (Ezzati-Rice et al., 2008) (33005 samples, 107 features): Large-scale survey dataset on medical utilization and costs of health care and health insurance coverage. Surveys are of individuals and families, their medical providers, and employers across the United States. We pre-process the data (for instance constructing dummy/indicator covariates) as described in Barber et al. (2021). Data available at https://meps.ahrq.gov/mepsweb/data_stats/download_data_files_detail.jsp?cboPufNumber=HC-192

## E.2. Feasibility Constraint Construction

Our datasets do not include recorded feasibility constraints, so we construct synthetic constraints based on the first principal component of the covariate matrix. The first principal component captures the dominant pattern of covariation in the data. Records with high PC1 values follow this typical covariation pattern and are likely feasible, while records with low PC1 values deviate from this pattern. Their covariates vary in an atypical or "weird" way that we treat as risky. This captures the intuition that unusual configurations are more likely to violate constraints.

We define the feasibility indicator for each record as follows:

1. **PCA projection**: Compute the first principal component of the full covariate matrix $X \in \mathbb{R}^{n \times d}$ and project all records onto this axis, yielding $z_i \in \mathbb{R}$ for each record $i$.

2. **Exponential tilting**: Apply min-max normalization to $\{z_i\}$ and compute exponentially-tilted values $\tilde{z}_i = \exp(\gamma \cdot z_i^{\text{norm}})$, where $\gamma > 0$ is the `initial_sampling_bias` parameter.

3. **Feasibility probability**: Convert relative ranks of $\{\tilde{z}_i\}$ to feasibility probabilities via a logistic CDF:

$$p_i^{\text{feasible}} = \sigma\Big(\text{rank}(\tilde{z}_i)/n; \ \mu, s\Big), \tag{73}$$

where $\sigma(\cdot; \mu, s)$ is the logistic CDF with location $\mu = \min(2.5\alpha, 0.98)$, scale $s = 0.1$, and $\alpha$ is the target risk level for CPC. We tie the logistic location to $\alpha$ so that the proportion of infeasible records in the pool scales with the target risk level, ensuring the constraint remains binding across different choices of $\alpha$. For our experiments with $\alpha = 0.2$, this yields $\mu = 0.5$.

4. **Bernoulli sampling**: Sample feasibility indicators $F_i \sim \text{Bernoulli}(p_i^{\text{feasible}})$.

The initial training data is sampled with the same exponential tilting toward high PC1 values (Section E.3), so the learner begins in a high-feasibility region where covariates follow the dominant covariation pattern. The pool contains many records from the opposite end of the PC1 spectrum, atypical configurations with low feasibility probability that a naive uncertainty-based acquisition policy would preferentially select.

## E.3. Active Learning Setup

**Initial data.** We hold out 20% of each dataset as a fixed test set for evaluating MSE. From the remaining records, we sample $n_{\text{initial}}$ points with exponential tilting toward high PC1 values (using the same bias parameter $\gamma$ from Section E.2), then split these into training (80%) and calibration (20%) sets. This biased initialization places the learner in the high-feasibility region where covariates follow typical covariation patterns, reflecting a scenario where the incumbent safe policy operates in well-understood regimes.

**Surrogate model.** We fit a Gaussian Process (GP) regression model with a DotProduct + WhiteKernel covariance function using the `scikit-learn` package (Pedregosa et al., 2011):

$$k(x, x') = \sigma_0^2 + x^\top x' + \sigma_n^2 \cdot \mathbf{1}[x = x'], \tag{74}$$

where $\sigma_0^2$ is the signal variance and $\sigma_n^2$ is the noise variance.

**Acquisition policy.** At each iteration, we define an acquisition distribution over pool records via exponential tilting toward posterior variance. The action $a \in \mathcal{A}_t$ is an index into the pool of unlabeled records at iteration $t$, and the acquisition policy is

$$\pi(a) \propto \exp\left(\lambda \cdot \frac{\hat{\sigma}_a^2}{\max_{j \in \mathcal{A}_t} \hat{\sigma}_j^2 - \min_{j \in \mathcal{A}_t} \hat{\sigma}_j^2}\right), \tag{75}$$

where $\hat{\sigma}_a^2$ is the posterior variance of the latent function $f$ (not the noisy labels $y$) at the covariates for record $a$, and $\lambda > 0$ controls the degree of uncertainty-seeking behavior. This policy preferentially selects records where the model is uncertain, which in our setup corresponds to regions with atypical covariation patterns (and thus potentially infeasible).

**Conformal policy control.** Before sampling, we apply CPC to constrain the acquisition policy relative to the initial source distribution (the biased sampling distribution used for initialization). This clips the likelihood ratio between the acquisition policy and source policy, as described in Section 4.2, to control the risk of selecting infeasible records.

**Sequential updates.** At each iteration, we sample one record from the (constrained) acquisition policy with replacement (the pool remains unchanged), observe a noisy version of its label (Gaussian noise with magnitude 0.05) and its feasibility indicator, and add the observation to either the training or calibration set with equal probability. The GP is then refit on the updated training set. We repeat this process for $T$ iterations.

## E.4. Hyperparameters

Table 3 summarizes the hyperparameters used in our experiments.

*Table 3.* Hyperparameters for constrained active learning experiments.

| Parameter | Symbol | Value |
|---|---|---|
| Initial pool size | $n_{\text{initial}}$ | 48 |
| Initial train/cal split | – | 80%/20% |
| Acquisition iterations | $T$ | 10 |
| New point train probability | – | 0.5 |
| Sampling bias | $\gamma$ | 1.0 |
| Acquisition temperature | $\lambda$ | 10.0 |
| GP signal variance | $\sigma_0^2$ | 1.0 |
| GP noise variance | $\sigma_n^2$ | 1.0 |
| Observation noise magnitude | – | 0.05 |
| Target risk level | $\alpha$ | 0.2 |

# F. Constrained Black-Box Sequence Optimization Implementation Details and Results

This appendix provides implementation details for the constrained black-box sequence optimization experiments in Section 5.3. We describe the Ehrlich test functions used for evaluation (Section F.1), the procedure for training the safe policy via supervised fine-tuning on genetic algorithm data (Section F.2), the policy improvement procedure via DPO (Section F.3), and supplementary results across additional risk levels (Figure 9).

## F.1. Ehrlich Test Functions

We evaluate CPC on Ehrlich functions (Chen et al., 2025), a class of synthetic test functions designed to capture the geometric structure of biomolecular sequence optimization problems such as antibody affinity maturation. An Ehrlich function $f : \mathcal{V}^L \to (-\infty, 1]$ maps sequences of length $L$ over a vocabulary $\mathcal{V}$ to a score indicating how well the sequence satisfies a collection of $c$ gapped motifs $\mathbf{m}^{(i)}$ with spacing $\mathbf{s}^{(i)}$. Formally,

$$f(\mathbf{x}) = \begin{cases} \prod_{i=1}^{c} g \circ h_q(\mathbf{x}, \mathbf{m}^{(i)}, \mathbf{s}^{(i)}) & \text{if } \mathbf{x} \in \mathcal{F} \\ -\infty & \text{otherwise} \end{cases}, \tag{76}$$

where $h_q$ measures motif satisfaction with precision $q$, $g$ is a response function capturing epistatic effects, and $\mathcal{F}$ is a feasibility set defined by a discrete Markov process with certain infeasible transitions (*i.e.* some bigrams have zero probability mass). Sequences outside $\mathcal{F}$ are assigned $-\infty$, modeling expression constraints in real biophysical assays.

We follow the naming convention **Ehr($|\mathcal{V}|$, $L$)-$c$-$k$-$q$** from Chen et al. (2025), where $k$ is the motif length and $q$ controls signal sparsity. Ehrlich functions are procedurally generated, have adjustable difficulty, and are provably solvable, making them well-suited for benchmarking constrained optimization algorithms. In our experiments, we use **Ehr(32, 32)-4-4-4**: sequences of length $L = 32$ over a vocabulary of size $|\mathcal{V}| = 32$, with $c = 4$ motifs of length $k = 4$ and quantization precision $q = 4$.

## F.2. Training the Safe Policy

Following Chen et al. (2025), we initialize the optimization loop with a data package $\mathcal{D}^{(0)}$ collected from a genetic algorithm (GA) pre-solver. Starting from a single feasible seed sequence $\mathbf{x}_0$ sampled from the discrete Markov process, we run $n_0 = 10$ rounds of the GA with 5000 particles per iteration. This produces approximately 50K scored sequences that provide initial coverage of the feasible region $\mathcal{F}$. The GA uses mutation probability $p_m = 0.05$ and survival quantile $\alpha = 0.5$.

From the GA history, we construct the initial supervised fine-tuning (SFT) dataset by identifying improving pairs: for each sequence $\mathbf{x}$ in the history, we search for sequences $\mathbf{x}'$ within edit distance $\delta$ such that $f(\mathbf{x}') > f(\mathbf{x})$. These $(\mathbf{x}, \mathbf{x}')$ pairs form instruction-response examples for training the safe policy $\pi_{\theta_0}$, which we initialize from a Pythia 14M-parameter decoder-only transformer (Biderman et al., 2023). See Table 4 for hyperparameter settings.

*Table 4.* Hyperparameters for training the safe policy via SFT.

| Hyperparameter | Value |
| --- | --- |
| Base model | Pythia 14M |
| Learning rate | $10^{-6}$ |
| Training epochs | 10 |
| Batch size | 32 |
| Edit distance threshold ($\delta$) | $0.3 \cdot L$ |

## F.3. Improving the Safe Policy

To improve the safe policy, we take a fixed set of seed sequences from the GA optimizer history and sample actions from $\pi_{\theta_0}$. From the resulting data, we construct DPO preference triplets of the form $(\mathbf{x}, \mathbf{x}^+, \mathbf{x}^-)$, where $\mathbf{x}^+$ is preferred over $\mathbf{x}^-$ based on the oracle scores. We initialize the improved policy $\pi_{\theta_t}$ at the safe policy weights and train using the DPO loss (Rafailov et al., 2023). See Algorithm 2 for pseudocode of a single round of CPC.

We chose DPO as our policy improvement method because Chen et al. (2025) demonstrated that it struggles with poor

**Algorithm 2** Conformal Policy Control

**Require:** Safe policy $\pi_0$; context draws $\{X_i\}_{i=1}^n \sim p(X)$; target risk level $\alpha$

    **Phase 1: Safe Policy Data Collection**

    Draw actions $A_i \sim \pi_0(\cdot \mid X_i)$ for $i = 1, \ldots, n$

    Observe rewards $R_i = r(X_i, A_i)$ and losses $L_i = \ell(X_i, A_i)$

    Split data: $\mathcal{D}_{\text{train}}, \mathcal{D}_{\text{cal}} \leftarrow \text{RANDOMSPLIT}(\{(X_i, A_i, R_i, L_i)\})$

    **Phase 2: Policy Improvement**

    $\pi_t \leftarrow \text{IMPROVEPOLICY}(\pi_0, \mathcal{D}_{\text{train}})$

    **Phase 3: Calibration**

    $\hat{\beta} \leftarrow \text{CALIBRATEBETA}(\pi_0, \pi_t, \mathcal{D}_{\text{cal}}, \alpha)$ (Algorithm 1)

    **Phase 4: Constrained Deployment**

    Draw improved actions $A_i' \sim \pi_t^{(\hat{\beta})}(\cdot \mid X_i)$ via accept-reject for $i = 1, \ldots, n$

    **Return:** $\{A_i'\}_{i=1}^n$

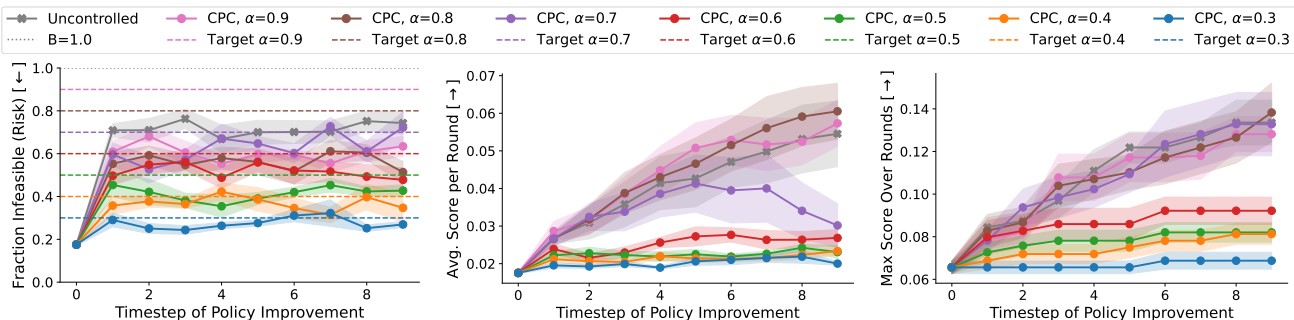

*Figure 9.* Supplementary experimental results applying CPC to constrained black-box optimization of token sequences scored with Ehrlich functions. In the **left** panel, we show the average feasibility rate round-over-round. In the **center**, we show the average objective value of solutions sampled from the DPO-tuned policy, and in the **right**, the best solution found so far at each round. CPC allows us to easily modify the constraint violation risk by directly manipulating $\alpha$. We find that moderate risk control may actually stabilize the algorithm and lead to better overall performance since fewer samples are wasted to infeasibility constraints.

feasibility rates compared to other fine-tuning approaches, making it a challenging test case for our risk control procedure. See Table 5 for hyperparameter settings.

*Table 5.* Hyperparameters for policy improvement via DPO.

| Hyperparameter | Value |
| --- | --- |
| Learning rate | $10^{-7}$ |
| DPO $\beta$ | 0.4 |
| Training epochs per round | 1 |
| Batch size | 32 |
| Number of DPO rounds | 10 |

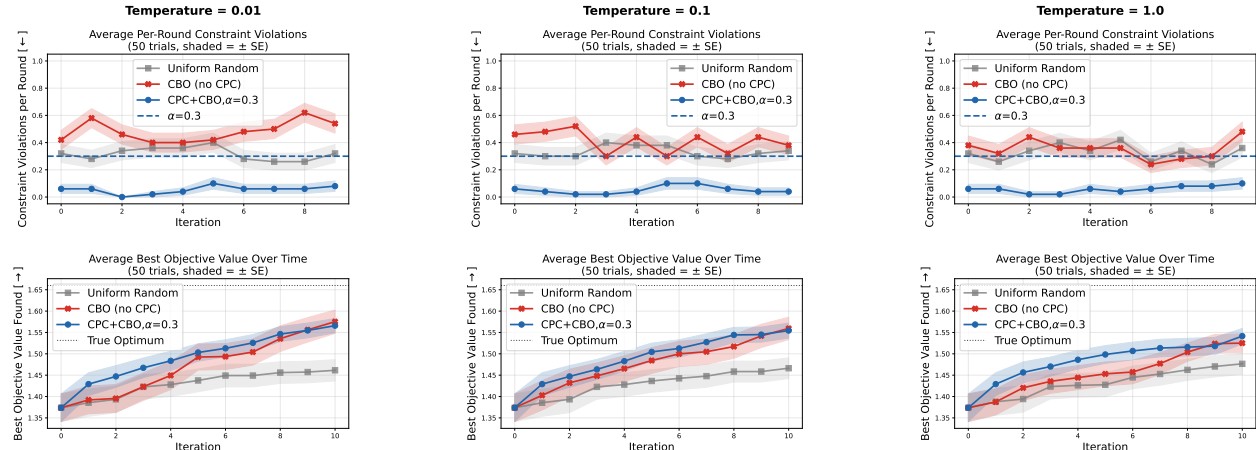

*Figure 10.* Empirical evaluation of a recent Constrained Bayesian Optimization (CBO) method (Tian et al., 2024) with CPC (blue) and without CPC (red) over various acquisition sampling temperature settings. Results: Across temperature settings of 0.01 (left column), 0.1 (middle column), and 1.0 (right column), the CPC-controlled CBO method controls the constraint violation rate well below the target level of alpha=0.3 (blue, top row). In contrast, the CBO baseline has much higher constraint violation rates above alpha=0.3; the baseline's violation rate appears to slightly decrease with increased sampling temperatures (red, top row), but not as a function of alpha, so hyperparameter tuning would be required to find an acceptable temperature value. The CPC+CBO method attains slightly improved reward over the CBO baseline without CPC for all temperature values (bottom row), possibly due to wasting fewer evaluations on infeasible samples.

## G. Comparison to Conservative Optimization on Constrained Bayesian Optimization Task

Whereas our previous experiments compared to conformal baselines and other methods with distribution-free guarantees, in Figure G we report an additional comparison to a standard conservative optimization baseline in a Constrained Bayesian Optimization (CBO) setting. Standard conservative optimization baselines have two key limitations relative to CPC: First, they typically assume a particular model class, whereas CPC does not; Second, they typically cannot translate a user's declared risk tolerance, $\alpha$, into imperative instructions for the algorithm, such as a specific divergence penalty or constraint parameter that achieves risk control. In practice, these limitations mean that extensive hyperparameter tuning is often required for such standard methods to meet desired operating constraints. These experiments illustrate that CPC can circumvent this hyperparameter tuning challenge. Moreover, CPC can be viewed as complementing rather than competing with standard safe exploration methods, since any policy for exploration can be used as the "optimized policy" in CPC.

**Experiment Details:** Figure G compares CPC-controlled CBO to the CBO baseline (Balandat et al., 2020) on a synthetic constrained Bayesian optimization task with unknown binary feasibility constraints. We evaluate across temperature settings for the CBO's probabilistic acquisition sampling to simulate an attempt at hyperparameter-tuning the CBO baseline. The data are 2-dimensional and continuous, and the unknown objective/reward function is the synthetic Townsend function from Townsend (2014) with unknown binary feasibility constraints. We use a Gaussian process from BoTorch (Balandat et al., 2020) to model the objective function and a classifier to model the unknown constraint. The CBO method's acquisition is done with probabilistic sampling from temperature-scaling on the qLogExpectedImprovement function in BoTorch (Balandat et al., 2020). The safe policy for CPC is a mean-zero Gaussian with standard deviation set to 1/6th of search space range for 99.7% coverage. A uniform random baseline is provided for reference (gray). Our implementation of the CBO baseline (Tian et al., 2024) was based on the following BoTorch notebook: `https://botorch.org/docs/notebooks_community/clf_constrained_bo/`

**Results:** Across temperature settings of 0.01 (left column), 0.1 (middle column), and 1.0 (right column), the CPC-controlled CBO method controls the constraint violation rate well below the target level of alpha=0.3 (blue, top row). In contrast, the CBO baseline has much higher constraint violation rates above alpha=0.3; the baseline's violation rate appears to slightly decrease with increased sampling temperatures (red, top row), but not as a function of alpha, so further hyperparameter tuning would be required to find an acceptable temperature value (if any exists). Notably, the CPC+CBO method even attains improved reward over the CBO baseline without CPC for all temperature values (bottom row), possibly due to wasting fewer evaluations on infeasible samples.

