# OpenReview forum: "Conformal Policy Control"
_ICML.cc/2026/Conference — ICML 2026 spotlight_

### Official Review · Reviewer_i3pZ · 2026-03-10

**Soundness:** 3
**Presentation:** 2
**Significance:** 3
**Originality:** 4
**Overall Recommendation:** 5
**Confidence:** 3

**Summary:**

This paper studies a policy learning problem that seeks to maximize the expected reward while satisfying a safety constraint requiring the expected loss to remain below a prespecified threshold. To achieve this, the author first extended the standard conformal risk control (CRC) framework to handle non-monotonic losses by selecting the empirical solution as the smallest hyperparameter whose empirical risk remains controlled for all larger values. Then, the decision policy is specified as a mixture of a "safe" policy and a "good" policy, where the mixture is controlled by the likelihood ratio that is calibrated from the proposed generalized CRC procedure, and sampling from this mixture is done via rejection sampling.

**Compliance With Llm Reviewing Policy:**

Affirmed.

**Final Justification:**

I thank the authors for the response. My concerns have been addressed, and I hope the author can update the contents of the paper to reflect the discussed changes. I will keep my original score since it's already favorable towards acceptance.

**Key Questions For Authors:**

# Additional Comments

- In Theorem 4.3, is $L_i \cdot w_i^{(\beta)}$ still exchangeable? Does its proof rely on the exchangeability of $L_i \cdot w_i^{(\beta)}$?

- In Fig. 5 (second row), the line markers are too densely spaced and are not visually identifiable. I suggest the authors simply remove the markers, as the colors should suffice to differentiate between them.

- One of the contributions of the paper is to extend CRC to handle non-monotonic losses. In parallel, there has been a recent paper dedicated to this topic. The authors should extend their discussion on related works and note their differences.

> Angelopoulos, A. N. (2026). Conformal Risk Control for Non-Monotonic Losses. arXiv preprint arXiv:2602.20151.

- The standard CRC paper was accepted at ICLR 2024, whereas the citation in the manuscript lists the year as 2023. This should be corrected.

- The authors may have used the ```ulem``` package, which converts italic text in the bibliography into underlined text. It would be better to load the package with the ```normalem``` option.

**Limitations:**

Yes

**Strengths And Weaknesses:**

The quality of the paper is high, with some minor weaknesses that I believe are addressable. I am recommending acceptance.

# Strengths

- The paper is a successful application of conformal prediction theory to decision-making problems, which is a very active research area. This unique combination leads to enabling the prescribed policy to satisfy the user-specified risk level with finite sample guarantees. This is something I believe is unique to the policy learning literature.

- The paper introduces several technical novelties that are relatively significant: a unique decision-making problem formulation, extending CRC to non-monotonic losses, and proposing a likelihood-ratio clipped policy and associated calibration mechanism.

- The proposed pipeline is model-agnostic and may find potential application in broader application domains. This is also demonstrated in their experiments (e.g., medical question answering, LLMs)

# Weaknesses

- There are too many (five) assumptions underlying the proposed framework: (i) known Lipschitz constant, (ii) known $\epsilon$-replace one stable, (iii) known loss upper bound $B$; (iv) access to safe policy $\pi_0$, (v) access to some good policy $\pi_t$.
Could the authors clarify how "strict" these assumptions are (e.g., can they be satisfied, can they be verified in practice, etc.) and whether they can be relaxed? I also suggest that the author use a dedicated section in the paper (possibly in the appendix if running out of space) to summarize this.

- The presentation is unclear in some parts.
(i) The second paragraph of Section 4.1 explains the intuition of the proposed method, but it was confusing to me when I first read it. The authors could consider expanding the discussion and providing more concrete and intuitive examples and/or pictorial illustrations of the high-level idea.
(ii) The "Calibrating $\beta$" paragraph of Section 4.2 abruptly introduced importance weights, but it was not mentioned anywhere previously in Section 4. The author should explain why this is needed (I suppose this is to correct for the distribution shift caused by using different policies?).

- The paper didn't discuss theoretical guarantees on the performance of the algorithm over the reward objective. I suggest the authors add a theorem for this or at least discuss it somewhere in the paper.

---

> ### Author Rebuttal · Authors · 2026-03-31
>
> We sincerely thank the reviewer for their interest, time, and constructive comments. We especially appreciate the reviewer’s recognition of the paper’s contribution to both the policy learning and conformal prediction literature, its technical novelties, and its broad applicability. We also thank the reviewer for their feedback on clarifications and discussion that can be improved.
>
> ***Weaknesses:***
>
> **1. Further discussing the practicality of our assumptions:** First regarding the Lipschitz and stability assumptions, we note that in our experiments, we do not observe any violations of the target safety constraint level (ie, no on-average violations of $\alpha$) despite not knowing nor estimating any Lipschitz constants ($K$) nor any replace-one stability values ($\epsilon$). In real applications, these terms may often be negligible (though this should be studied further), and the extent to which they can be relaxed is a promising theory direction.
>
> We are happy to add a dedicated paper section to further clarify the following points:
>
> - i) *Lipschitz assumption:* Please see response to **Reviewier 4Tk7**, point 4.i.
>
> - ii) *Replace-one stability:* Please see response to **Reviewier 4Tk7**, point 4.ii.
>
> - iii) *Known loss upper bound:* While there may be losses of interest where the bound is not known, many practical settings know or assume some “worst case” loss value, $B$. For instance, in our natural language QA experiments (Fig 4), the False Discovery Rate (expected proportion of false subclaims) has $B=1.0$; similarly, in the sequence design experiments (Fig 6), the indicator for whether a given action is infeasible (or unsafe) has $B=1.0$.
>
> - iv) *Safe & optimized policy likelihoods:* The access to a known safe policy’s likelihoods (and same for current optimized policy) may be our most critical assumption, which is why we take care to highlight it. Typically, however, developers often have access to a previous model version (eg, previous LLM version) that has been tested and is considered safe, and it is the safety of a new, untested model (eg, new LLM) that is of main concern.
>
> **2. Further improving presentation:** We appreciate the helpful suggestions, and we also refer to our responses to **Reviewers sfKi**, and **xLRi** point 1, where we outline other changes for improving presentation.
>
> - *i) 4.1:* We will revise the discussion of the proposed method for further clarity and consider adding additional intuitive examples or illustrations beyond the current example (Figure 2).
>
> - *ii) 4.2:* The reviewer is correct that the importance weights are needed to correct for distribution shift between policies. We are revising 4.2. to further clarify why and how importance weights are used.
>
> **3) Analysis of the reward objective:** We will further clarify that while the objective introduced in Eq. (1) is the overall goal, our paper only provides guarantees on the constraint, but not on the reward. Guarantees on improvement in the reward objective have been thoroughly studied in previous conservative optimization literature, such as TRPO (Schulman, et al, 2015). We are happy to add discussion of this.
>
>
> ***Additional Comments:***
>
> **4. Thm 4.3:** No, here $L_i \cdot w_i^{(\beta)}$ is not exchangeable. That is, the weights $w_i^{(\beta)}$ are carefully designed (conformal) weights to handle non-exchangeability. We will further clarify this.
>
> **5. Fig 5 line density:** Thank you for the suggestion, we will remove the markers.
>
> **6. Relation to concurrent pre-print:** Thank you for bringing this to our attention. We have added the following paragraph at the end of our Section 3 (Abbreviated Related Work):
>
> - “Concurrently with this paper, Angelopoulos (2026) also presents CRC guarantees for non-monotonic losses. A primary difference is that paper studies parameterized losses, whereas our paper introduces and focuses on policy control, where the control parameter tunes the policy rather than the loss. Even in the comparable setting of parameterized losses, however, the contributions are distinct, with neither subsuming the other: Relative to the univariate method in Angelopoulos (2026), our method (Section 4) will always be at least as a-priori conservative (e.g., as in Figure 2, right); consequently, whereas that paper's analogous guarantee relies on leave-one-out stability, our results use replace-one stability, which is generally a more relaxed assumption (Bousquet & Elisseeff, 2002; Shalev-Shwartz et al., 2010).”
>
> **7. CRC paper date:** We have changed the date to 2024.
>
> **8. Style package:** Thank yout, we changed to normalem!
>
> Refs:
>
> - Angelopoulos, A. N. (2026). Conformal Risk Control for Non-Monotonic Losses. arXiv preprint arXiv:2602.20151.
>
> - Bousquet & Elisseeff, A. (2002). Stability and generalization. JMLR.
>
> - Schulman, et al. (2015). Trust region policy optimization. ICML. PMLR.
>
> - Shalev-Shwartz, Shamir, Srebro, & Sridharan. (2010). Learnability, stability and uniform convergence. JMLR.

---

> > ### Author Rebuttal · Reviewer_i3pZ · 2026-04-03
> >
> > I thank the authors for the response. My concerns have been addressed, and I hope the author can update the contents of the paper to reflect the discussed changes. I will keep my original score since it's already favorable towards acceptance. I have two additional questions:
> > - Does Theorem 4.3 rely on the exchangeability of $L_i \cdot w^{(\beta)}_i$?
> > - Regarding the upper bound assumption $B$, there are many real-world cases (i.e., when not dealing with indicator losses) where $B$ is intractable. In such cases, how would one go about implementing the proposed framework? What are some limitations?

---

> > > ### Author Response · Authors · 2026-04-04
> > >
> > > Thanks again for your feedback, we are indeed incorporating revisions into the manuscript based on our discussion to improve clarity and accessibility.
> > >
> > > In the general split conformal case, we always work with weighted averages of calibration losses (plus a conservative correction for the unknown test point) $\frac{1}{W} (B + \sum_{i=0}^{n-1} w_i L_i)$, where $W = \sum_{i=0}^n w_i$. How the weight is calculated depends on what is assumed about the test loss $L_n$. If $L_n$ and $L_0, \dots, L_{n-1}$ are exchangeable (a _simplifying_ assumption), then for all $i$, $w_i = 1$. We do not assume $L_n$ and $L_0, \dots, L_{n-1}$ are exchangeable. The correction for non-exchangeability is exactly the conformal weights $w_i$. See [1] to read more about non-exchangeability and weighted conformal methods.
> > >
> > > In the most general case, if it really is true that there is no finite $B$ such that for all $i$, $L_i < B$ almost surely, then there is no method that can guarantee the expected loss of any policy is less than $\alpha$ (without further assumptions), or even finite, since even a single rare observation of an infinite loss value would send the expected loss to $\infty$. Fortunately real world cases where the loss is actually unbounded are actually quite rare, and those that remain can be addressed with simple transforms such as $L_i' = \texttt{sigmoid}(L_i)$, or even by clipping $L_i' = \min(L_i, B)$ for some large $B$. As an analogy, consider that clipping messy real-world labels is a relatively common feature engineering technique in regression to avoid infinite residuals. If such a transformation is used then naturally the guarantee would hold for the expected transformed losses.
> > >
> > > [1] Prinster, D., Stanton, S., Liu, A., & Saria, S. (2024). Conformal validity guarantees exist for any data distribution (and how to find them).

---

### Official Review · Reviewer_xLRi · 2026-03-13

**Soundness:** 3
**Presentation:** 3
**Significance:** 3
**Originality:** 3
**Overall Recommendation:** 4
**Confidence:** 1

**Summary:**

Conformal Policy Control (CPC) is a method for safe exploration that uses a safe reference policy as a probabilistic regulator for an untested optimized policy. It parameterizes the balance between these policies as a likelihood-ratio threshold, which is determined via conformal calibration on data from the safe policy. The framework extends conformal risk control theory to provide finite-sample guarantees for non-monotonic bounded constraint functions. At deployment, the interpolated policy is realized through rejection sampling, allowing the agent to self-regulate within a risk tolerance declared by the user. The authors validate this approach on medical question answering, constrained active learning, and black-box biomolecular sequence optimization.

**Compliance With Llm Reviewing Policy:**

Affirmed.

**Final Justification:**

The authors have largely addressed my concerns during the rebuttal period, thus I keep my initial positive outlook on this paper.

**Key Questions For Authors:**

See weaknesses

**Limitations:**

Yes

**Strengths And Weaknesses:**

### Strengths
- The paper addresses an important problem of safe exploration in high-stakes environments, where violating safety constraints can lead to significant harm or the termination of an agent's deployment.
- The authors advance existing literature by extending CRC to handle non-monotonic bounded loss functions, providing finite-sample guarantees where previous methods were limited to monotonic or asymptotic cases.
- The work is technically sound, featuring a rigorous mathematical derivation of the generalized CRC (gCRC) framework. This theoretical foundation is well-presented and further bolstered by experiments across diverse domains including medical QA and biomolecular engineering which provide convincing evidence that the method effectively controls risk and outperforms previous conformal prediction baselines.
### Weaknesses
- The paper is technically detailed, but the presentation currently hinders accessibility, particularly for readers who may not be deeply embedded in the Conformal Prediction sub-field (like myself). I strongly recommend the authors transition the current Background section into a formal Preliminaries section. Currently, the paper references domain-specific notation without sufficient introduction. For instance, in Section 2.2, the authors discuss calibration loss functions and their monotonicity, yet these terms and their specific functional roles are not thoroughly introduced within the paper’s own problem setup.
- While the paper provides a thorough comparison against recent conformal prediction methods, it lacks comparisons against established safe exploration or constrained optimization baselines. The Related Work section (Section 3 and Appendix A) discusses several alternative approaches (e.g. Safe Bayesian Optimization), but these are not included in the empirical evaluation. Given that these are standard approaches for the tasks presented, the omission makes it difficult to assess the practical utility and efficiency of CPC relative to the broader state-of-the-art in safe exploration. Can the authors clarify why these methods were excluded, or provide results comparing CPC to a standard non-conformal safe exploration baseline?
- As the authors acknowledge, the proposed rejection sampling procedure can become very inefficient, particularly in high-dimensional action spaces where the proposal and target distributions may significantly diverge. While the paper notes that CPC allows for a trade-off with test-time compute, the empirical cost of this trade-off is not clearly reported. Could the authors please provide the empirical sampling efficiency metrics for the presented experiments, specifically the average acceptance rates and the mean number of candidate samples required to yield a single accepted action?

---

> ### Author Rebuttal · Authors · 2026-03-31
>
> We sincerely thank the reviewer for their interest, time, and constructive comments, which we plan to integrate to improve the paper.
>
> **Response to Weaknesses:**
>
> 1) **Improving presentation accessibility:** We will integrate this feedback in an effort to make our paper broadly accessible. For examples of our revisions for clarity, see our response to **Reviewer sfKi**. Eg, we accepted your suggestion of turning Sec 2 into a “Background and Formal Preliminaries” section, revised to standardize terms and ensure notation is clearly defined. We revised the paragraphs you referred to in Section 2.2 to improve clarity and coherence with our paper’s problem setup:
>
> - “CRC is a framework for setting a control parameter, denoted $\lambda$, in a way that controls the expected value of a user-specified loss function, $L_i(\lambda)$. Given exchangeable (e.g., IID) calibration loss functions, $L_1(\cdot), \ldots, L_n(\cdot)$, and test loss function, $L_{n+1}(\cdot)$, the key assumption is that each $L_i(\lambda)$ is monotonically nonincreasing with the control parameter, $\lambda$. [...]
>
> - “For policy optimization, however, it is not immediately clear how CRC could be applied. For many losses of practical interest, one does not have direct access to a tunable control parameter, let alone one known to control the loss monotonically. For instance, for an unknown set of feasible or safe actions, $\mathcal{F}$, consider the indicator for whether an action, $A_t$ is infeasible or unsafe, $L_t := 1 \{A_t \not\in \mathcal{F}\}$. Unlike in the case of prediction sets, where one can set the size of the prediction set $\hat{C}_{\lambda}$ to modulate the loss value, feasibility/safety sets are typically fixed; one cannot directly control $\mathcal{F}$, the set of actions that the environment determines to be feasible or safe. Moreover, for generative agents, large action spaces can make it impossible to obtain a monotonic ordering on possible actions with respect to their true conservativeness. [...]”
>
> 2) **Non-Conformal Baselines:**
> - *Key differences with standard baselines:* Non-conformal conservative optimization baselines have two key limitations relative to CPC we wish to highlight: First, they typically assume a particular model class (which CPC does not); Second, they typically cannot translate a user’s declared risk tolerance, $\alpha$, into imperative instructions for the algorithm, such as a specific divergence penalty or constraint parameter that achieves risk control.
>
> - *Added experiments with constrained optimization baselines:* We first note that DPO (Rafailov, et al 2023) has a KL-divergence penalty term (the DPO beta) that controls its conservativeness (larger values constrain closer to the initial policy). Our Fig 6 experiments set this DPO beta penalty to 0.4 (Appendix Table 5), larger than the default value of 0.1, meaning that the “Uncontrolled” baseline (red) in Fig 6 can be considered a conservative optimization baseline. We now also add additional comparisons to this DPO baseline at the default penalty values used by Rafailov et al. (2023) (0.1 & 0.5) and a more conservative value than typical (0.75). Please see Fig S1 of this anonymous link: https://sites.google.com/view/cpc-icml-author-response
>
>     - *Results:* A reasonable range of KL-divergence penalties make little-to-no difference on the feasibility violation rates (left panel). Only CPC methods allow for controllable feasibility rates in these experiments (left panel). The unresponsiveness of the DPO baselines (red) violation rate to the KL penalty strength highlights how standard conservative optimization methods typically result in a hyperparameter tuning problem: A grid of many penalty parameters needs to be tried before one is found that appears to suffice. In contrast, CPC fully specifies exactly how to set the control parameter to attain a user’s desired risk control.
>
> 3) **Sample Efficiency:** We have added empirical results on test-time compute in Fig S2 of this anonymous link: https://sites.google.com/view/cpc-icml-author-response In the setting of Fig 6, average rejection-sampling acceptance probabilities are between ~1-10% for CPC methods (left). The number of (valid\*) proposal sequences needed by CPC rejection sampling varies between ~20-200 depending on $\alpha$ and the policy improvement step; the worst efficiency is for the intermediate CPC ($\alpha=0.6$, blue) at later optimization rounds where divergence is greatest (middle).
>
> \*Here, valid sequences are parsable as protein sequences. The first two panels assume valid proposals; in contrast, the right panel in Fig S2 plots the total number of generation calls *including invalid sequences.* Non-CPC DPO (red) can over-optimize in later rounds and ultimately waste test-time compute on generating invalid sequences. CPC can help stabilize this issue.
>
> Refs:
> - Rafailov, et al. (2023). Direct preference optimization: Your language model is secretly a reward model. NeurIPS.

---

> > ### Author Rebuttal · Reviewer_xLRi · 2026-04-04
> >
> > I thank the authors for their thorough response. While I still maintain that a comparison with established safe exploration baselines would strengthen the work (as the DPO KL-term results are not entirely convincing), I agree with the highlighted design differences. These distinctions do make a fair and direct comparison with standard baselines somewhat difficult.
> > I will keep my original positive score.

---

> > > ### Author Response · Authors · 2026-04-08
> > >
> > > We thank the reviewer again for their time and thoughtful feedback, which has helped us to improve our paper. We are sincerely grateful for the reviewer’s continued support.
> > >
> > > Additionally, in response to the reviewer’s comment, we are happy to report that **we have added a new comparison with a standard safe exploration baseline in a new Constrained Bayesian Optimization (CBO) setting, at Fig S3 in our anonymous link: https://sites.google.com/view/cpc-icml-author-response**
> > >
> > > We believe that this added experiment helps to more clearly illustrate how CPC addresses a key limitation of standard safe exploration methods (beyond CPC being model agnostic): namely, it illustrates how CPC can circumvent costly hyperparameter tuning by directly translating a user’s declared risk tolerance ($\alpha$) into imperative instructions for the algorithm (eg, clipping divergence at some level).
> > >
> > > Fig S3 compares CPC-controlled CBO to the CBO baseline (Tian, et al., 2024) on a synthetic constrained Bayesian optimization task with unknown binary feasibility constraints. Note that comparing the CBO method with and without CPC illustrates that **CPC can be viewed as complementing (rather than competing with) standard safe exploration methods,** since any policy for exploration can be used as the “optimized policy” in CPC. We evaluate across temperature settings for the CBO’s probabilistic acquisition sampling to simulate an attempt at hyperparameter-tuning the CBO baseline.
> > >
> > > - ***Results:*** Across temperature settings of 0.01 (left column), 0.1 (middle column), and 1.0 (right column), the **CPC-controlled CBO method controls the constraint violation rate** well below the target level of alpha=0.3 (blue, top row). **In contrast, the CBO baseline has much higher constraint violation rates above alpha=0.3**; the baseline's violation rate appears to slightly decrease with increased sampling temperatures (red, top row), but not as a function of alpha, so further hyperparameter tuning would be required to find an acceptable temperature value (if any exists). **Notably, the CPC+CBO method even attains improved reward over the CBO baseline without CPC for all temperature values** (bottom row), possibly due to wasting fewer evaluations on infeasible samples.
> > >
> > > - ***Experiment details:*** The data are 2-dimensional and continuous, and the unknown objective/reward function is the synthetic Townsend function from Townsend (2014) with unknown binary feasibility constraints. We use a Gaussian process from BoTorch (Balandat, et al., 2020) to model the objective function and a classifier to model the unknown constraint. The CBO method's acquisition is done with probabilistic sampling from temperature-scaling on the qLogExpectedImprovement function in BoTorch (Balandat, et a., 2020). The safe policy for CPC is a mean-zero Gaussian with standard deviation set to 1/6th of search space range for ~99.7% coverage. A uniform random baseline is provided for reference (gray). Our implementation of the CBO baseline (Tian, et al., 2024) was based on the following BoTorch notebook: https://botorch.org/docs/notebooks_community/clf_constrained_bo/
> > >
> > > We will add this experiment to our paper’s Appendix. We thank the reviewer again for the helpful feedback.
> > >
> > > Refs:
> > >
> > > - Tian, Y., Zuniga, A., Zhang, X., Dürholt, J. P., Das, P., Chen, J., ... & Luković, M. K. (2024). Boundary exploration for Bayesian optimization with unknown physical constraints. International Conference on Machine Learning (ICML).
> > >
> > > - Townsend, A. (2014). Constrained optimization in Chebfun. https://www.chebfun.org/examples/opt/ConstrainedOptimization.html
> > >
> > > - Balandat, M., Karrer, B., Jiang, D., Daulton, S., Letham, B., Wilson, A. G., & Bakshy, E. (2020). BoTorch: A framework for efficient Monte-Carlo Bayesian optimization. Advances in neural information processing systems, 33, 21524-21538. Note: Implementation of CBO baseline builds on https://botorch.org/docs/notebooks_community/clf_constrained_bo/

---

### Official Review · Reviewer_nDAn · 2026-03-13

**Soundness:** 4
**Presentation:** 4
**Significance:** 4
**Originality:** 4
**Overall Recommendation:** 5
**Confidence:** 3

**Summary:**

This paper introduces conformal policy control [CPC] to tackle the _safe policy improvement problem_.
The goal is to use an initial safe policy $\pi_0$ to find an optimal policies that adheres to a given safety objective.
Given an initial policy $\pi_0$ and a, not necessarily safe, policy $\pi_t$, CPC  iteratively improves $\pi_t$ as a mixture of past policies and $\pi_0$.
The core contribution of this paper is twofold: provide an extension of _conformal risk control_ [CRC] to allow for non-monotonic loss functions and proposing a framework to combine $\pi_0$ and $\pi_t$ to compute an improved $\pi_{t+1}$.

CRC is a statistical framework that extends conformal prediction to allow to control the expected value of a monotone loss function, thereby allowing the scale a trade-off between conservative predictions and optimality.
In this work, the authors extend CRC to allow non-monotonic loss functions, thereby greatly extending the applicability of this statistical framework.

The second part of the contribution is the framework of CPC. The authors assume access to (1) a safe policy $\pi_0$ that adheres to the safety objective and (2) to an optimized policy $\pi_t$. The iterative refinement of the policies $\pi_t$ goes as follows: based on $\pi_t$, a constrained version $\pi_t^{(\beta_t)}$ is computed for which the execution of action is clipped to limit deviation from $\pi_0$. $\beta$ serves as the parameter that controls how large such a deviation may be. $\beta$ is computed by means of a grid search, such that it allows $\pi_t^{(\beta_t)}$ to be optimal, while adhering to the safety objective.

The authors provide an extensive set of experimental evaluations, showing that CPC is able to learn policies in various domains, outperforming existing approaches.

**Compliance With Llm Reviewing Policy:**

Affirmed.

**Final Justification:**

The authors have addressed my concerns hence I will leave my score in favor of acceptance.

**Key Questions For Authors:**

Q1: Can the authors comment on how the quality of the safe policy influences learning in the CPC framework?

Q2: Can $\pi_0$ be defined by means of domain-knowledge as "the most conservative" policy?

**Limitations:**

yes

**Strengths And Weaknesses:**

# Strengths

This work tackles a research topic that is of utmost interest, not only to the research community. Providing guarantees in expectation is a necessary step to allow for the application of ML in various domains where safety is non-negoiable.

The authors provide extensive supplementary material, covering proofs, details of the methodology, and details w.r.t. the experimental evaluation.

Extending CRC to allow for non-monotonic loss functions is useful not only in the context of this work, but might be useful in other context as well.

# Weaknesses

A section commenting on the assumptions of the quality of the (initial) safe policy would improve the paper. It is not clear whether it has to fulfill some level of optimality in order for CPC to be applicable. In optimal control problems, a policy that does not necessarily need to meet any progress objectives might just stall, a classifier might output prediction sets that are impractical, etc.
Additionally, the effect of the _conservativeness_ of $\pi_0$ on the training with CPC could be studied.

# Minor

In the definition of $\epsilon$-replace-one stability, the applications of h are missing closing multiset paranthesis.

---

> ### Author Rebuttal · Authors · 2026-03-31
>
> We sincerely thank the reviewer for their interest, time, and constructive comments. We especially appreciate and are humbled by the reviewer’s opinion that the paper “tackles a research topic that is of utmost interest, not only to the research community.”
>
> **Response to Weaknesses and Questions:**
>
> **1. (Weakness 1 and Question 1) Effect of safe policy quality on optimization performance:** We are happy to add further discussion to the paper on the effect of the quality of the safe policy and how this might affect CPC’s utility in practice. However, first to be clear, to achieve the target risk control guarantee on the safety constraint, CPC requires no assumptions on the safe policy other than (i) access to its likelihoods and (ii) that it is indeed safe (ie, that the expected loss under the safe policy is indeed no greater than $\alpha$).
>
> With respect to reward optimization and sample efficiency, however, it is very plausible that better optimization of the safe policy, and less divergence of the safe policy from the optimized one, will result in better average reward and greater efficiency of the CPC-controlled agent. This becomes clear, for instance, if we assume that the optimized policy $\pi_t$ is truly optimal and itself safe. That is, if $\pi_t$ is the globally optimal safe policy, then the constrained policy $\pi_t^{(\beta)}$ will approach the optimal policy as the safe policy does so, as $\pi_0 \rightarrow \pi_t$. We also note that we provide analysis of accept-reject sampling efficiency considerations and strategies for improving sampling in Appendix C.2; in those analyses, one can see that the acceptance probability increases if the safe and optimized policy are closer together.
>
> *Further note on updating the safe policy:* While throughout our submission we assumed for simplicity a single fixed safe policy, $\pi_0$, an interesting related question is how one could update the safe policy over time to avoid stalling (which the reviewer points out could be an issue). For instance, at some later time one may have enough reason or evidence to justify resetting their safe policy as $\pi_0 \leftarrow \pi_{t’}^{(\beta)}$ or even as $\pi_0 \leftarrow \pi_{t’}$, for some $t’>0$. Developing such a procedure could be a promising direction for improving the ability to reliably improve on progress objectives with CPC. We will add discussion on this point to the paper.
>
> **2. (Question 2) Defining $\pi_0$ by domain-knowledge as the “most conservative” policy:** The safe policy $\pi_0$ could certainly be defined via domain knowledge, when a particular default safe policy is known. Such examples might include defaulting to the standard of care in medical treatment advising, standard library construction techniques in biomolecular design, or citing only trusted sources in retrieval-augmented generation. $\pi_0$ could be defined as the “most conservative” policy as long as it is indeed safe (true risk below $\alpha$), but $\pi_0$ does not necessarily need to be the most conservative policy. That is, among the set of true safe policies, any would suffice to use as $\pi_0$ in CPC, $\pi_0$ need not be the most conservative one.
>
> **3. (Minor) $\epsilon$-replace-one stability def closing parenthesis:** Thank you for pointing this out this typo, we have fixed it!

---

> > ### Author Rebuttal · Reviewer_nDAn · 2026-04-02
> >
> > The authors have addressed my questions. As it seems that concerns from the other reviewers w.r.t. clarity are being actively worked on/have been resolved by the authors, I will leave me score as it is.

---

> > > ### Author Response · Authors · 2026-04-03
> > >
> > > Thank you very much again for your time, feedback, and continued support. Cheers!

---

### Official Review · Reviewer_4Tk7 · 2026-03-22

**Soundness:** 4
**Presentation:** 4
**Significance:** 3
**Originality:** 4
**Overall Recommendation:** 5
**Confidence:** 4

**Summary:**

# **Summary**

**Overall, a general area assessed by this paper** is safe exploration and constrained decision-making under uncertainty, with applications spanning reinforcement learning, LLM deployment, and black-box optimization.

**The paper investigates the key problem** of how to safely deploy an optimized (but potentially unsafe) policy while guaranteeing that constraint violations remain below a user-specified risk threshold, *without requiring hyperparameter tuning or strong modeling assumptions*.

The authors propose **Conformal Policy Control (CPC)**, a method that:

* Uses a **safe reference policy** and an **optimized policy**
* Interpolates between them via **likelihood-ratio clipping**
* Calibrates the interpolation parameter using **conformal risk control (CRC)**
* Provides **finite-sample guarantees** on constraint violations

A key theoretical contribution is extending CRC to **non-monotonic loss functions**, enabling guarantees in settings where standard conformal methods fail. The method operates entirely at **test time**, allowing flexible reuse of models under different risk tolerances.

Empirical results across medical QA, active learning, and sequence optimization demonstrate that CPC achieves **tight risk control** and can even **improve performance under constraints**.

**Compliance With Llm Reviewing Policy:**

Affirmed.

**Key Questions For Authors:**

# **Key Questions for Authors**

1. **Safe policy assumption**
   * How realistic is access to a truly safe π₀?
   * Can CPC operate with partially safe or estimated-safe baselines?

2. **Approximation robustness**
   * How sensitive are guarantees to approximations in conformal weights?
   * Can the gap between theory and practice be quantified?

3. **Exploration vs suppression**
   * How does CPC affect discovery of **low-probability, high-reward actions**?
   * Can the method be adapted to preserve useful novelty?

4. **Sequence-level behavior**
   * How does CPC behave in autoregressive generation?
   * Are there guarantees at the **sequence level**, not just per action?

5. **Scalability**
   * How does rejection sampling scale for large vocabularies and long horizons?

6. **Comparison to RL baselines**
   * How does CPC compare empirically to constrained RL and offline RL methods?

7. **Distribution shift and recalibration**
   * How often must CPC be recalibrated under non-stationary environments?

**Limitations:**

# **Limitations**

1. Requires a safe baseline policy
2. Relies on approximations that may weaken guarantees
3. Provides only marginal (not conditional) guarantees
4. Depends on assumptions (Lipschitz, stability) that may not hold
5. May be inefficient in large or divergent action spaces
6. Limited benchmarking against broader baselines
7. Suppresses novel predictions outside safe policy support
8. Distorts sequence-level distributions in autoregressive settings

**Strengths And Weaknesses:**

# **Strengths and Weaknesses**

## **Strengths**

### 1. Clean and principled formulation

The paper presents a **conceptually elegant solution** to safe policy improvement by reframing it as **distributional interpolation with calibrated likelihood ratios**, avoiding ad hoc divergence penalties.

---

### 2. Significant theoretical contribution

The extension of conformal risk control to **non-monotonic bounded losses** is novel and addresses a known limitation in prior work, broadening applicability.

---

### 3. Direct alignment with user objectives

CPC allows users to specify **risk tolerance α directly**, removing the need to tune indirect control parameters. This is both practical and conceptually appealing.

---

### 4. Test-time modularity and flexibility

The method operates entirely at inference time:

* No retraining required
* Same policies reusable under different risk levels
* Flexible trade-off between safety and performance

---

### 5. Strong empirical validation across domains

Experiments demonstrate:

* Reliable finite-sample risk control (e.g., Figure 4 on page 6)
* Improved or competitive performance
* Evidence that moderate safety constraints can improve efficiency

---

### 6. Practical algorithmic design

Likelihood-ratio clipping combined with rejection sampling is:

* Simple
* General
* Compatible with large action spaces

---

## **Weaknesses**

### 1. Strong dependence on a safe reference policy

The method assumes access to a **safe policy π₀** satisfying the constraint (page 5), which may:

* Be difficult to obtain in practice
* Limit applicability in truly novel or high-risk domains

---

### 2. Theory–practice gap in conformal weighting

Theoretical guarantees rely on exact conformal weights, but implementation uses approximations (page 6), introducing an unclear gap between:

* Proven guarantees
* Actual deployed behavior

---

### 3. Scalability and efficiency concerns

Rejection sampling may become inefficient when:

* The optimized policy diverges significantly from π₀
* Action spaces are high-dimensional or continuous

---

### 4. Assumptions may be unrealistic in deep learning settings

Guarantees rely on:

* Lipschitz continuity
* Replace-one stability

These are difficult to verify and may not hold for neural policies.

---

### 5. Limited comparison with broader safe RL literature

The evaluation focuses on conformal baselines but lacks comparison to:

* Constrained RL methods
* Offline RL safety approaches

---

### 6. Marginal (not conditional) guarantees

The method provides **population-level risk control**, not per-instance guarantees, limiting applicability in high-stakes decision-making.

---

### 7. **Novel prediction suppression (key concern)**

The likelihood-ratio clipping mechanism inherently biases the deployed policy toward the support of the safe policy π₀.

* Actions where πₜ ≫ π₀ are systematically suppressed
* This discourages **rare but potentially high-reward or innovative behaviors**
* The method effectively enforces a **support overlap constraint**, which may:

  * Limit exploration beyond known safe regions
  * Bias outputs toward conservative, previously observed behaviors

This is particularly concerning in:

* LLM generation (novel reasoning or phrasing)
* Scientific discovery (e.g., molecule design)

While CPC aims to enable safe exploration, it may inadvertently **over-constrain exploration in precisely the regimes where innovation is most valuable**.

---

### 8. **Long-sequence joint probability distortion (key concern)**

CPC operates via likelihood-ratio clipping at the action level, but in autoregressive models (e.g., LLMs), actions form sequences.

* Sequence probabilities factorize as products of token-level probabilities
* Per-step clipping introduces **nonlinear distortions in the joint sequence distribution**

Consequences include:

* The resulting policy may not correspond to a **coherent sequence-level distribution**
* Local accept/reject decisions can:

  * Break long-range dependencies
  * Bias toward high-probability prefixes
  * Reduce fluency or logical consistency

Additionally:

* Theoretical guarantees are **per-step**, while evaluation is often **sequence-level**, creating a mismatch between:

  * What is controlled
  * What matters in practice

This issue is especially critical for LLM applications and is not explicitly addressed.

---

> ### Author Rebuttal · Authors · 2026-03-31
>
> We sincerely thank the reviewer for their interest, time, and constructive comments, which we plan to integrate to improve the paper.
>
> Anonymous link for figs referenced below: https://sites.google.com/view/cpc-icml-author-response
>
> ***Responses to Weaknesses(=W), Questions(=Q), & Limitations(=L) in order of W:***
>
> **1. (W1,Q1,L1) Safe policy assumption:** This assumption is key so we take care to highlight it. However, developers often have access to a previous model (eg, prior LLM version) that is considered safe from deployment testing.
>
> **2. (W2,Q2,L2) Conformal weighting approx:** In our experiments, we do not observe any statistically-significant violations of CPC’s target risk level. In theory, robustness guarantees could be obtained by extending analysis from Barber, et al. (2023) on their total-variation distance “coverage gap” to general loss functions. Tighter bounds might be obtained by analyzing how well the mixture of past policies ($\pi_{0:t-1}^{mix}$) approximates the true joint distribution ($\pi_{0:t-1}$), or via other estimation approaches. We will add discussion on this.
>
> **3. (W3,Q5,L5) Scalability & efficiency:** Rejection sampling still works in large action spaces. As noted in Sec 4.2, sample efficiency may decrease with divergence of the safe & optimized policies. We provide sampling analysis in Appendix C.2 & add empirical test-time compute analysis at Fig S2 in the link above.
>
> **4. (W4,L4) Practicality of Lipschitz and replace-one stability:**
>
> - i) *Lipschitz assumption:* For CPC, Lipschitz smoothness is determined by known quantities (the agents’ own policies) rather than the unknown constraint function, which makes the assumption more practical. That is, as we note in the paragraph following Theorem 4.3, the Lipschitz assumption therein is on $L_i \cdot w_i^{(\beta)}$ as a function of $\beta$. So, Lipschitz for CPC is determined by the weights $w_i^{(\beta)}$, which are in turn a function of the ratio between safe and optimized policies. For CPC, one could in principle thus test for or estimate the Lipschitz constant, $K$, which could be an important direction for future study.
>
> - ii) *Replace-one stability:* First note that the replace-one (RO) assumption is on the CPC selection of $\hat\beta$ (or for gCRC, on $\hat\lambda$), not on the agent policies or black-box model (eg, not on $\pi_t$). Broadly, however, $\epsilon$-RO stability is considered a relaxed assumption. Classic results establish that an algorithm does not overfit if and only if it is RO stable (Sec 13.2 of Shalev-Shwartz & Ben-David (2014)). A more common assumption is leave-one-out (LOO) stability (w.r.t. removing one datapoint), and RO-stability is more relaxed than LOO: LOO-stable algos are RO stable, but there are RO-stable algos that are not LOO stable (see Soloff, et al. (2024) Sec 5.3.3; Bousquet & Elisseeff (2002)).
>
> **5. (W5,Q6,L6) Comparison to broader safe RL lit:** Please see our response to **Reviewer xLRi**, point 2. There, we describe new experiments to compare to constrained DPO baselines, in Fig S1 at the link above.
>
> **6. (W6,L3) Marginal vs conditional guarantees:** Advancing toward conditional guarantees is an important direction for future work. We will extend our current discussion on this in the 2nd paragraph of Section 6 (Discussion).
>
> **7. (W7,Q3,L7) Novel prediction suppression due to safety bias:** Some amount of safety bias is likely needed to ensure that safety constraints are respected. While over-constraining is possible in some settings, our experiments demonstrate that CPC does not always reduce performance, and can even *improve* it. Relative to uncontrolled baselines, CPC attains lower MSE in constrained active learning on the Airfoil data (Fig 5, middle) and CPC with $\alpha=0.8$ attains higher average reward (Fig 6, middle).
>
> **8. (W8,Q4,L8) Long-sequence joint probabilities:** To clarify, while CPC could in principle be applied to token-level actions, in this paper we focus on applying CPC to filter *sequence-level* actions. Accordingly, in our methods and experiments, there is no issue of long-sequence joint probability distortion. The guarantees thus apply at the sequence-action-level, the same level as evaluations.
>
> **9. (Q7) Distribution shift & recalibration:** In non-stationary environments there could be external distribution shifts. As we discuss in Section 6 (Discussion), monitoring methods such as conformal martingales can be used to detect such shifts to trigger recalibration on new data. Integration with monitoring & robustness results are promising future directions.
>
> Refs:
> - Barber, et al. (2023). Conformal prediction beyond exchangeability. The Annals of Statistics.
>
> - Bousquet & Elisseeff, A. (2002). Stability and generalization. JMLR.
>
> - Shalev-Shwartz & Ben-David. (2014). Understanding machine learning: From theory to algorithms. Cambridge university press.
>
> - Soloff, Barber, & Willett. (2024). Bagging provides assumption-free stability. JMLR.

---

> > ### Author Rebuttal · Reviewer_4Tk7 · 2026-04-04
> >
> > Thanks to the authors for the detailed rebuttal and additional results. The clarifications around conformal weighting and the added empirical analysis are helpful and improve the paper.
> >
> > That said, a few concerns remain:
> >
> > (1) The safe policy assumption still feels somewhat strong in practice. In settings with distribution shift or fast iteration, a previously deployed model may not reliably remain “safe.” It would help to better discuss robustness when this assumption is violated.
> >
> > (2) While the discussion on approximation is helpful, there is still a gap between the theoretical guarantees and the practical conformal weighting used in experiments.
> >
> > (3) On scalability, rejection sampling may become inefficient when policies diverge significantly. Some clearer characterization of this regime would strengthen the work.
> >
> > (4) The additional comparisons are appreciated, but the novelty relative to broader safe RL / alignment methods could still be more clearly articulated.
> >
> > Overall, the rebuttal improves clarity, and the problem is important, but the above points would benefit from further strengthening.

---

> > > ### Author Response · Authors · 2026-04-08
> > >
> > > Thank you again for your feedback. We’re glad that our clarifications and added experiments further improved your view of the paper.
> > >
> > > **(1) Safe policy assumption & discussion of robustness to distribution shifts:** We are happy to further expand our existing discussion. Eg, in Methods 4.2, we have revised our exposition of the safe policy assumption to the following (additions in italics):
> > >
> > > - **Assumption 1: An Initial Safe Policy** Assume access to an initial safe policy, $\pi_{0}$, for which the true risk is controlled at the desired level $\alpha$ [...]. *Note that this assumption is weaker than it may appear: any existing policy (a previous deployment, a domain default, or even a uniform baseline) qualifies as $\pi_0$ so long as its risk is at most $\alpha$. When little is known, $\alpha$ can be set loosely and any reasonable baseline suffices; the strength of the resulting guarantee scales with how much is actually known about what is safe.*
> > >
> > > Re robustness, we are happy to expand our current discussion. For easy reference, here is a relevant excerpt from our Sec 6 (Discussion):
> > >
> > > - “Our guarantees also depend on the stability of the context distribution. If the context distribution begins to shift then recalibration may be necessary, or some risk control gap may need to be tolerated (Barber et al., 2023). Monitoring methods based on conformal martingales offer a principled approach to detecting such shifts (Prinster et al., 2025).”
> > >
> > > We will make an effort to revise to further clarify the meaning of “recalibration” (eg, via online learning of the shift’s new importance weights or via adaptive methods) and the types of risk control gaps known in the literature (eg, total variation distance).
> > >
> > >
> > > **(2) Gap between CPC theory and practical conformal weighting:** We are glad our clarifications on this point have been helpful and we have integrated them into the paper. Overcoming or analyzing the estimation error between the theoretical (and intractable) conformal weights and the practical weights is an important direction for future study.
> > >
> > >
> > > **3) Clearer characterization of sample efficiency with policy divergence:** ​​We kindly refer to Fig S2 of our link: https://sites.google.com/view/cpc-icml-author-response In the setting of Fig 6, average sampling acceptance probabilities are between ~1-10% for CPC methods (left). The number of valid (ie, parsable as protein sequences) proposals needed by CPC rejection sampling varies between ~20-200 depending on $\alpha$ and the policy improvement step; the worst efficiency is for the intermediate CPC ($\alpha=0.6$, blue) at later optimization rounds where divergence is greatest (middle). The right panel in Fig S2 plots the total number of generation calls *including invalid sequences,* and finds that unconstrained DPO baseline can waste generation on many invalid sequences in later rounds, and CPC can help stabilize this issue.
> > >
> > > We also refer to our Appendix C.2 for analysis of sampling methods. CPC is not restricted to rejection sampling, and designing efficient proposal and sampling algorithms is an independent direction of immense practical significance.
> > >
> > >
> > > **(4) Further clarifying novelty relative to safe RL & alignment methods:** Non-conformal conservative optimization methods in this space have two key limitations relative to CPC we wish to highlight: First, they typically assume a particular model class (which CPC does not); Second, they typically cannot translate a user’s declared risk tolerance, $\alpha$, into imperative instructions for the algorithm, such as a specific divergence penalty or constraint parameter that achieves risk control. The experiments we added illustrated these limitations with DPO baselines with varying divergence penalties.
> > >
> > > Additionally, we are now happy to report **additional experiments comparing to a standard Constrained Bayesian Optimization (CBO) method (Tian, et al., 2024).** Please see Fig S3 at this link: https://sites.google.com/view/cpc-icml-author-response
> > >
> > > We compare CPC-controlled CBO to the CBO baseline (without CPC) on a synthetic Bayesian optimization task with unknown feasibility constraints. We evaluate across temperature settings for the CBO’s probabilistic acquisition sampling to simulate an attempt at hyperparameter-tuning the CBO baseline. Across all settings, the proposed CPC-controlled CBO method controls the constraint violation rate well below the target level of alpha=0.3 (blue, top row). In contrast, the CBO baseline has much higher constraint violation rates above alpha=0.3; its violations appears to decrease with increased sampling temperatures (red, top row), but not as a function of alpha, so further tuning would be required. The CPC+CBO method attains improved reward over the CBO baseline without CPC for all temperature values (bottom row), possibly by selecting fewer infeasible samples.
> > >
> > > Ref: Tian, et al. (2024). Boundary exploration for Bayesian optimization with unknown physical constraints. ICML

---

### Official Review · Reviewer_sfKi · 2026-03-23

**Soundness:** 3
**Presentation:** 1
**Significance:** 3
**Originality:** 3
**Overall Recommendation:** 4
**Confidence:** 4

**Summary:**

The paper proposes a mechanism to provide a risk control guarantee on new untested policy in the decision-making scenarios. An example to make this clear is the post-training of language models where an initial policy is fined-tuned to achieved further tailored performance. However, the problem paper addresses is how does one transform this policy so as desirable safe condition that is already certified on the base policy does not deteriorate. In the context of language model, a standard mechanism is to put some penalty on the post-training loss based on divergences like KL. The paper's proposal is to design a policy that can interpolate between the safe tested policy and the untested one where the degree of interpolation is defined by a control parameter (a likelihood ratio), and this control parameter can be tuned to provable put the desired risk control guarantee on the interpolated policy. In terms of methodological advancement, the paper proposes a risk control procedure to arbitrary non-monotonic losses to achieve this goal.

**Compliance With Llm Reviewing Policy:**

Affirmed.

**Final Justification:**

My concerns with the paper are mostly with respect to clarity, which the authors have agreed to see further. Hence no major concerns remain. I'm not against the paper getting accepted.

**Key Questions For Authors:**

None

**Limitations:**

yes

**Strengths And Weaknesses:**

Strengths:
1. The paper's contribution is solid, and encompasses several aspects like originality, soundness, and significance. I like the discussion around lines 123-135, and in my opinion, the paper provides a conceptual reframing on achieving risk control in policy adaptation in decision-making problems. I also like the discussion around exploration-exploitation trade-off, and the proposal of interpolated policy captures that.
2. The paper also advances the standard risk control procedure that has gained quite some attention recently in the community to non-monotonic losses. Under the assumption of algorithmic stability and Lipschitz (ness) of the loss function in the control parameter, the risk control procedure can be extended to apply to non-monotonic losses. In a concurrent work, similar problem was addressed [https://arxiv.org/abs/2602.20151], and I'd be curious to hear any comments on the connection here.


Weaknesses:
1. While I do not see any major weaknesses scientific contribution wise, I strongly believe the paper can be improved tremendously writing-wise. Currently, the paper does not seem to be nicely organised. Let's start with the abstract itself: the motivation of the paper is done rightm, how much behavior change is too much? However the paper then switches to using terms that are not directly clear from the context: optimized but untested policy, conformal calibration, safe policy, conservative optimization. There is a very simple reframing to articulate the same argument: "How much behavior change is too much? In this work, we develop a risk control framework that regulates this trade-off. Our framework assumes access to a reference policy that has been tested for desired safety (or risk) standards, and we use it [well, I hope the idea is clear]. In doing so, we extend the standard risk control framework towards non-monotonic losses which might be of independent interest." The point I'm trying to make is that the abstract is too much information without clear coherence. Similarly with the introduction.
2. The paper also needs a unifying section or a paragraph to state all the notation and symbols it is using. The paper also interchangeably uses different examples of policy or sequential decision-problems, like LLM (context), black-box optimization, etc. The paper could simply use an abstract notation to set the problem: the problem we consider has the space $\mathcal{X}$, action space $\mathcal{A}$, and reward and losses. An agent sees $X$ and uses the policy $\pi(A|X)$ to take an action, and the goal is to maximize the reward with risk control. Then eventually the paper can chart out one or two examples to further explemplify the abstract problem formulation as described above. Right now, everything is mixed together. I also believe Section 4.2 should proceed before Section 4.1. First the paper can introduce the formulation of policy control with calibrating $\beta$ as the technical problem, and then can bridge towards the general approach of risk control with non-monotonic losses.
3. I like the writing of the proofs in the Appendix though, as they are nicely organized. But if my opinion is to be considered, I believe the main paper needs a major re-writing.

Overall, I do not have any significant issues with the scientific contributions of the work. But it needs re-writing.

---

> ### Author Rebuttal · Authors · 2026-03-31
>
> We sincerely thank the reviewer for their interest, time, and constructive comments, which we plan to integrate to improve the paper. We especially appreciate the reviewer’s recognition that the “contribution is solid, and encompasses several aspects like originality, soundness, and significance” and their suggestions on improving clarity.
>
> *Discussion of concurrent preprint:* Please see response to **Reviewer i3pZ**, point 6.
>
> ***Response to Weaknesses (Presentation Clarity):***
>
> We appreciate the suggestions for improving presentation clarity. We recognize the importance of making our paper as clear and accessible as possible, and we will make an earnest effort to incorporate your feedback. For example, we are making the following changes (open to further revision):
>
> **1. Abstract:** We are revising our abstract based on your suggestions, so that terms either are clearer from context or they invite further reading for gradual exposition. For example, we revised the part you suggested edits to as follows (italic for modifications):
>
> - “How much behavior change is too much? *We develop a framework to control this tradeoff, by using any pre-verified* safe reference policy as a probabilistic regulator for any optimized but untested policy. *To do so, we introduce conformal methods that calibrate* how aggressively the new policy can act […]”
>
> Eg, notice that “pre-verified” conveys your clarification that the reference policy “has been tested for desired safety (or risk) standards”; and, by saying “we introduce conformal methods that calibrate” it is implied that readers unfamiliar with conformal methods should read further for explanation.
>
> **2. Intro:** We have also revised the Intro section to standardize and give intuition for terms, clarify examples, etc. (Note that we reserve formal definitions for later in Section 2, which we have updated to a “Background and Formal Preliminaries” section.) For example, we revised the 2nd paragraph of the Intro as follows (italic for modifications):
>
> - “Consider an agent that observes *contexts*, takes actions, and receives both rewards and constraint feedback. [...] For example, a language model *responding to medical questions aims to give answers that are as helpful as possible while avoiding false claims. In biomolecular engineering, a generative model proposing improvements to exemplar molecules aims to maximize the efficacy of its designs while ensuring that they are synthesizable. In each such setting, the agent updates an optimized policy for future performance but does not yet know if it is safe. From past experience, the agent usually has a default safe policy it knows controls the risk, but playing it safe may be far from optimal. To compromise, the agent could select an intermediate policy that weights historical actions according to how important they are to the deployed policy, relative to their previously observed frequency. However, these ``importance weights''* depend on the deployed policy, [...]”
>
> Eg, note that we standardize to using “context” (vs state), the meaning of “optimized” and “safe” policies is clarified, and provide intuition for “importance weights” while marking them in quotations to indicate they are a technical term to be explained later.
>
> **3. Unifying section for notation:** We have updated Section 2 from “Background” to a “Background and Formal Preliminaries” section for this important purpose. While we would like to kindly note that the first paragraph of the original Background section did introduce the context space $\mathcal{X}$, the action space $\mathcal{A}$, the reward $R_t\in\mathbb{R}$, the loss $L_t\in \mathbb{R}$, and the policy $\pi(A\mid X)$ as the reviewer suggests, we now begin this paragraph with boldfont label “**Notation:**” to ensure it is not missed. We also note that the running medical QA and biomolecular design examples are explained further in 1st paragraph of Section 2.1, but we will expand these examples for further clarity. We acknowledge that we used terms from different domains (eg, “context” and “state”); we have standardized this where possible (eg, sticking with “context”).
>
> **4. Ordering of Sections 4.1 and 4.2:** While we prefer to keep the current ordering of Section 4.1 before 4.2, we acknowledge the reviewer’s feedback and will revise to smooth the flow. Eg, we realize that our initial version may have implied that the CPC (4.2) is a subset of gCRC (4.1); however, this is not the case. This is because CRC & gCRC assume that the control parameter directly tunes some loss function (not the policy), while CPC relaxes this to allow the control parameter to tune the policy or probabilities. Also, some technical details in Section 4.2 are more complex than those in 4.1 (eg, the permutation importance weights for handling non-exchangeable data), so we prefer to begin more gently with 4.1. Overall, we really appreciate the reviewer’s feedback on these points and will certainly revise to further clarify.

---

> > ### Author Rebuttal · Reviewer_sfKi · 2026-04-02
> >
> > Thanks for the rebuttal. My concerns with the paper are mostly with respect to clarity, which the authors have agreed to see further. Hence no major concerns remain. I'll update my score a bit.

---

> > > ### Author Response · Authors · 2026-04-02
> > >
> > > Thanks again for your feedback on the manuscript, your perspective has been very helpful in identifying ways to make our paper accessible to a broad audience. Cheers!

---

### Decision · Program_Chairs · 2026-04-30

**Decision:**

Accept (spotlight)

**Comment:**

The paper provides a conceptually elegant reframing of safe policy improvement as a distributional interpolation with calibrated likelihood ratios. The reviewers recognize the originality and technical soundness, highlighting the significant theoretical contribution of extending conformal risk control to handle non-monotonic bounded loss functions. The method is practical and interpretable. It is supported by strong empirical validation across high-stakes domains. The paper should clearly be accepted.